# Measurements to determine mixing state of black carbon emitted from the 2017/2018 California wildfires and urban Los Angeles

Joseph Ko[1], Trevor Krasowsky[1,*], George Ban-Weiss[1]

[1]Department of Civil and Environmental Engineering, University of Southern California, Los Angeles, 90089, USA

[*]now at: SpaceX, 1 Rocket Rd, Hawthorne, CA 90250 USA

*Correspondence to*: George Ban-Weiss (banweiss@usc.edu)

**Abstract.** The effects of atmospheric black carbon (BC) on climate and public health have been well established, but large uncertainties remain regarding the extent of BC's impacts at different temporal and spatial scales. These uncertainties are largely due to BC's heterogeneous nature in terms of its spatiotemporal distribution, mixing state, and coating composition. Here, we seek to further understand the size and mixing state of BC emitted from various sources and aged over different timescales using field measurements in the Los Angeles region. We measured refractory black carbon (rBC) with a single-particle soot photometer (SP2) on Catalina Island, California (~70 km southwest of downtown Los Angeles) during three different time periods. During the first campaign (September 2017), westerly winds were dominant and measured air masses were representative of well-aged background over the Pacific Ocean. In the second and third campaigns (December 2017, November 2018), atypical Santa Ana wind conditions allowed us to measure biomass burning rBC ($BC_{bb}$) from air masses dominated by large biomass burning events in California and fossil fuel rBC ($BC_{ff}$) from the Los Angeles basin. We observed that emissions source type heavily influenced both the size distribution of rBC cores and rBC mixing state. $BC_{bb}$ had thicker coatings and larger core diameters than $BB_{ff}$. We observed a mean coating thickness ($CT_{BC}$) of ~40–70 nm and count mean diameter (CMD) of ~120 nm for $BC_{bb}$. For $BC_{ff}$, we observed $CT_{BC}$ of ~5–15 nm and CMD of ~100 nm. Our observations also provided evidence that aging led to increased $CT_{BC}$ for both $BC_{bb}$ and $BC_{ff}$. Aging timescales < ~1 d were insufficient to thickly coat freshly-emitted $BC_{ff}$. However, $CT_{BC}$ for aged $BC_{ff}$ within aged background plumes was ~35 nm thicker than $CT_{BC}$ for fresh $BC_{ff}$. Likewise, we found that $CT_{BC}$ for aged $BC_{bb}$ was ~18 nm thicker than $CT_{BC}$ for fresh $BC_{bb}$. The results presented in this study highlight the wide variability in BC mixing state and provide additional evidence that emissions source type and aging influence rBC microphysical properties.

## 1 Introduction

Atmospheric black carbon (BC) is a carbonaceous aerosol that can result from the incomplete combustion of carbon-containing fuels. Major energy-related sources of BC include vehicular combustion, power plants, residential fuel-use, and industrial processes. Biomass burning, which can be either anthropogenic or natural, is another significant BC source. BC is a pollutant of particular interest for two main reasons: (1) it absorbs solar radiation, which results in atmospheric warming (Ramanathan and Carmichael, 2008), and (2) it is associated with increased risk of cardiopulmonary morbidity and mortality (World Health Organization, 2012). Regarding its effect on climate, BC is widely considered to be the second strongest contributor to climate warming, after carbon dioxide (Bond et al., 2013). Although it has been established that BC is a strong radiative forcing agent in Earth's atmosphere, there remains considerable uncertainty about the extent to which BC affects Earth's radiative budget, from regional to global scale (IPCC, 2013; Bond et al., 2013).

Since the lifetime of BC is relatively short (~days to weeks), the spatiotemporal distribution of BC is highly heterogeneous and thus difficult to quantify (Krasowsky et al., 2018). The quantification of where and when BC is emitted around the world is also a challenging task that causes significant uncertainty (Bond et al., 2013). In addition to the difficulties that come with tracking the emissions and distribution of BC, there are complex physical and chemical processes that govern the transformation of BC in the atmosphere, which ultimately impact its climate and health effects. These atmospheric processes, in addition to the emissions source type, influence the BC mixing state in a highly dynamic manner. A BC particle that is physically separate from other non-BC aerosol species is considered *externally mixed*. On the other hand, BC is considered *internally mixed* if it is physically combined with another non-BC aerosol species (Bond et al., 2006; Schwarz et al., 2008a). As freshly emitted BC particles are transported in the atmosphere, they can obtain inorganic and organic coatings from either gaseous pollutants that condense onto the BC, oxidation reactions on the BC surface, or the coalescence of other aerosol species onto the BC, making them more internally mixed (He et al., 2015). In general, the *mixing state* of BC describes the degree to which BC is internally mixed (Bond et al., 2013). The BC mixing state near the point of emission as well as the evolution during aging in the atmosphere of the mixing state can vary widely, depending on the source of emissions and atmospheric context.

The evolution of rBC mixing state as BC ages in the atmosphere is crucial to understand for two reasons. First, it has been shown that non-refractory coatings on BC can enhance its absorption efficacy, implying that internally mixed BC with thick coatings can have stronger warming potential in the atmosphere compared to uncoated or thinly-coated BC (Moteki and Kondo., 2007; Wang et al., 2014). Second, coatings on BC can alter the aerosol's hygroscopicity and effectively shorten its

lifetime by increasing the probability of wet deposition (McMeeking et al., 2011a; Zhang et al., 2015). In short, freshly emitted BC particles are generally hydrophobic, but coatings acquired during the aging process can make BC-containing particles hydrophilic, and therefore, more susceptible to wet deposition. Thus, uncertainties in the evolution of rBC mixing state directly translate to uncertainties regarding BC's impact on Earth's climate due to both the radiative impact per particle mass and spatiotemporal variation of atmospheric BC loading.

Although there have been a number of laboratory experiments (Wang et al., 2018; He et al., 2015; Slowik et al., 2007; Knox et al., 2009) and field campaigns (Krasowsky et al., 2018; Metcalf et al., 2012; Cappa et al., 2012; Schwarz et al., 2008a) studying rBC mixing state, there is considerable variability in results. For example, field studies in China suggest that the mass absorption cross-section (MAC) of BC that has aged for more than a few hours should be enhanced by a factor of ~2 (Wang et al., 2014), while other studies in California reported an absorption enhancement factor of ~1.06 (Cappa et al. 2012) and ~1.03 (Krasowsky et al., 2016). Preceding these studies, Bond et al. (2006) suggested an enhancement factor of ~1.5 based on a review of laboratory and field studies. The wide range of reported values is not surprising given that rBC mixing state is expected to be influenced by a variety of spatiotemporal factors such as source type, season, and regional atmospheric composition (Krasowsky et al., 2018). In other words, BC aged in different places and at different times may have significantly varying mixing states, resulting in a wide range of absorption and hygroscopicity enhancements in the real world.

Quantifying BC mixing state is challenging because it requires single-particle analysis (Bond et al., 2006). There are two main methods to measure rBC mixing state: (1) microscopy (Johnson et al., 2005; Adachi et al., 2010, 2016), and (2) real-time, in-situ measurements (Hughes et al., 2000). In our study, we quantify rBC mixing state by taking real-time, in-situ measurements with a single particle soot photometer (SP2). The SP2 uses laser-induced incandescence to measure refractory black carbon (rBC) mass per particle, which can be used to directly compute the mass concentration, number concentration, and mass size distribution, and indirectly compute the number size distribution (Stephens et al., 2003). The SP2 can also measure the mixing state of rBC using one of two different methods. In the lag-time method, each sensed rBC-containing particle is deemed as either *thinly-coated* or *thickly-coated* using the measured time difference between the peak of the incandescence and scattering signals induced by the particle (Moteki and Kondo, 2007, 2008). In the leading-edge-only (LEO) method, the actual coating thickness for rBC-containing particles can be explicitly quantified (Gao et al., 2007). Further detail regarding these two methods can be found in section 2.3 and 2.4. In this study, we used both methods to quantify the rBC mixing state.

In this study, we measured rBC with an SP2 on Catalina Island, California (~70 km southwest of Los Angeles) during three different time periods, with the goal of observing how rBC loading and mixing state varied as a function of source type and source-to-receptor timescale. During the first campaign (September 2017), westerly winds dominated and thus the sampling

location was upwind of the dominant regional sources of rBC (i.e., urban emissions from the Los Angeles basin). We suspect measurements during this period to represent well-aged particles; evidence suggests that some of the measured particles originated from wildfires in Oregon and Northern California. In contrast, the second and third campaigns (December 2017,

November 2018) were dominated by northerly-to-easterly "Santa Ana conditions", which advected fresh and aged rBC-containing particles from both biomass burning emissions and urban emissions. Several significant wildfires were active in the Southern and Northern California regions throughout the second and third campaigns. In particular, the Thomas Fire, which was active in Southern California during the second campaign, was the second largest wildfire in modern California history. The Camp Fire, which was active in Northern California during the third campaign, was the 16th largest fire in terms

of burn area size and was also considered the deadliest and most destructive wildfire in modern California history. Table 1 lists the two most significant wildfires for each campaign period that impacted our rBC measurements, along with the total burn area and time period of non-containment for each fire. Mass and number concentrations of rBC-containing particles, rBC size distributions, the number fraction of thickly-coated rBC-containing particles (i.e., using the lag-time method), and absolute coating thickness values (i.e., using the LEO method) are reported. We then evaluate how the rBC loading, size

distribution, and mixing state relate to the meteorology and major sources at the time of measurements in order to further understand the microphysical transformation of BC as it ages in the atmosphere. While a few past studies have investigated the mixing state of rBC in the Los Angeles region using the SP2 (Metcalf et al., 2012; Cappa et al. 2012; Krasowsky et al. 2018), this study is the first to use fixed ground-based measurements off the coast of Los Angeles to focus on how (a) wildfire source-to-receptor travel time, and (b) wildfire versus urban emissions, influence rBC mixing state.

**Table 1.** Major wildfires that were active during the three campaigns. Only the two largest fires from each campaign (in terms of burn area) are listed in the table below. Note that there were numerous other smaller fires that were active during the three campaigns, but not listed in this table.

| Campaign | Wildfire name | Location | Area (km2) | Start date | Containment date |
|---|---|---|---|---|---|
| First (September 2017) | Chetco Bar Fire | Rogue River–Siskiyou National Forest, Oregon | 773 | 12 July, 2017 | 2 November, 2017 |
| | Eclipse Complex | Siskiyou, California | 318 | 15 August, 2017 | 29 November, 2017 |
| Second (December 2017) | Thomas Fire | Ventura and Santa Barbara, California | 1,140 | 4 December, 2017 | 12 January, 2018 |
| | Creek Fire | Los Angeles, California | 63 | 5 December, 2017 | 9 January, 2018 |
| Third (November 2018) | Camp Fire | Butte, California | 620 | 8 November, 2018 | 25 November, 2018 |
| | Woolsey Fire | Ventura and Los Angeles, California | 392 | 8 November, 2018 | 22 November, 2018 |

## 2 Methods

### 2.1 Measurement location and time periods

All measurements reported in this study were conducted at the USC Wrigley Institute for Environmental Studies on Catalina Island (~33°26'41.68"N, 118°28'55.98"W). Catalina Island is located approximately 70 km (43.5 miles) southwest of downtown Los Angeles. Figure 1 shows the location of the sampling site relative to the Los Angeles metropolitan area. The three campaigns were conducted from 7 to 14 September 2017, 20 to 22 December 2017, and 12 to 18 November 2018, Pacific Time (local time).

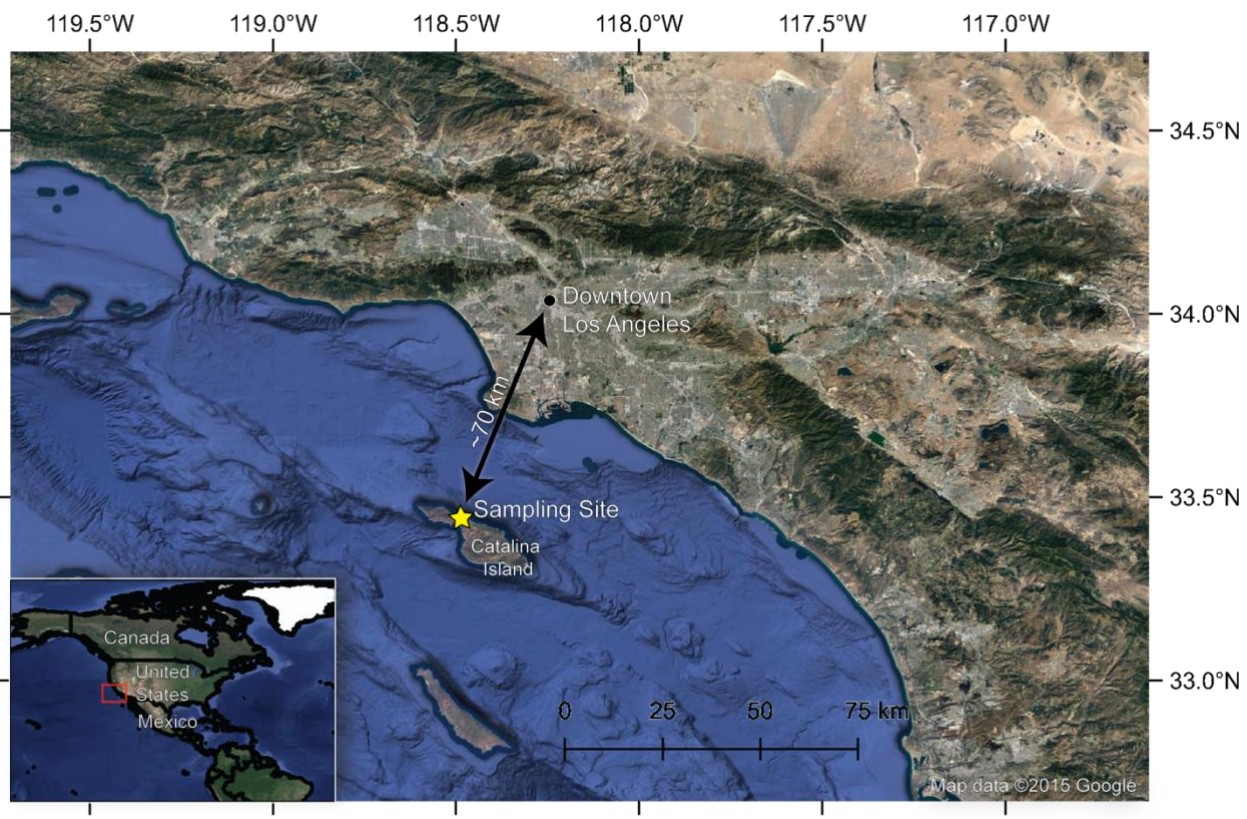

**Figure 1.** Overview map showing the location of the sampling site with respect to the Greater Los Angeles (LA) area. Map data © 2015 Google.

## 2.2 Instrumentation

An SP2 (Droplet Measurement Technologies, Boulder, CO) was used to quantify the physical characteristics of rBC-containing particles for all three campaigns. In short, the SP2 uses laser-induced incandescence to quantify rBC mass on a particle-by-particle basis. The SP2 uses a continuous Nd:YAG laser ($\lambda = 1064$ nm) that is oriented perpendicular to the flow of air containing rBC-containing particles. As each particle passes through the intra-cavity laser, any coating on the rBC particle vaporizes while the core incandesces and emits thermal radiation. The scattered and thermally emitted radiation is measured by optical sensors and converted to signals that can then be used to obtain information about the mass and mixing state of the sampled rBC-containing particles. In this study, an assumed rBC density of 1.8 g cm$_{-3}$ was used. The SP2 has detection limits from ~0.5 to 50 fg rBC per particle. The incandescence signal was calibrated using Aquadag, and the scattering signal was calibrated using polystyrene latex spheres. Further details regarding the governing principles and operation of the SP2 can be found in numerous publications (Stephens et al., 2003; Schwarz et al., 2006; Gao et al., 2007; Moteki and Kondo, 2007; Laborde et al., 2012; Dahlkötter et al., 2014; Krasowsky et al., 2016).

The inlet of the SP2 was positioned on the roof of a three-story research building at the Wrigley Institute as shown in Fig. S1. The height of the inlet was approximately 15 meters above ground level. A fine mesh was secured to the tip of the inlet to prevent clogging by small insects, and a small plastic cone was also attached to block any potential precipitation from entering the inlet. The inlet tube was fed in through a window of a secure laboratory room on the top floor of the building where the SP2 was housed for the duration of sampling. The SP2 ran continuously for the duration of the three measurement periods. Desiccant used to remove moisture from the sample air was replaced on a daily basis, and the data during these replacement periods were subsequently removed during the data analysis.

## 2.3 Auxiliary data

Model simulations and publicly available auxiliary datasets were used to supplement our SP2 measurements.

The National Oceanic and Atmospheric Administration's (NOAA) Hybrid Single-Particle Lagrangian Integrated Trajectory (HYSPLIT) model (Stein et al., 2015) was the primary tool used to identify dominant emissions sources. The HYSPLIT back-trajectories were also used to estimate the age range of measured rBC-containing particles and the path of the air masses carrying these particles. The HYSPLIT trajectory model requires the user to specify the following input parameters: meteorological database, starting point of the back-trajectory, height of source location, run time, and the vertical motion method. A height of 15 meters above ground level was chosen to approximately represent the height of the SP2 inlet positioned on the roof of the laboratory facility. For the first campaign (September 2017), the Global Data Assimilation System (GDAS) meteorology database with 1-degree resolution (~110 km for 1-degree latitude and ~85 km for 1-degree

longitude) was selected, and one-week back-trajectories were simulated for every day of the first campaign. For the second and third campaigns (December 2017, November 2018), the High-Resolution Rapid Refresh (HRRR) meteorology database with a 3-km resolution was selected, and 72-hour back-trajectories were simulated starting on every hour. The GDAS database was selected for the first campaign simulations because a 1-degree resolution was sufficient to show that the

measured air masses were generally coming from the west. In contrast, the HRRR database was used for the second and third campaigns because a finer resolution helped determine the sources that contributed to measured rBC. The default vertical motion method was selected for all back-trajectory simulations.

Data from local weather stations were used to identify the meteorological regimes during all three campaigns, and to

supplement the HYSPLIT back-trajectories used for source characterization. Hourly weather data from Los Angeles International Airport (LAX), Long Beach Airport, Avalon (Catalina Island), Santa Barbara, and Oxnard, during September 2017, December 2017, and November 2018, were obtained using the NOAA National Center for Environmental Information online data tool (https://www.ncdc.noaa.gov/cdo-web/datatools/lcd, last access: 26 August 2019). Five-minute weather data at the same weather stations and time periods were obtained from the Iowa Environmental Mesonet website

(https://mesonet.agron.iastate.edu, last access: 26 August 2019; Todey et al.,2002). Wind data from the USC Wrigley Institute on Catalina Island were also examined when available (7 to 13 September 2017) on the Weather Underground website (https://www.wunderground.com/weather/us/ca/catalina, last access: 26 August 2019), though these data are not validated by NOAA. Data from Santa Barbara, Oxnard, and USC Wrigley Institute were assessed to support conclusions made in this study but are not directly presented in any of the analyses here.


In addition to meteorological data, weather information from local news reports, NASA satellite imagery, and global aerosol model data were used in conjunction to explain the variability in rBC concentrations and mixing state during the sampling campaigns. Local weather news reports between 20 December and 22 December 2017 were used to obtain information about the active fires in Southern California and the dominant wind conditions for each day in the second campaign (December

2017) (CBS Los Angeles, 2017a, 2017b, 2017c, 2017d, 2017e, 2017f). There were generally two local weather reports retrievable per day: one in the early morning and one later on in the evening. The information from these reports was used to get a holistic picture of the local fire and weather conditions at the time of sampling. Data from the California Department of Forestry and Fire Protection (https://www.fire.ca.gov/incidents/, last access: 26 August 2019) was also used to verify basic spatial and temporal information about significant fires occurring during sampling periods. The local weather reports were

used to cross-validate wildfire timelines, but they are not directly presented here.

NASA satellite imagery and data were accessed through NASA's Worldview online application (https://worldview.earthdata.nasa.gov/, last access: 26 August 2019), which provides public access to NASA's Earth Observing System Data and Information System (EOSDIS). Moderate Resolution Imaging Spectroradiometer (MODIS)

images taken from two satellites (Aqua and Terra) were examined for all sampling days. MODIS images were used to identify visible plumes of aerosols, particularly those from large wildfires. The general movement of air masses was also assessed from the visible movement of large-scale clouds from these satellite images. In addition to the MODIS images, aerosol index, aerosol optical depth (AOD), and fires and thermal anomalies data products were examined to supplement the source identification process. For aerosol index, the OMAERUV (Torres, 2006) and OMPS_NPP_NMTO3_L2 (Jaross,

2017) products were used. For AOD, the MYD04_3K MODIS/Aqua and MYD04_3K MODIS/Terra products were used (Levy et al., 2013). For fires and thermal anomalies, the VNP14IMGTDL_NRT (Schroeder et al., 2014) and MCD14DL (Justice et al., 2002) products were used. Examples of NASA data products used for source identification analysis can be found in the Supplement.

An open-source online visualization tool (earth.nullschool.net, last access: 26 August 2019) was used to visually assess the European Centre for Medium-Range Weather Forecasts (ECMWF) Copernicus Atmosphere Monitoring Service (CAMS) model output data (Beccario, 2019; https://atmosphere.copernicus.eu/, last access: 26 August 2019). The CAMS model provides "near-real-time" forecasts of global atmospheric composition on a daily basis. Specifically, the $PM_{2.5}$ concentration output data from CAMS were examined using earth.nullschool.net. The CAMS output visualizations were particularly

helpful for understanding where certain sources were located and when they were likely affecting our measurements. The concentration gradients of $PM_{2.5}$ were examined on the visualization tool on an hourly interval for every day of active sampling in order to supplement the HYSPLIT analysis and confirm the contribution of certain emission sources. Access to the CAMS visualizations for the three campaigns can be found in the Supplement and Video Supplement.

## 2.4 Estimation of source-to-receptor timescale

Characteristic timescales of transport between the sampling site and nearest source(s) were estimated based on the HYSPLIT trajectories simulated for source identification. The approximate source-to-receptor timescale characterizations by HYSPLIT were cross-validated with approximate calculations of transport time performed with representative length scales between sources and the sampling site, and the average wind speeds during the time periods of interest. Further details regarding the calculations of the timescale characterizations are in section S1 of the Supplement. Although we cannot fully capture the

intricacies of particle aging timescales with our estimates, they are meant to be conservative approximations based on available meteorological data. These estimated source-to-receptor timescales were used to help categorize different LEO periods by source(s) (see Table 2 and Fig. 10), and also used in our discussion of how rBC mixing state evolves with particle aging (see Section 3.7).

## 2.5 Time series filtering

rBC mass and number concentrations during the first campaign (September 2017) showed anomalous spikes likely due to unexpected local sources. In an effort to obtain representative background concentrations, we filtered these spikes by removing values above a threshold of 0.08 µg m$^{-3}$ and 40 cm$^{-3}$ for mass and number concentrations, respectively. Figure S2 in the Supplement shows the time series for the first campaign before and after removal of spikes. Figure S3 in the Supplement shows the median rBC concentration for the first campaign as a function of the cut-off threshold value. Median

rBC mass and number concentrations appeared to asymptote at cut-off values of approximately 0.08 µg m$^{-3}$ and 40 cm$^{-3}$, suggesting that the median rBC concentration values become insensitive to the choice of cut-off threshold above these values.

## 2.6 Lag-time method

The mixing state of rBC was examined using two different methods. The first method used to characterize mixing state is

called the lag-time method. This method categorizes each rBC particle as either "thickly-coated" or "thinly-coated" based on a measured time delay (i.e., "lag-time") between the scattering and incandescence signal peaks. This method has been previously described and used in various studies (Moteki and Kondo, 2007; McMeeking et al., 2011a; Metcalf et al., 2012; Sedlacek et al., 2012; Wang et al., 2014; Krasowsky et al., 2016; Krasowsky et al., 2018). In short, as a coated rBC-containing particle passes through the SP2 laser, the sensors will detect a scattering signal as the coating vaporizes. Shortly

after, there will be a peak in the incandescence signal as the rBC core heats up and emits thermal radiation. A probability density function of the lag-time values often results in a bimodal distribution (Moteki and Kondo, 2007; McMeeking et al., 2011b). Based on the data for a particular campaign, a lag-time cut-off is chosen between the two peaks of the bimodal distribution to bin each rBC particle as either thinly or thickly-coated. The fraction of rBC particles that are thickly-coated ($f_{BC}$) is then determined based on this categorization. For our study, a lag-time cut-off of 1.8 µs was chosen to quantify

whether an rBC-containing particle was thickly-coated. Only particles with an rBC core diameter greater than 170 nm were included in the calculation of $f_{BC}$ to account for the scattering detection limit of the instrument. As discussed previously by Krasowsky et al. (2018), the lag-time method is inherently susceptible to biases since $f_{BC}$ can depend on the selection of the lag-time cut-off value. For example, Krasowsky et al. selected a cut-off value of 1 µs for a near-highway SP2 campaign in the Los Angeles Basin, which is significantly different than the value of 1.8 µs used in this study and others. There remains

an unresolved issue of maintaining consistency between different studies utilizing the lag-time method, while simultaneously representing the unique mixing state characterization of each measured rBC population; the definition of "thickly-coated" likely varies by the aerosol population sampled and thus is not necessarily comparable from one study to the next.

## 2.7 Leading-edge-only (LEO) method

BC mixing state was also characterized using the LEO method. In brief, this method reconstructs a Gaussian scattering
function from the leading edge of the scattering signal for each rBC-containing particle. The width and location of the
reconstructed Gaussian scattering function is determined by a two-element avalanche photodiode. Assuming a core-shell
morphology, the rBC coating thickness is subsequently calculated using Mie theory from the reconstructed scattering signal
and the incandescence signal (Gao et al., 2007; Moteki and Kondo, 2008). Refractive indices of (2.26+1.26i) and (1.5+0i)
were selected for rBC cores and rBC coating material, respectively. These parameters were selected based on
recommendations and results from previous studies (Moteki et al., 2010; Dahlkötter et al, 2014; Taylor et al., 2014; Taylor et
al., 2015). The Paul Scherrer Institute's single-particle soot photometer toolkit version 4.100b (developed by Martin Gysel et
al.) was used to perform the LEO method in Igor Pro version 7.09.

In this study, the LEO "fast-fit" method was used with three points, and particles analyzed were restricted to those with rBC
core diameters between 180 and 300 nm. Although the SP2 has been reported to accurately measure the volume equivalent
diameter (VED) of scattering particles down to ~170 nm, a more conservative lower threshold of 180 nm was used for our
study to reduce instrument noise at smaller VED values near the detection limit (Krasowsky et al., 2018). Specific rBC core
diameter ranges were used for different analyses in this study and these ranges are explicitly defined within each respective
discussion. One exception was made to the 180–300 nm rBC core diameter restriction in section 3.7. For the analyses and
discussion presented in section 3.7, the LEO coating thickness was calculated for all detectable rBC particles with non-
saturated scattering signals. The rBC core size was not restricted in this section because the relative comparisons between
characteristic coating thickness values were more important for the analysis, rather than the absolute value (which would
likely be biased, as discussed further in section 3.8). In other words, the LEO-derived coating thickness values in section 3.7
were not used to report representative averages for selective time periods, but rather were used for comparative and/or
qualitative purposes.

Negative LEO-derived coating thickness values are reported throughout the results and discussion section. These non-
physical results are caused by instrument noise from both the incandescence and scattering detectors. The per-particle
uncertainty associated with both of these signals results in a spread of coating thickness values that are at times less than
zero. As a hypothetical example, a thinly-coated particle that has its scattering cross section underestimated and its rBC mass
equivalent diameter overestimated may result in a negative coating thickness value. According to Metcalf et al. (2012), per-
particle coating thickness uncertainty is ~40%, with the uncertainty reduced for larger rBC particles. In contrast to per-
particle uncertainty, systemic uncertainty, which is that associated with the average of the population of particles, is largely
caused by the choice of assumed parameters for the rBC core, namely its refractive index and density (Taylor et al., 2015).
These systemic errors complicate direct comparison between measurements conducted with different sets of parameters

assumed for the rBC core, but they do not affect comparisons within the same set of measurements. Negative LEO-derived coating thickness values have been reported in numerous past studies, and further details regarding both per-particle and systemic uncertainties can be found in these studies (Metcalf et al., 2012; Laborde et al., 2013; Krasowsky et al., 2018; Taylor et al., 2015).

## 3 Results and discussion

This section starts by discussing the major identifiable sources and meteorological patterns in each of the three campaigns (section 3.1). Then, the overall mass and number loading of rBC is discussed and compared to past literature values (section 3.2). Following that, the rBC mixing state results from the lag-time and LEO analyses are discussed (sections 3.3–3.5). The impacts of emissions source type and atmospheric aging on rBC mixing state and core size are subsequently discussed (sections 3.6, 3.7). Section 3 then ends by comparing rBC coating thickness values calculated in this study to reported values from past studies

### 3.1 Source identification and meteorology

In this section, we summarize the dominant pollutant sources and wind patterns for each of the three campaigns. For all three campaigns, we used HYSPLIT back-trajectories, HYSPLIT dispersion model, CAMS model data, and NASA data products (i.e., satellite imagery, aerosol index products, and AOD products) in conjunction to identify the most likely sources of measured rBC-containing particles. For the first campaign (September 2017), the Oregon wildfires were identified as probable sources of measured rBC. Furthermore, we also identified long-range transport from East Asia and ship/aviation emissions as potential sources contributing to measured rBC. Overall, we expect measured rBC during the first campaign to be aged.  For the second campaign (December 2017), fresh urban emissions from the Los Angeles basin and biomass burning emissions from the Thomas Fire in Santa Barbara and Ventura County (along with other smaller Southern California fires) were the main sources identified by our analysis. For the third campaign (November 2018), fresh urban emissions from the Los Angeles basin and fresh biomass burning emission from the Woolsey Fire in Ventura (along with other smaller Southern California fires) were the main sources identified for approximately the first four days of the campaign. For the last two days of the third campaign, the Camp Fire in Northern California (along with other smaller fires in Northern and Central California) contributed significantly to measured rBC. Figure 2 displays wind roses for each campaign at three different weather station locations (public data provided by NOAA, see section 2.3). Furthermore, Fig. 3 shows HYSPLIT back-trajectories simulated for each of the three campaigns and further highlights the differences in wind conditions between the three campaigns. These figures show the distinct meteorological regimes of each campaign. A more detailed description of the source identification process can be found in section S2 of the Supplement.

For the remainder of the manuscript, we refer to rBC measured when the dominant source was biomass burning emissions as $BC_{bb}$, and rBC measured when the dominant source was fossil fuel (e.g., urban) emissions will be referred to as $BC_{ff}$. rBC measured in the first campaign (September 2017), when the measured air masses were representative of well-aged background over the Pacific Ocean, will be referred to as $BC_{aged,bg}$.

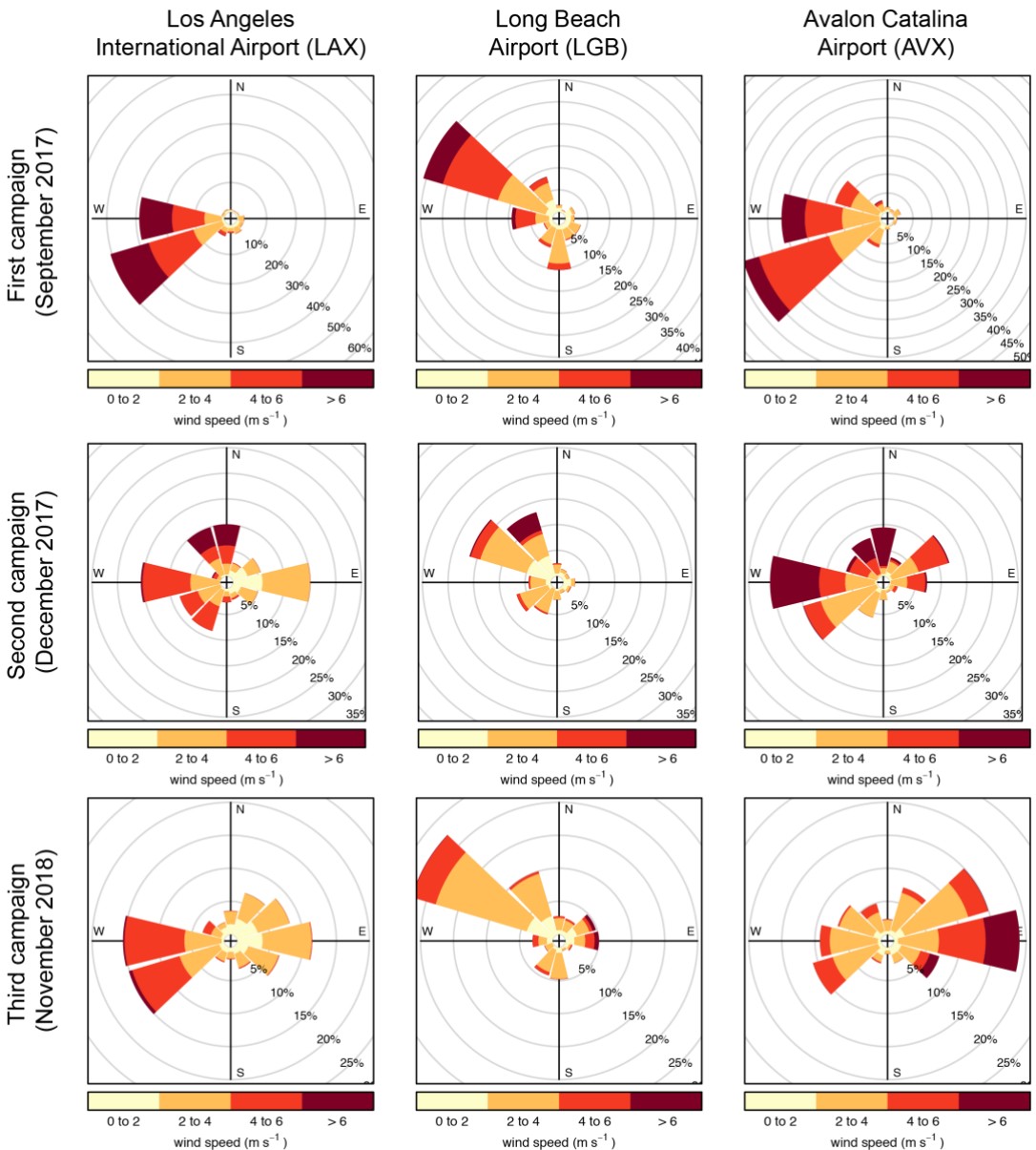


**Figure 2.** Wind roses for the September 2017 (first row), December 2017 (second row), and November 2018 (third row) sampling periods. Wind roses are based on five-minute ASOS airport data from LAX (first column), LGB (second column), and AVX (third column), provided by NOAA.

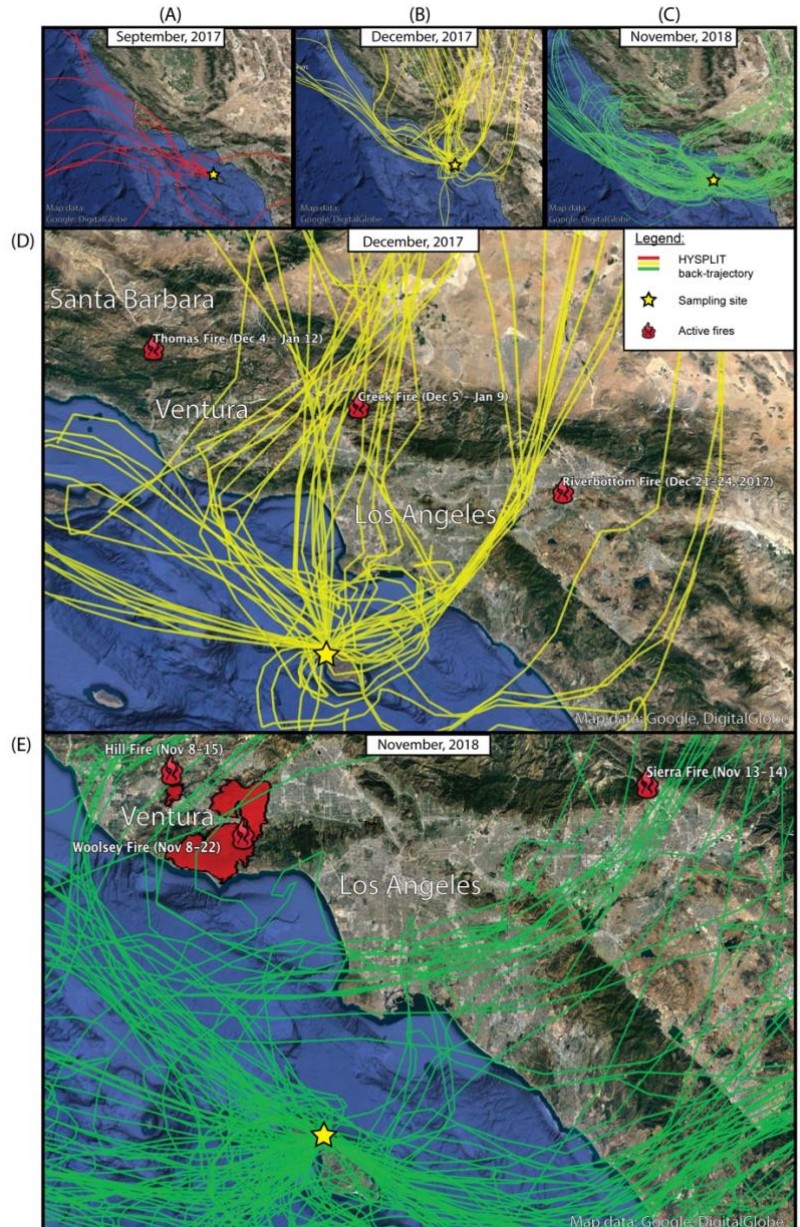

**Figure 3.** HYSPLIT back-trajectories for all three campaigns. The star denotes the start location of each back-trajectory, i.e., the sampling location. The trajectories for the first period (September 2017) (i.e., panel a) represent week-long back-trajectories for each day of the campaign. The trajectories for the (b) second (December 2017) and (c) third (November 2018) periods represent 72-hour back-trajectories for each hour of the campaign. Panels (d) and (e) show more zoomed-in maps of the second and third campaign back-trajectories along with active Southern California fires. Map data: Google DigitalGlobe.

### 3.2 rBC mass and number concentration

Figure 4 shows time series for rBC mass and number concentrations, rBC coating thickness ($CT_{BC}$), number fraction of thickly-coated particles ($f_{BC}$), and rBC count mean diameter (CMD) for all three measurement campaigns. The mixing state ($CT_{BC}$ and $f_{BC}$) and rBC size are discussed in following sections. The mean mass and number concentration (±standard

deviation) for the first campaign (September 2017) was 0.04 (±0.01) µg m-3 and 20 (±7) cm-3, respectively. For the second campaign (December 2017), the corresponding mean concentrations were 0.1 (±0.1) µg m-3 and 63 (±74) cm-3, with concentrations reaching as high as 0.6 µg m-3 and 381 cm-3. Likewise, for the third campaign (November 2018), the corresponding mean concentrations were 0.15 (±0.1) µg m-3 and 80.2 (±54.5) cm-3. The range of observed rBC concentrations is larger for the second and third campaigns compared to the first campaign, and there are distinct prolonged

peaks in concentrations that can be observed during these times. In comparison, the first campaign shows relatively stable concentrations.

Given the remote location of the sampling site and the consistent westerly winds during the first campaign (September 2017), the observed rBC concentrations establish an appropriate baseline for ambient conditions away from the broader

urban plume in the Los Angeles basin. On the other hand, the concentrations during the second and third campaigns (December 2017, November 2018) were more variable, with mean concentrations that were higher than the first campaign due to periods of northerly-to-easterly winds driven by Santa Ana wind conditions as described in section 3.1. Figure 5 shows rBC mass and number concentrations along with wind speed and direction during the second campaign. Wind direction was directly related to elevated concentrations for all three peaks shown. Peak P1 is clearly preceded by a

prolonged period of northerly winds. Similarly, Peaks P2 and P3 are preceded by periods of easterly winds. An analogous plot for the third campaign is shown in Fig. S8, but the relationship between wind direction measured at LAX and the rBC concentration is not clearly discernible since long distance biomass emissions were impacting the measurements in addition to local sources near the LA basin. The impacts of different sources on measurements during the third campaign are described in detail in section S2 of the Supplement.


The mean concentration for the first campaign (September 2017) was approximately an order of magnitude lower than the mean concentration of ~0.14 µg m-3 observed by Krasowsky et al. (2018) near the downwind edge (assuming dominant westerly wind flows) of the LA Basin (i.e., Redlands, CA). Concentrations during the most polluted time periods in our measurements were comparable to recently measured concentrations in the Los Angeles basin (Krasowsky et al., 2018) but

at least one to two orders of magnitude lower than average concentrations found in other heavily polluted cities around the world. Mass concentration values of ~0.9 µg m-3, ~0.5 to 2.5 µg m-3, ~0.9 to 1.74 µg m-3, and ~0.6 µg m-3 were measured with an SP2 in Paris, Mexico City, London, and Houston, respectively (Laborde et al., 2013; Baumgardner et al., 2007; Liu

et al., 2014; Schwarz et al., 2008a). In urban areas of China, an average mass concentration of ~9.9 µg m-3 was reported for a polluted period in Xi'an (Wang et al., 2014).

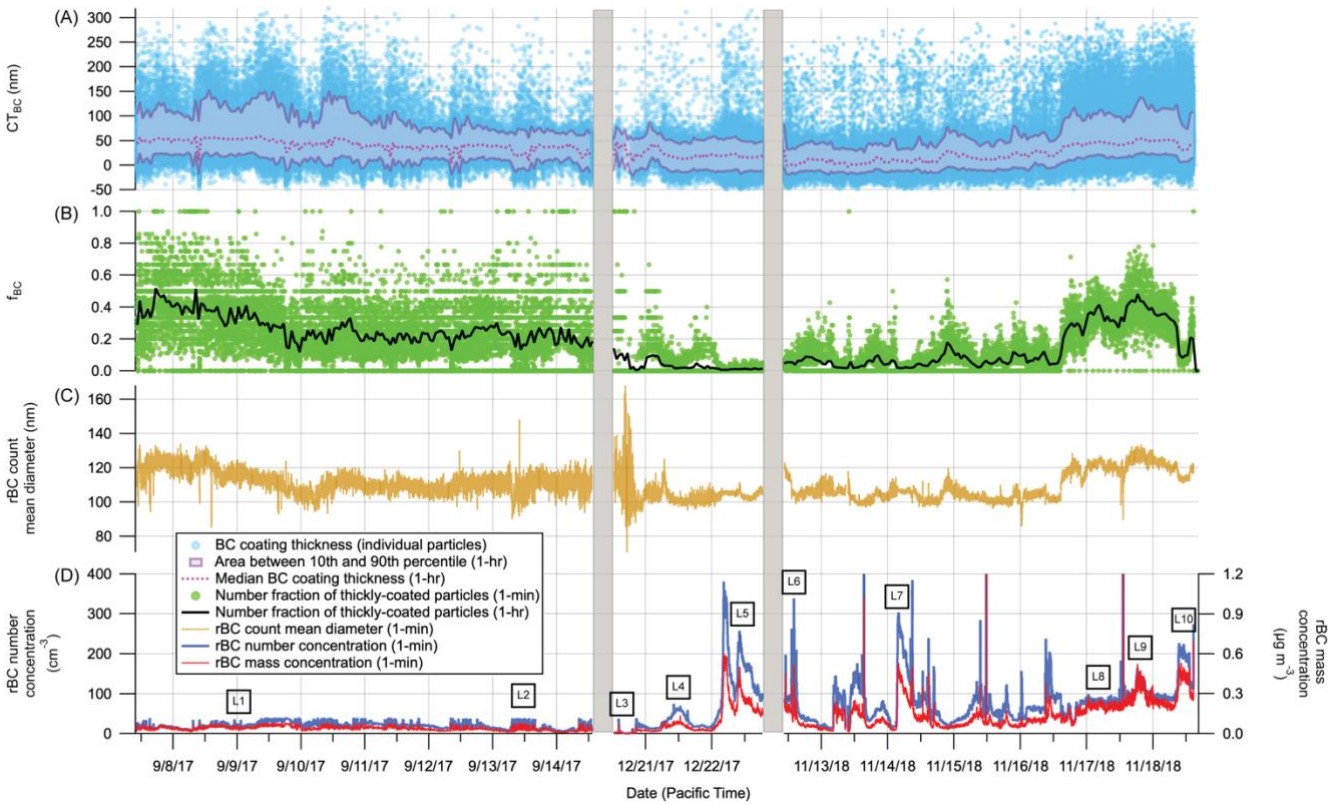


**Figure 4.** Time series of (a) BC absolute coating thickness, (b) number fraction of thickly-coated rBC particles, (c) rBC count median diameter, and (d) rBC concentrations, for all three measurements campaigns. The boxed annotations (i.e., L1 to L10) refer to specific LEO periods, which are further described in Section 3.5. In panel (a), each blue dot represents an individual particle. The hourly median is shown in the dotted pink line, and the corresponding 10th and 90th percentiles are shown in purple. In panel (b), green dots represent one-minute means while the black curve shows hourly means. Panel (c) shows the one-minute mean for the count mean diameter. Panel (d) shows the one-minute means for rBC concentration.


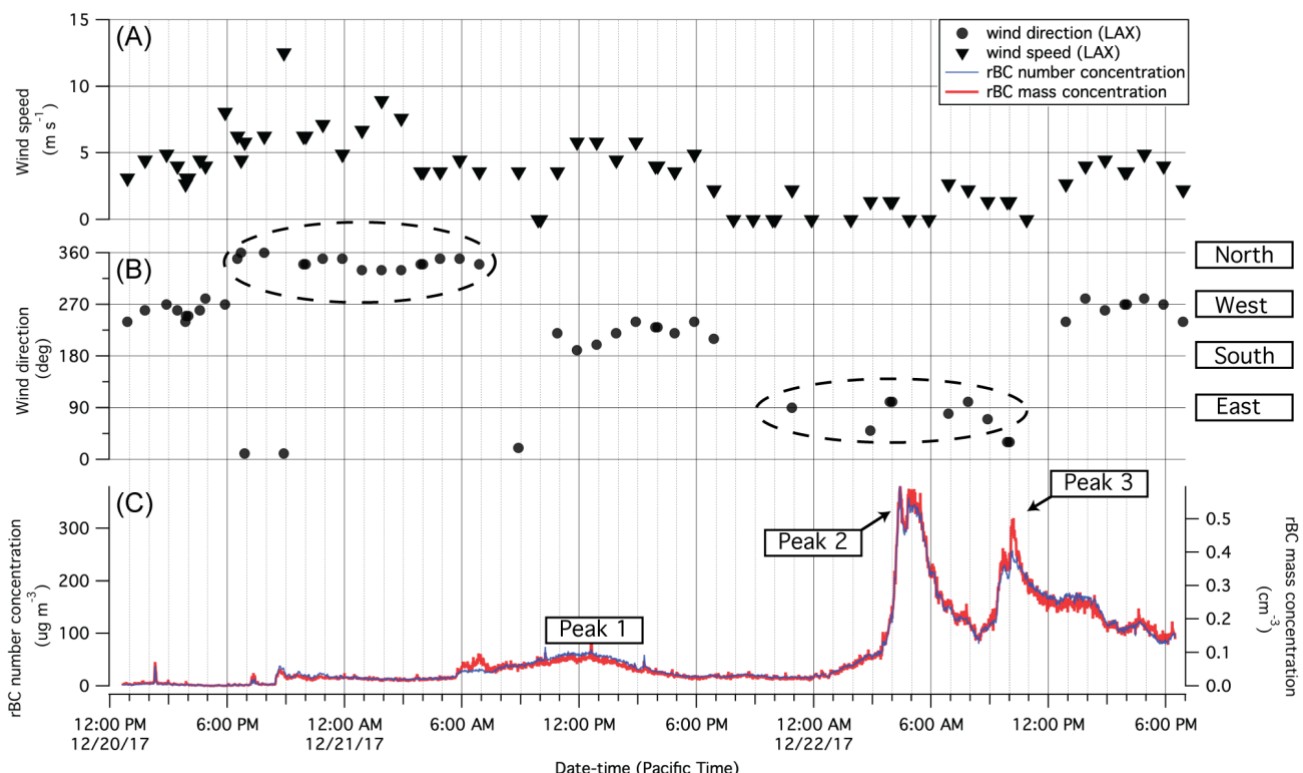

**Figure 5.** Meteorological variables and rBC concentrations during the second campaign (December 2017). Panel (a) shows
wind speed and (b) shows wind direction measured by a NOAA weather station located at Los Angeles International Airport
(LAX). Panel (c) shows rBC mass and number concentrations and identifies three peaks of interest. The two dashed ovals in
panel (b) highlight periods of northerly and easterly winds, which occur ~0.5-1 days before each of the three peaks,
suggesting that the elevated rBC concentrations included important contributions from the local Thomas Fire (and other
smaller fires) and urban emissions from the Los Angeles basin.

### 3.3 Lag-time analysis: number fraction of thickly-coated rBC-containing particles

Figure 4, panel (b) shows both one-minute and one-hour means for $f_{BC}$ over the course of all three campaigns. On average,
$f_{BC}$ was larger during the first campaign (September 2017) than during the second and third campaigns (December 2017,
November 2018). The mean values (±standard deviation) of $f_{BC}$ were 0.27 (±0.19), 0.03 (±0.09), and 0.14 (±0.15) for the
first, second, and third campaigns, respectively. This implies that about one-quarter of the rBC-containing particles that were
measured in the first campaign either had sufficient time in the atmosphere to become aged with thick coatings or originated
from biomass burning emission sources, which have been shown to emit more thickly-coated particles compared to fossil
fuel emissions (Dahlkötter et al., 2014; Laborde et al., 2013; Schwarz et al., 2008a). Most of the rBC particles measured in
the second campaign were thinly-coated, implying BCff dominated measurements. The rBC from the third campaign
exhibited mostly thinly-coated rBC for the first ~four days of the campaign and an increased $f_{BC}$ for the last ~two days of the
campaign.

Compared to past studies in the Los Angeles region, the mean $f_{BC}$ for the first campaign (September 2017) ($f_{BC} = 0.27$) is close to the lower end of values from aircraft measurements ($f_{BC} = 0.29$) (Metcalf et al., 2012) and the upper end of previous ground-based measurements ($f_{BC} = 0.21$) (Krasowsky et al., 2016). In contrast, the mean value of $f_{BC}$ for the second campaign

(December 2017) is almost an order of magnitude lower than for the first campaign. There are some periods with slightly elevated $f_{BC}$ during the second campaign, but the overall trend suggests that most of the rBC-containing particles in this period are thinly-coated or essentially uncoated. The Santa Ana wind conditions during the second campaign advected fresh (a) urban emissions from the Los Angeles basin, and (b) biomass burning emissions from active fires in Southern California, as discussed in section 3.1.


The third campaign (November 2018) is unique in that both "fresh" and "aged" $BC_{bb}$, in addition to fresh $BC_{ff}$ were measured. As shown in Fig. 4, there is a distinct period of relatively higher $f_{BC}$ and rBC concentrations starting at ~noon on 16 November 2018 and lasting through the end of the campaign on 18 November 2018. This is the only period from all three measurement campaigns where we observed both high rBC mass/number loadings and high $f_{BC}$ values. In section 3.1, we

identified the Camp Fire to be the dominant source during this time period within the third campaign. Thus, the biomass burning rBC particles measured in this portion of the third campaign are more thickly-coated than our measured urban rBC. Previous field studies have reported that $BC_{ff}$ generally have a lower $f_{BC}$ relative to $BC_{bb}$ (Schwarz et al., 2008a; Sahu et al., 2012; Laborde et al., 2013; McMeeking et al., 2011b; Akagi et al., 2012). For example, Schwarz et al. (2008a) reported that $f_{BC} \sim 10\%$ for urban emissions and $f_{BC} \sim 70\%$ for biomass burning emissions. The impact of source type on rBC mixing state

will be further discussed in section 3.7.

**3.4 Negative lag-times and rBC morphology**

A number of previous studies (Moteki and Kondo., 2007; Sedlacek et al., 2012; Moteki et al. 2014; Dahlkötter et al. 2014; Sedlacek et al., 2015) reported negative lag-times from both laboratory and field measurements of rBC. It has been hypothesized that a negative lag-time is observed when rBC fragments from its coating material, resulting in a scattering

signal that follows an incandescent signal. Dahlkötter et al. (2014) summarized that negative lag times can occur when either: (i) rBC is very thickly-coated in a core-shell configuration, (ii) rBC is thickly-coated and the core is offset from the center in an eccentric arrangement, or (iii) rBC is located on or near the surface of an rBC-free particle. The morphology of rBC-containing particles is of importance because the enhancement of BC light absorption can vary widely depending on whether the morphology more closely resembles a core-shell configuration or near-surface attachment (Moteki et al., 2014).

Although the fraction of negative lag-times ($f_{lag,neg}$) cannot definitively identify the morphology of individual rBC-containing particles (Sedlacek et al., 2015) or accurately quantify the actual percentage of all fragmenting rBC-containing particles (Dahlkötter et al. 2014), it can offer some general insights about rBC morphology, especially when it is paired with other

information like the emissions source type and rBC coating thickness. $f_{lag,neg}$ is a conservative lower-bound estimate for the fragmentation rate since there may be rBC particles with positive lag-times that still fragment in the SP2 (Dahlkötter et al.,

2014). Dahlkötter et al. (2014) used a method examining the tail end of the time-dependent scattering cross-section in order to determine if a rBC-containing particle was fragmenting, thereby calculating a higher fragmentation rate relative to $f_{lag,neg}$. Details of the time-dependent scattering cross-section method can be found in Laborde et al. (2012) and Dahlkötter et al. (2014). This method to calculate a refined fragmentation rate was not used in Sedlacek et al. (2012), nor in this study.

Furthermore, Sedlacek et al. (2012, 2015) suggest that $f_{lag,neg}$ and the lag-time distributions may assist in source attribution. More specifically, Sedlacek et al. (2012) measured a confirmed biomass burning plume in August 2011 and found high positive correlation between biomass burning tracers and $f_{lag,neg}$ during the period of impact, suggesting that $f_{lag,neg}$ may be a useful indicator of biomass burning influence.

In this study, we observed negative lag-times, although at a relatively low rate, with $f_{lag,neg}$ calculated to be much less than 0.1 throughout most of the measurement periods (see Fig. 6). We defined $f_{lag,neg}$ to be identical to the "fraction of near surface rBC particles" metric used by Sedlacek et al. (2012), using a lag-time threshold of -1.25 μs to account for uncertainties associated with the lag-time determination. The campaign-wide $f_{lag,neg}$ was 0.017 for the first campaign (September 2017), 0.018 for the second campaign (December 2017), and 0.026 for the third campaign (November 2018). Comparatively,

Dahlkötter et al. (2014) observed $f_{lag,neg}$ of ~0.046 during an airborne field campaign measuring an aged biomass burning plume, and additionally calculated a higher fragmentation rate of ~0.4 to 0.5, based on their aforementioned alternative method (Laborde et al., 2012). Sedlacek et al. (2012) reported $f_{lag,neg} > 0.6$ for ground-based measurements of a biomass burning plume in Long Island, New York, originating from Lake Winnipeg, Canada.

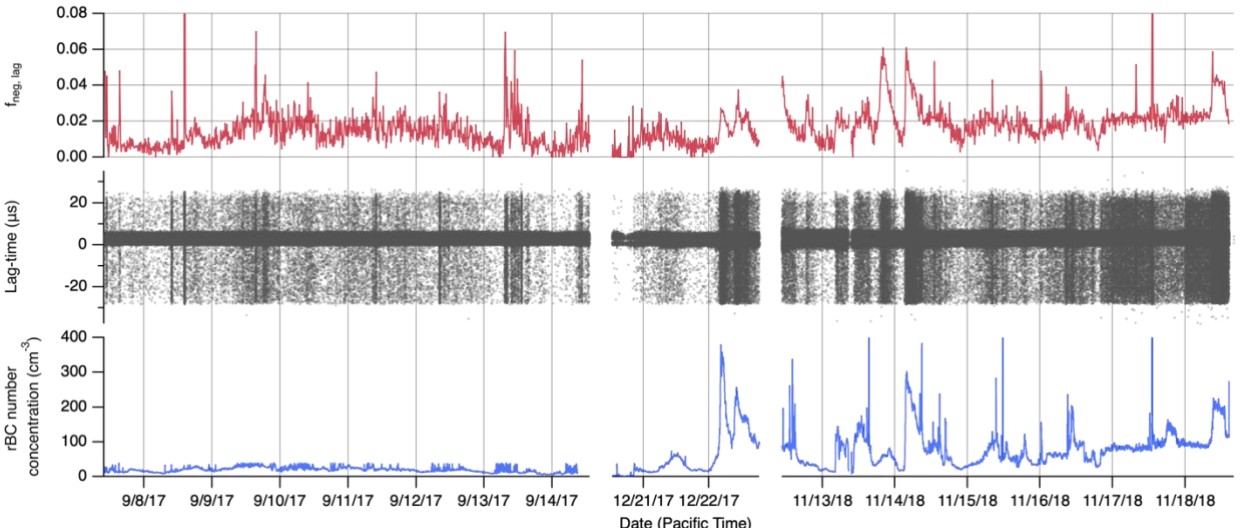

**Figure 6.** Panel (a) shows the 10-minute mean time series for number fraction of rBC particles with negative lag-times ($f_{neg,}$ $_{lag}$). The threshold for negative lag-times was set to -1.25 µs to account for uncertainties in the lag-time determination (Sedlacek et al., 2012). Panel (b) shows the time series of lag-time values for each individual particle, corresponding to individual dots on the graph. Panel (c) shows the one-minute mean rBC number concentration for reference.

The widely varying $f_{lag,neg}$ between these different studies (including our study) suggests that $f_{lag,neg}$ may not be a useful metric when comparing between studies. One of the key findings from Sedlacek et al. (2015) shows that SP2 operating conditions strongly affects the frequency of negative lag-times, and suggests that inter-study comparisons of $f_{lag,neg}$ could be meaningless, or at worst misleading, if the laser power and sample flow rate are not reported. See the section S3 in the Supplement for more details.


The higher mean value of $f_{lag,neg}$ (0.026) during the third campaign (November 2018), relative to the first (0.017) and second (0.018) campaigns, shows that $f_{lag,neg}$ could potentially be a useful as a supplemental metric when identifying impacts from biomass burning sources, as mentioned by Sedlacek et al. (2012, 2015). Figure 7 also shows that the 10-minute mean negative lag-times increase in magnitude with increasing rBC core diameter between the range of ~100 to 115 nm (i.e.,

higher rates of fragmentation with increasing core size). This follows a similar trend observed by Sedlacek et al. (2012, 2015), who attributed this trend to increased heat dissipation to surrounding gases for smaller rBC cores, which in turn decreases the particle heating rate and consequently decreases the fragmentation rate. Our observations add to the limited past observations that show that the fragmentation rate of rBC particles in the SP2 depend on physical factors like the core size. This further complicates the practical use of $f_{lag,neg}$ as a biomass burning indicator.


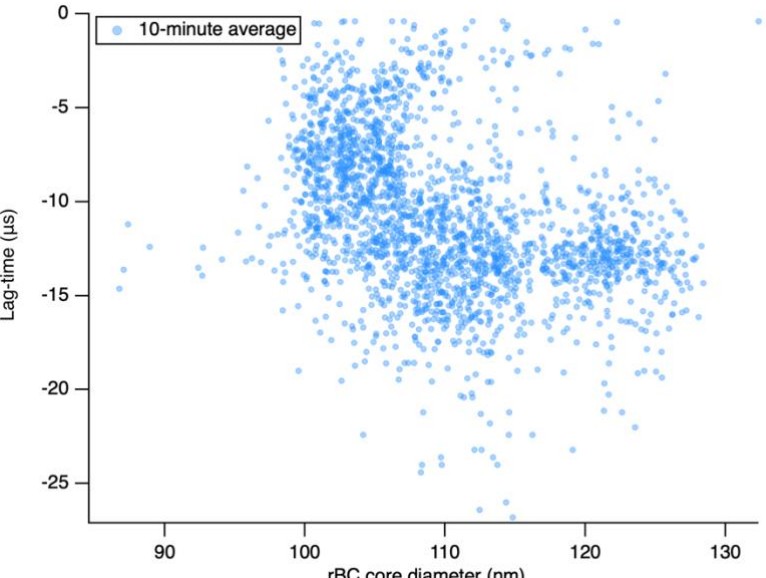

**Figure 7.** Scatter plot of 10-minute mean negative lag-times versus 10-minute mean rBC core diameters.

Certain trends in $f_{lag,neg}$ for this study further indicate that it should not be used in isolation to verify the relative abundance of biomass burning aerosol versus non-biomass burning aerosols. There are peaks in the $f_{lag,neg}$ time series (Fig. 6) that do not follow the expected trends based on identifiable source impact time periods. For example, the two peaks on 22 December 2017 ($BC_{ff}$ periods) correspond to $f_{lag,neg}$ values exceeding 0.02, but $f_{lag,neg}$ hovers around 0.02 on 17 November 2018, when we had expected direct impact from the Camp Fire. As evidenced from the meteorology (Section 3.1), mixing state (Section 3.3), rBC concentrations (Section 3.2), and rBC core size (to be discussed in Section 3.6), measurements on 17 November 2018 were dominated by biomass burning emissions, but $f_{lag,neg}$ fails to show that independently. These anomalous observations show that $f_{lag,neg}$ needs to be used with caution, and that future studies are necessary to extensively quantify the relationship between $f_{lag,neg}$ and source type.

The observations of negative lag-times in this study confirm that ambient rBC likely do not adhere strictly to core-shell morphology. The exact morphology of measured rBC cannot be quantified based on our measurements, but the presence of negative lag-times in this study highlights the need to further understand rBC morphology and its effect on absorption enhancement in future studies, as well as the potential for $f_{lag,neg}$ to be used as a supplemental source identification tool.

### 3.5 Leading-edge-only (LEO) fit analysis: rBC coating thickness

To further examine the mixing state of rBC-containing particles, the leading-edge-only (LEO) fit method was used to quantify rBC coating thickness ($CT_{BC}$) on a particle-by-particle basis. Figure 4 shows the time series of $CT_{BC}$ throughout all three campaigns. The time series of $CT_{BC}$ shows that each campaign was characterized by different mixing states, and that there are distinct trends within each campaign.

The inter-campaign differences are highlighted in Fig. 8, which shows the $CT_{BC}$ distribution for each campaign, as well as the distribution including rBC from all campaigns. For both rBC core diameter ranges (180–220 nm and 240–280 nm), we observe that the first campaign has the largest mean $CT_{BC}$, followed by the third and second campaign, respectively. The mean $CT_{BC}$ ($\pm$ standard deviation) for the first, second, and third campaign was 52.5 ($\pm$ 45.5) nm, 22.3 ($\pm$ 25.0) nm, and 40.3 ($\pm$ 41.5) nm, respectively, for particles with a rBC core diameter between 180 and 220 nm.

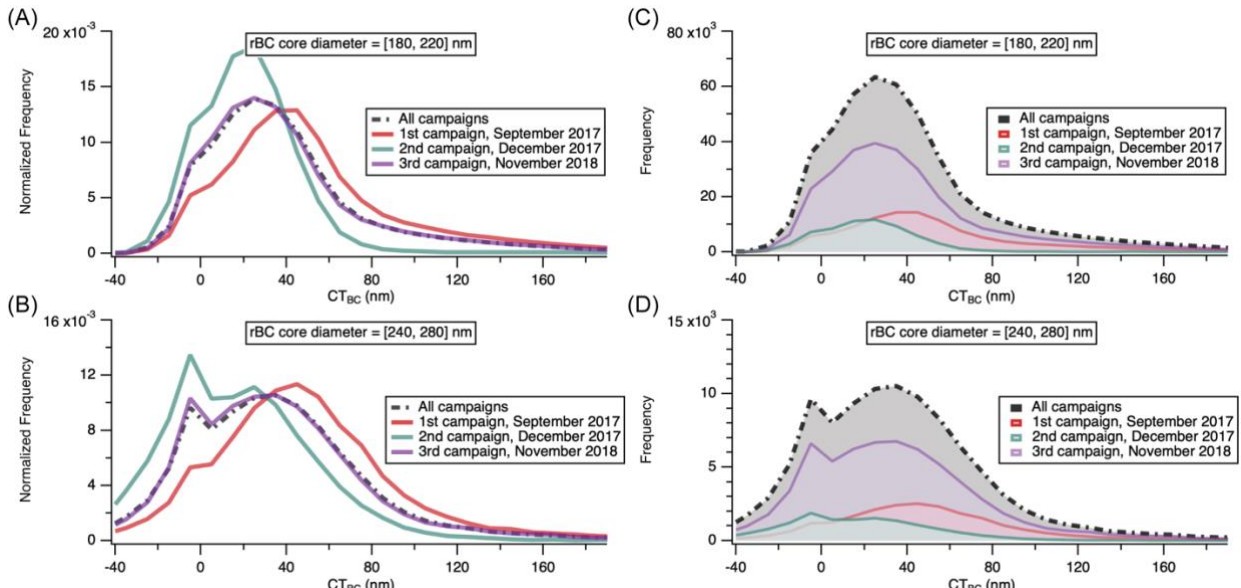

**Figure 8.** Distributions of BC coating thickness ($CT_{BC}$) aggregated by campaign are shown in red (1st campaign), green (2nd campaign), and purple (3rd campaign). The combined distributions for all campaigns are shown in black. Panels (a) and (b) show the normalized frequency distributions, while panels (c) and (d) show the absolute frequency distributions. The distributions are also distinguished by the rBC core diameter ranges included in the LEO analysis. The top panels (a) and (c) show distributions for particles with rBC core diameters between 180 and 220 nm. The bottom panels (b) and (d) show distributions for particles with rBC core diameters between 240 and 280 nm.

Comparing the time series of $CT_{BC}$ to the time series of $f_{BC}$ (Fig. 4), we observe similar trends over time, which is expected and also reported in past studies that have employed both the lag-time and LEO methods (Metcalf et al., 2012; Laborde et al., 2012; McMeeking et al., 2011a). Figure 9 shows a positive correlation between 10-minute mean $CT_{BC}$ and $f_{BC}$ (r = 0.82, r2 = 0.67) throughout all three campaigns. This strong correlation confirms that these two methods are in general agreement, and that they can be used together to robustly describe the rBC mixing state.

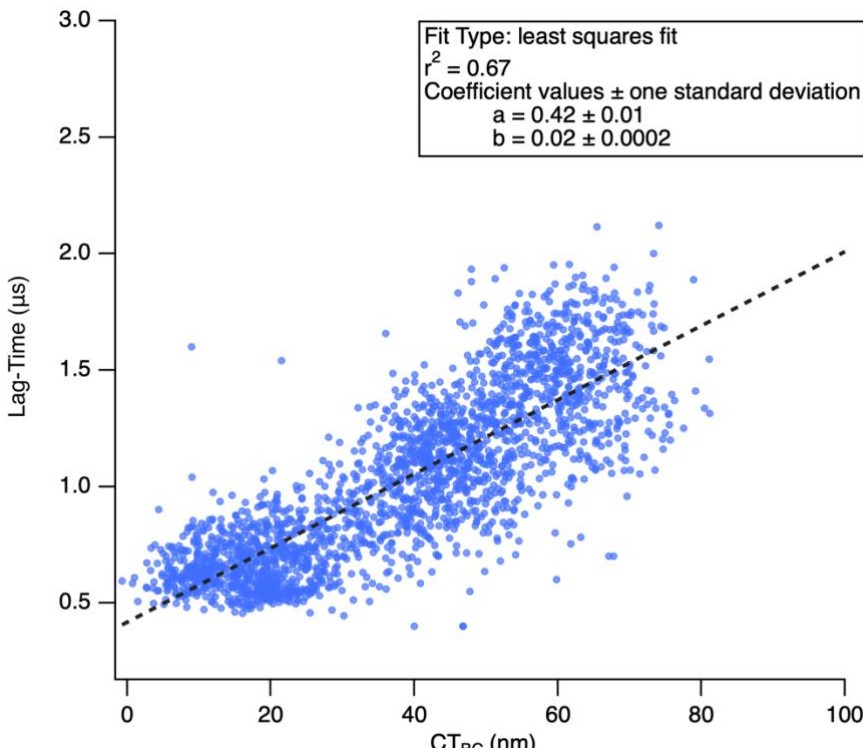

**Figure 9.** Scatter plot as a function of lag-time (i.e., delay time) and BC coating thickness ($CT_{BC}$). Each point on the plot represents a 10-minute mean value. Data shown include average values from all three campaigns. A significant correlation is confirmed using a linear correlation test.

In addition to aggregating $CT_{BC}$ by campaign, we also examined ten discrete time periods of interest to obtain a more detailed understanding of the mixing state variability. Particles with rBC diameters between 200 and 250 nm were used in the comparison of these ten time periods. Two time periods from the September campaign, three time periods from the December campaign, and five time periods from the November campaign were selected to represent a diverse range of meteorological conditions, emission sources, and age of aerosols. Table 2 lists the ten LEO-fit periods, and their median and mean $CT_{BC}$. The LEO-fit periods are also annotated on the rBC concentration time series (see Fig. 4) to show when they occurred in the context of all three campaigns. The median $CT_{BC}$ for the LEO periods ranged from –0.4 to 54.0 nm. L6 had the lowest median $CT_{BC}$ (–0.4 nm), while L9 had the highest median $CT_{BC}$ (54.0 nm).

**Table 2.** Details of the ten different LEO time periods. Further details about the source-to-receptor characteristic timescales can be found in the Supplement, section S1.

| LEO Time Period | Date/Time (Pacific Time) | Period Length (mins) | Total number of rBC particles analyzed[a] | Mean coating thickness (nm) | Median coating thickness (nm) | Characteristic timescale |
|---|---|---|---|---|---|---|
| L1 | 9 Sep. 2017, 12:00-1:00am | 60 | 397 | 62.2 | 53.5 | ~days to week |

| | | | | | | |
|---|---|---|---|---|---|---|
| L2 | 13 Sep. 2017, 11:59-12:58pm | 59 | 467 | 28.1 | 23.6 | ~minutes to hours |
| L3 | 20 Dec. 2017, 12:59-2:00pm | 61 | 79 | 49.3 | 47.7 | ~days to week |
| L4 | 21 Dec. 2017, 12:29-1:00pm | 31 | 318 | 14.3 | 12.0 | ~3 hours |
| L5 | 22 Dec. 2017, 9:59-10:15am | 16 | 1,176 | 14.6 | 12.2 | ~12 hours |
| L6 | 12 Nov. 2018, 12:00-1:00pm | 60 | 1,752 | 5.6 | -0.4 | ~8 hours |
| L7 | 14 Nov. 2018, 5:00-6:00am | 60 | 2,879 | 10.7 | 8.2 | ~17 hours |
| L8 | 17 Nov. 2018, 5:00-6:00am | 60 | 2,712 | 57.2 | 48.4 | ~days to week |
| L9 | 17 Nov. 2018, 7:00-8:00pm | 60 | 1,254 | 67.2 | 54.0 | ~days to week |
| L10 | 18 Nov. 2018, 10:00-11:00am | 60 | 4,778 | 40.6 | 31.2 | ~days to week |

[a] LEO coating thickness calculations shown in the table only include rBC-containing particles with core sizes between 200 and 250 nm.

Figure 10 illustrates the $CT_{BC}$ distributions and statistics of each LEO period. L1 and L2 were from the first campaign (September 2017). L1 is representative of ambient background rBC-containing particles from the first campaign. A period that did not exhibit any anomalously large rBC mass concentration values was chosen so that contributions from possible

nearby sources would not skew the mean $CT_{BC}$. On the other hand, L2 intentionally spans a period with many anomalously high rBC mass concentration values. Although these anomalous values were removed from the concentration time series discussed previously in section 2.4, the values were *not* removed for the LEO analysis of L2 in order to examine the relationship between $CT_{BC}$ and possible nearby emissions. As hypothesized, the rBC-containing particles from L2 were generally more thinly-coated than those from L1. The median $CT_{BC}$ from L2 was ~30 nm lower than that for L1, which

corroborates our hypothesis that the anomalously high mass concentration values in the first campaign included contributions from nearby, unidentified fossil fuel sources.

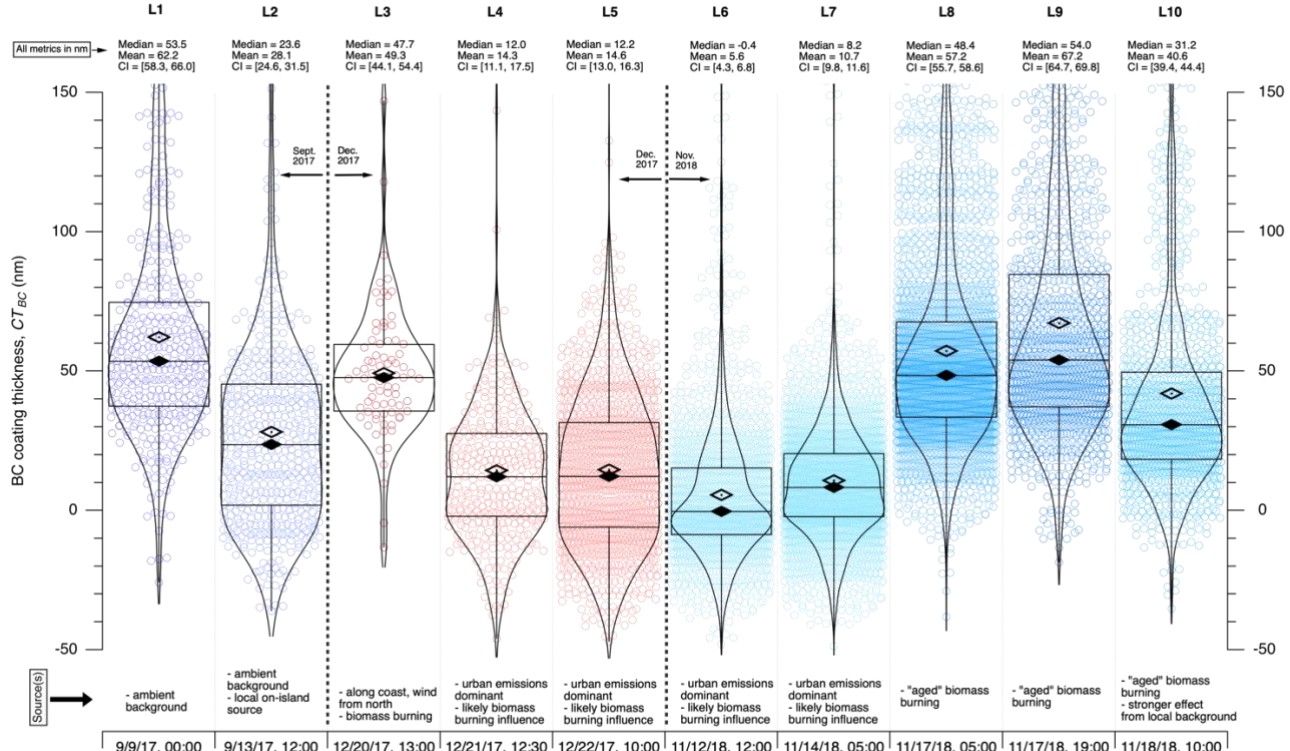

**Figure 10.** Violin plots that show the distribution of rBC coating thickness values for particles with rBC diameters between 200 and 250 nm, calculated for each LEO time period, L1 through L10. Each circle marker in the plot represents a particle analyzed by the LEO analysis and the curves for each "violin" shape represents the normalized probability density function of the coating thickness for each LEO period. The violin shape results from mirroring each probability density distribution along a vertical axis. Box-and-whiskers plots are also overlaid to show the quartiles (25th, 50th, and 75th percentiles) of the coating thickness distributions. The 95% confidence intervals (CI) based on Student's t-distribution are shown above each violin plot to demonstrate when the mean coating thickness values are statistically distinguishable from one another. The mean (unfilled diamond) and median (solid diamond) coating thicknesses are also indicated above each violin plot, and a brief description of sources for each LEO period is annotated below each distribution.

L3 through L5 are time periods from the second campaign (December 2017). L3 represents a period near the start of the second campaign (December 2017). The predominant wind direction during L3 was westerly, with a mean wind speed of ~4.5 m s$_{-1}$. HYSPLIT back-trajectories and CAMS data show that L3 likely included important contributions from the Thomas Fire in Santa Barbara and Ventura County. The PM$_{2.5}$ concentration gradient from CAMS was examined over time to track the movement of plumes that influenced the measurements during this time period. A few days prior to the start of the second campaign, the Thomas Fire resulted in a large aerosol plume westward over the Pacific Ocean. From visually tracking PM$_{2.5}$ concentration gradients, it appears that a large-scale, clockwise, atmospheric circulation brought aerosols from the Thomas Fire to Catalina Island around the time of L3 (see video 2 of Video Supplement). The average concentration during L3 was about an order of magnitude lower than the average concentration for the September campaign.

This could be partially attributed to the fact that L3 was around 1 to 2 pm, when the planetary boundary layer would be expected to increase in height, causing pollutant concentrations to decrease due to dilution. The median $CT_{BC}$ for L3 was

47.7 nm, which is slightly lower than for L1, which is representative of the ambient background conditions. The slightly smaller $CT_{BC}$ for L3 likely reflects the fact that mixing state is sensitive to the source of emissions. In this time period, urban emissions were likely mixed into the regional air mass, slightly lowering the median $CT_{BC}$. In this case, we have evidence to support that a larger fraction of measured rBC during L3 came from the local Thomas Fire mixed with nearby urban emissions, while L1 represents a mix of influences, including, but not limited to, aged biomass burning aerosols. The effect

of emissions sources on rBC mixing state is discussed in section 3.7.

L4 through L7 represent periods when the Los Angeles basin, Santa Barbara/Ventura counties, and San Diego county (to a lesser degree) were identified as major sources. Air masses measured during these periods likely contained a mixture of both urban emissions and biomass burning emissions (see Supplement section S2 and accompanying figures), although urban

emissions were likely dominant. Overall, these LEO periods exhibit the lowest median $CT_{BC}$, ranging from -0.4 to 12.2 nm. The potential relationship between aging time and $CT_{BC}$ is discussed further in section 3.7.

L8, L9, and L10 are the unique LEO periods from the third campaign (November 2018) with concurrently increased rBC concentrations and $f_{BC}$ (discussed in the previous section). We also observed significantly higher $CT_{BC}$ values during these

periods compared to L4–L7, with median $CT_{BC}$ ranging from 31.2 to 54.0 nm. We have strong evidence to support that the sampled particles include important contributions from aged rBC from the Northern California fires, particularly the Camp Fire (see section S2 in Supplement). The relatively high $CT_{BC}$ values in L8 and L9 (compared to other LEO periods) further support our claim that rBC-containing particles from Northern California fires were dominating our measurements during this time. L10 has a median $CT_{BC}$ of 31.2 nm, which is ~23 nm lower than the median value for L9. This reduction in the

median $CT_{BC}$ is also reflected in the decrease of the $f_{BC}$ values near the end of the campaign. Meteorological data, MODIS satellite images, and CAMS data during this time period suggest that sources from the Southern California (and possibly Central Valley) region contributed more to measurements during L10 than they did during L8 and L9, explaining the lower $CT_{BC}$ and higher overall concentrations. Wind speeds were lower on average for L10 compared to L8 and L9. The mean wind speed for L10 at LAX, based on 5-minute NOAA data, was ~1.3 m s-1, while the mean wind speeds for L8 and L9 were

~2.1 m s-1 and 1.6 m s-1, respectively. There was also a general shift of wind direction from westerly to north-easterly, approximately a half day before L10 (see Fig. S8 in Supplement). MODIS satellite imagery and CAMS data also confirm that local to regional sources were likely impacting the measurements more during this period (see video 3 and 4 of Video Supplement), compared to L8 and L9. The meteorology, in addition to local to regional sources of emissions from the Los Angeles basin and Southern California, likely explain the reduction in $CT_{BC}$ and the near doubling of the rBC concentration

level.

## 3.6 rBC core size

The number- and mass-based size distributions for rBC cores were assessed for periods L1 to L10. Similar to past studies, rBC core mass equivalent diameters between 70 and 450 nm are reported (Gao et al., 2007; Moteki and Kondo, 2007; Dahlkötter et al., 2014; Krasowsky et al., 2018). Figure 11 shows the rBC core size distributions and the corresponding log-normal fits for three LEO periods (L1, L5, and L10); we investigated these three LEO periods to assess whether log-normal fits adequately represent the actual rBC size distributions before presenting log-normal fits for all LEO periods. Previous studies have shown that rBC core size distributions are generally log-normal in the accumulation mode (Metcalf et al., 2012). Figure 11 shows that log-normal fits adequately capture the measured size distributions, though we cannot rule out the possibility of another rBC mode outside the detection limits of the SP2. Each of the ten LEO periods were characterized by a single mode within the range of the SP2 detection range. Although the peak of the measured number size distribution is not always discernible (e.g., L5 in Fig. 11), the conclusions made in the following analysis of rBC core size are unaffected by the uncertainty in the actual count median diameter. In addition, even in cases of ambiguous number size distribution peaks, we found that the right-hand side edge of the measured distribution was fit well by the log-normal distribution. Even if the distribution below the detection limit of ~70 nm deviated from the assumed log-normal fit, the median diameter is unlikely to be sufficiently affected to substantively change any of the following conclusions made in this section.

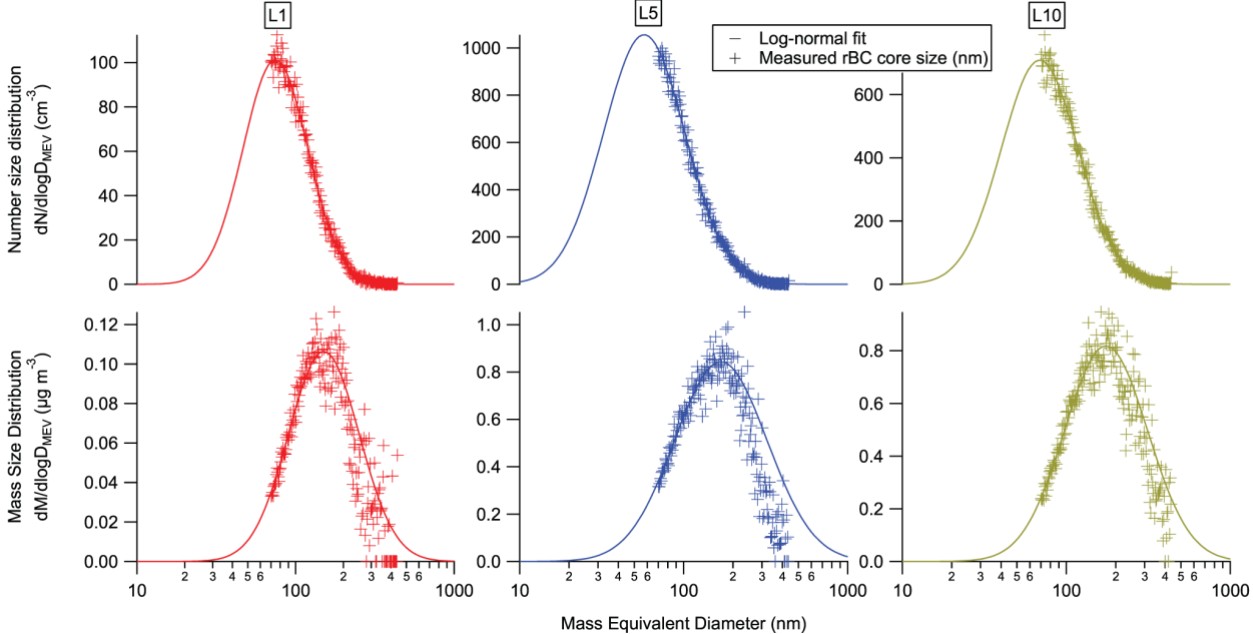

**Figure 11**. rBC core size distributions and corresponding log-normal fits for LEO periods L1, L5, and L10.

A survey of past studies that have reported log-normal fit rBC mass median diameter (MMDfit) and count median diameter (CMDfit) shows that the source of emissions has a strong influence on rBC core diameter (Cheng et al., 2018). The MMDfit

[CMD$_{fit}$] for BC$_{bb}$, which has been reported to range from ~130 nm to 210 nm [100 to 140 nm], is generally larger than the MMD$_{fit}$ [CMD$_{fit}$] for BC$_{ff}$, which has been reported to range from ~100 nm to 178 nm [38 to 80 nm] (Shiraiwa et al., 2007; Schwarz et al., 2008a; McMeeking et al. 2010; Kondo et al., 2011; Sahu et al. 2012; Metcalf et al., 2012; Cappa et al., 2012; Laborde et al., 2013; Liu et al., 2014; Taylor et al., 2014; Krasowsky et al., 2018). The MMD$_{fit}$ [CMD$_{fit}$] for well-aged background BC were reported to range from ~180 nm to 225 nm [90 nm to 120 nm] (Shiraiwa et al., 2008; Liu et al, 2010; McMeeking et al., 2010; Schwarz et al., 2010).

Figure 12 shows the rBC MMD$_{fit}$ and CMD$_{fit}$ for each LEO period in this study. Based on the source identification discussed in section 3.1 and section S2 in the Supplement, the MMD$_{fit}$ and CMD$_{fit}$ values in this study are generally consistent with the ranges reported in past studies. For BC$_{bb}$ (L3, L8, L9, L10), MMD ranged from 149 nm to 171 nm, which is within the range of ~130 nm to 210 nm reported in past studies. For BC$_{ff}$ (L2, L4, L7), the MMD$_{fit}$ dropped, ranging from 112 nm to 129 nm. This falls within the range of ~100 nm to 178 nm previously reported for measurements of urban emissions.

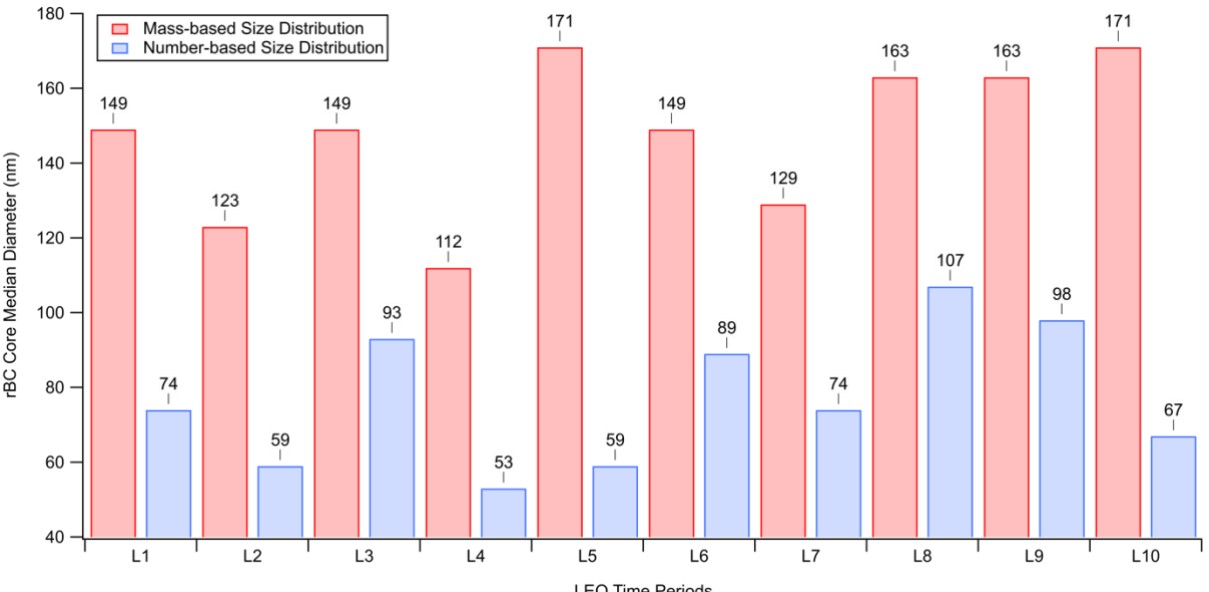

**Figure 12.** Median rBC core diameter for both mass and number size distribution lognormal fits.


Past literature has also mentioned the possibility of coagulation affecting rBC core size (Bond et al., 2013). Shiraiwa et al. (2008) observed an increase in rBC core diameters in aged plumes compared to fresher urban plumes in the East Asian outflow, suggesting that coagulation can alter the rBC size distribution during atmospheric transport (i.e., aging). Although the emissions source type appears to be the dominant influence on rBC core sizes in our study, we cannot completely eliminate the possibility of any coagulation occurring between the point of emission and point of measurement. Our measurements suggest that coagulation would have a negligible impact at measured number concentrations, but number

concentrations would be orders of magnitude higher near points of emission (especially in dense biomass burning plumes), leaving open the possibility of non-negligible coagulation near sources, in certain cases.

## 3.7 Impact of emissions source and aging on rBC mixing state

The dominant factor that influences rBC core size (i.e., emission source type) also strongly influences rBC mixing state. Figure 13 shows a scatter plot of one-minute mean $CT_{BC}$ versus one-minute mean rBC core diameter, using data from all three campaigns. A positive correlation was found, with $r = 0.55$ and $r_2 = 0.30$. This correlation suggests that larger contributions from biomass burning (as opposed to fossil fuel) are associated with increases in both the rBC core size and the BC coating thickness. Figure S22 shows the $CT_{BC}$ distributions for different rBC core size ranges, and a similar relationship

between the two variables can be observed. As the core size increases (lighter to darker curves), a broader right-hand side tail is observed in the $CT_{BC}$ normalized distributions for each campaign, implying an increased probability density for thicker coatings as rBC diameter increases.

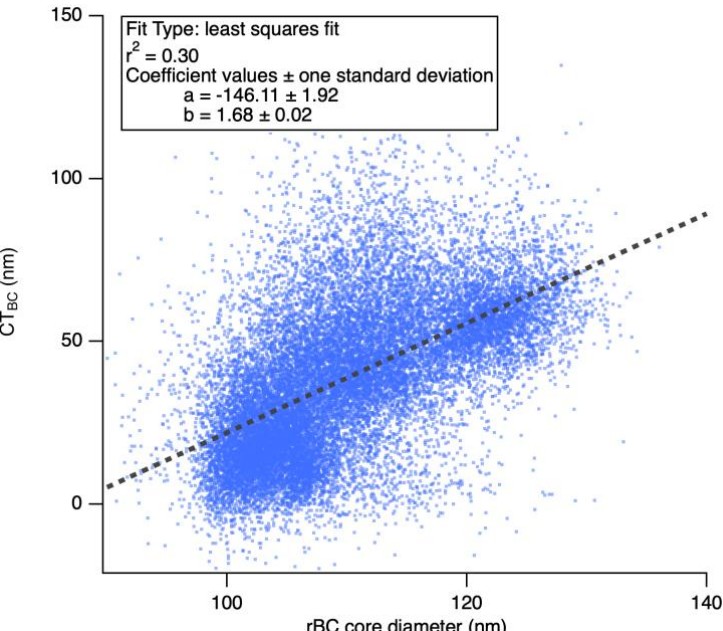

**Figure 13.** rBC coating thickness versus rBC core diameter. Each point on the plot represents a 1-minute mean. Data from
all three campaigns are shown. $CT_{BC}$ values are calculated for particles with rBC core diameters between 200–250 nm. The line represents the least-squares linear regression to the one-minute mean data points. There is a statistically significant positive correlation shown between $CT_{BC}$ and rBC core diameter, as shown in the summary box in the top left corner.

The time evolution of both $CT_{BC}$ and rBC core size is represented in a series of scatter plots in Fig. 14 and 15. In each of the figures, the scatter between one-minute mean $CT_{BC}$ and rBC CMD are grouped into six-hour time intervals for both the

second (December 2017) and third (November 2018) periods, respectively. In these figures, the time evolution of the rBC physical properties can be examined in detail and compared to periods of known emissions source impacts. There are a few significant patterns worth highlighting here. First, the influence of $BC_{bb}$ can be observed between 16 and 18 November 2018

in Fig. 15. Both $CT_{BC}$ and CMD drastically increase for a prolonged period of time, implying an impact from the Camp Fire plume from Northern California. Second, the scatter plots for 20 to 22 December 2017 and 12 to 15 November 2018 show that there is some variability over time in the cluster shapes, which can be explained by the local wildfires that were confirmed to influence the broader LA basin plume (see section S2 for details regarding source attribution). Although the scatter plots during these time periods support that $BC_{ff}$ was largely dominant, there are some periods where the CMD spread

deviates quite noticeably (e.g., Fig. 14, 06:00 21 Dec. 2017), or even periods that show two distinct clusters (e.g., Fig. 15, 12:00 12 Nov. 2018), supporting our claim that local wildfires in Southern California were indeed influencing our measurements. A similar figure for the first campaign (September 2017) is included in the Supplement as Fig. S23, but not shown here because of the relatively stable mixing state and size of $BC_{aged,bg}$. These figures confirm the general patterns noted in previous sections regarding the effects of different sources on rBC mixing state.

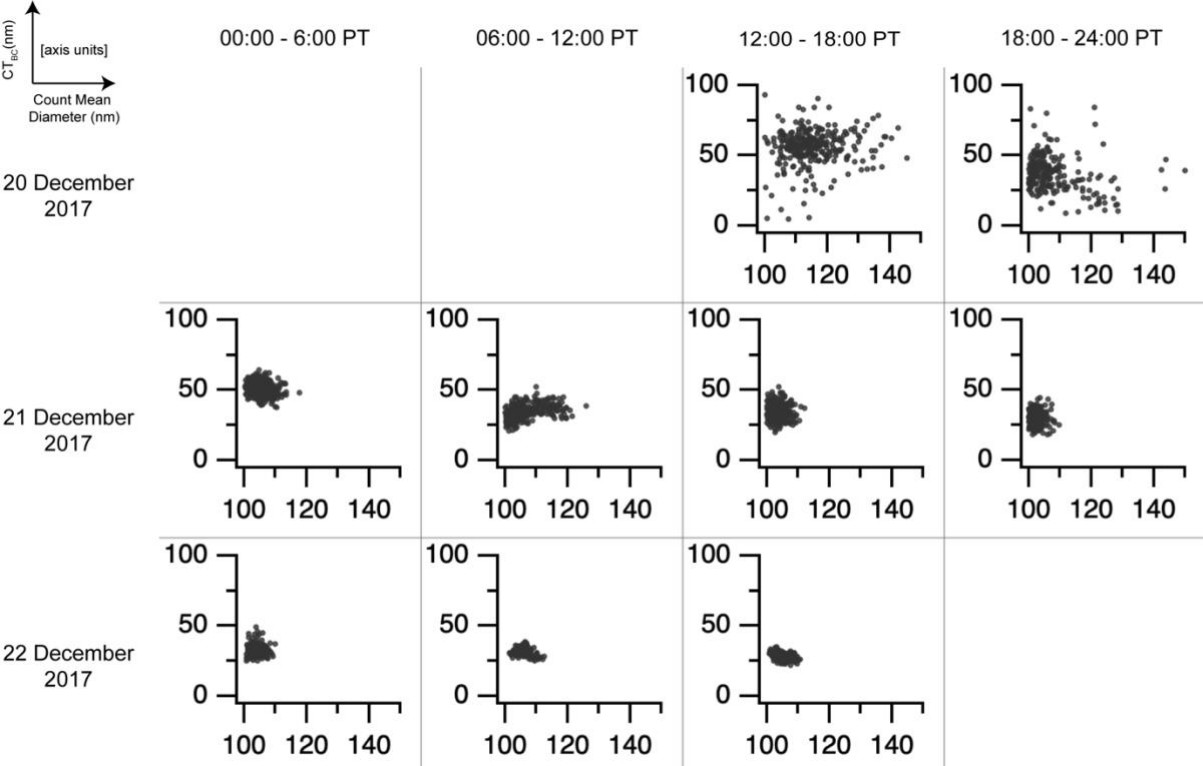

**Figure 14.** Matrix of scatter plots showing the time evolution of $CT_{BC}$ (nm) and rBC count mean diameter (nm) for the second campaign (December 2017). Axes labels are shown in the upper left. A scatter plot is shown for each six-hour time interval of the day, starting at 00:00 Pacific Time, and for each day of the campaign. The columns of the matrix denote the time interval of the day, and the rows of the matrix denote the days of the campaign. Each point within a plot represents a

one-minute mean value for both $CT_{BC}$ and count mean diameter.

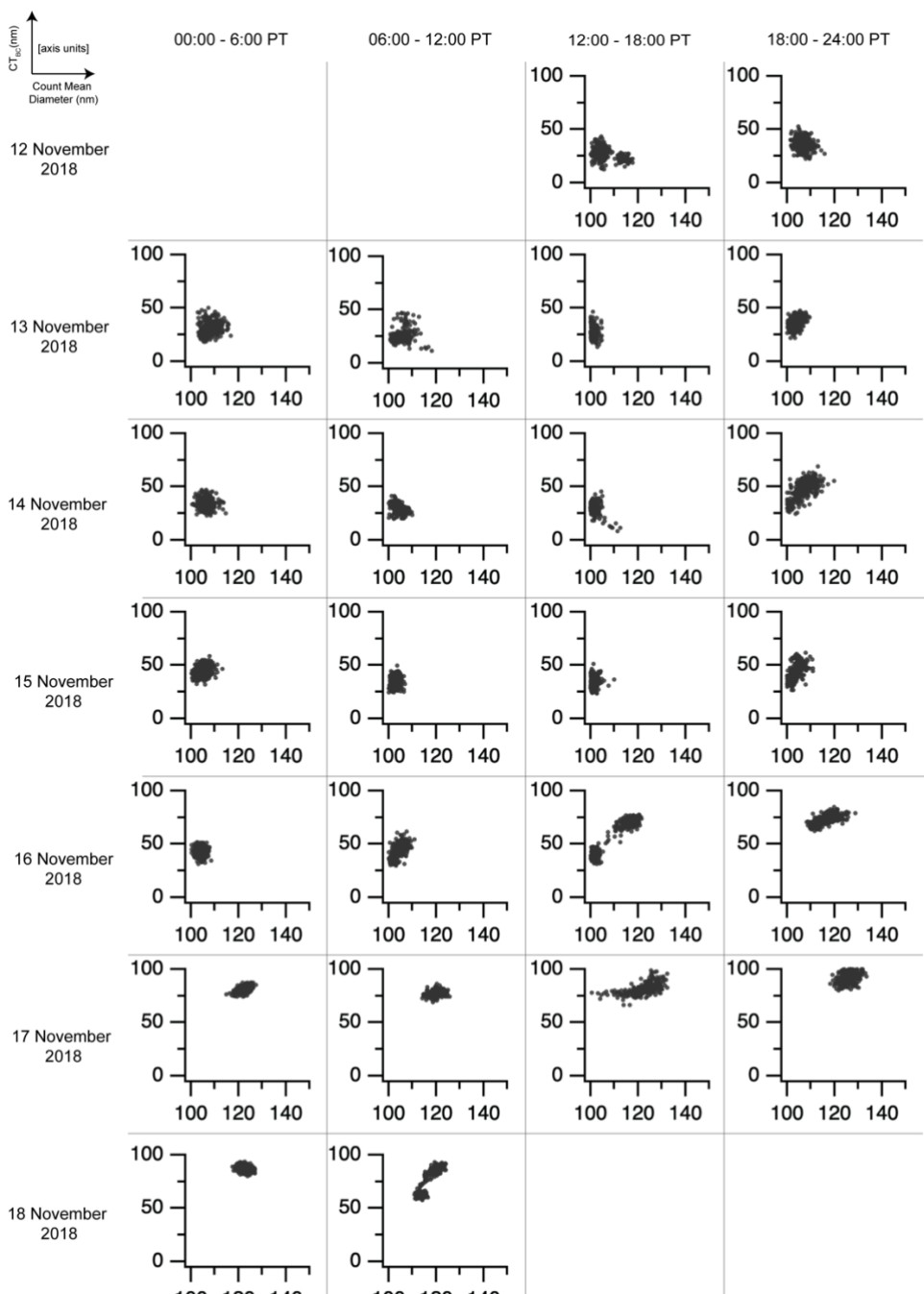

**Figure 15.** Matrix of scatter plots showing the time evolution of $CT_{BC}$ (nm) and rBC count mean diameter (nm) for the third campaign (November 2018). Axes labels are shown in the upper left. A scatter plot is shown for each six-hour time interval of the day, starting on 00:00 Pacific Time, and for each day of the campaign. The columns denote the time interval of the day, and the rows denote the day of the campaign. Each point within a plot represents a one-minute mean value within that six-hour interval for both $CT_{BC}$ and count mean diameter.

When the scatter plots of one-minute mean $CT_{BC}$ and rBC mean diameter are aggregated by campaign, distinct patterns emerge. Contour plots representing the 2-d joint histograms of these two variables are shown in Fig. 16. Each campaign

exhibits a distinct pattern that is representative of the emissions sources and relative age of the measured air masses. Figures 16b and 16e show a single cluster for the second campaign (September 2017) characterized by relatively thin coatings and smaller rBC core diameters, compared to the other campaigns. Figures 16c and 16f on the other hand show two distinct clusters for the third campaign (November 2018). One cluster represents thickly-coated particles with larger rBC core diameters, and the other represents more thinly-coated particles with smaller rBC core diameters. The thinly-coated/smaller

rBC core cluster for the third campaign exhibits some similarities to the single cluster for the second campaign. Figures 16a and 17d show two overlapping clusters for the first campaign (September 2017), which fall loosely in between the thickly-coated and thinly-coated clusters from the third campaign. For easy reference, a cluster characterized by thin coatings and smaller rBC cores will be referred to as a "BC$_{ff}$ cluster," a cluster with thick coatings and larger rBC cores will be referred to as a "BC$_{bb}$ cluster," and the bimodal, mixed cluster will be referred to as the "BC$_{aged,bg}$ cluster."

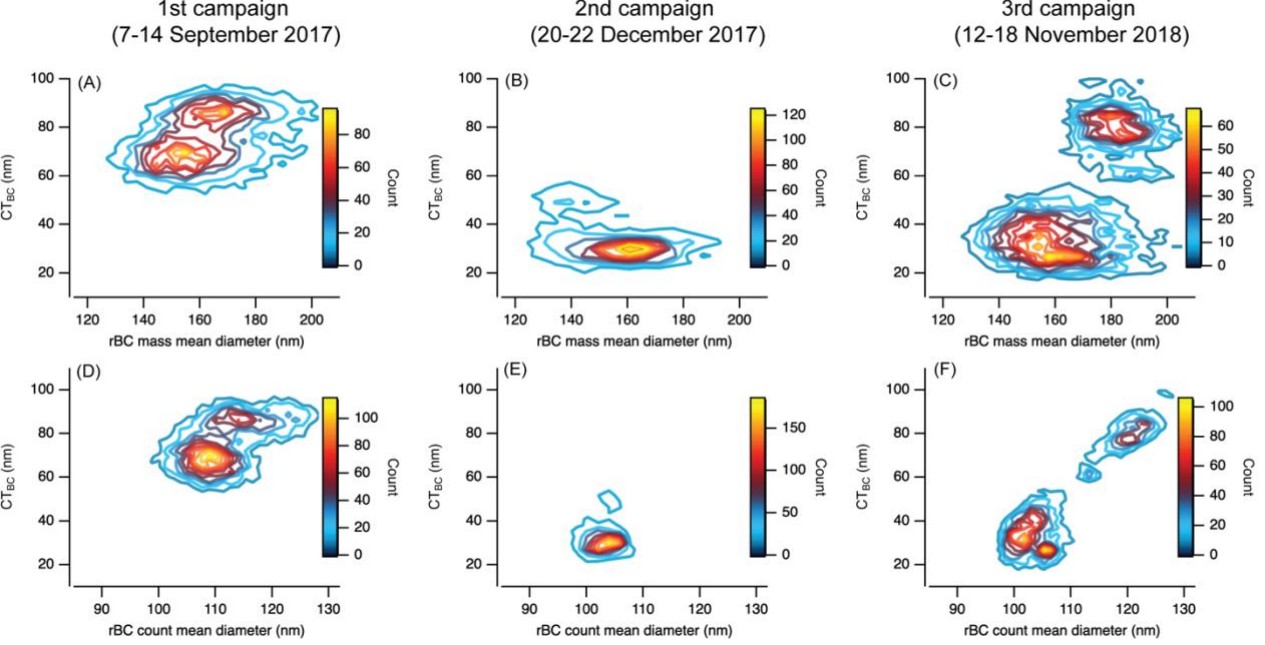


**Figure 16.** Contour plots of count as a function of one-minute mean BC coating thickness ($CT_{BC}$) and one-minute mean rBC core diameter. This figure can be interpreted as a 2-d joint histogram, converted to a contour plot. Each count represents a single one-minute mean data point. The contours are created based on the 2-d joint histogram that is calculated using a 50x50 grid within the range of all one-minute mean data. Panels (a), (b), and (c) in the first row show mass mean diameter on the horizontal axes, while panels (d), (e), and (f) in the second row show count mean diameter.


Within the context of the identifiable sources discussed in previous sections (section 3.1 and S2), it is evident that these distinct clusters in Fig. 16 are strongly influenced by emissions source type. A BC$_{bb}$ cluster is present in the third campaign

(November 2018) when impacts from long-range transported biomass burning emissions were identified, but not in the second campaign (December 2018). Furthermore, a BC$_{ff}$ cluster is present in both the second (December 2017) and third campaign, but not in the first campaign (September 2017). This shows that fresh (age < 1 d), fossil fuel emissions from the LA basin and the surrounding southern California region are characterized by thin coatings and smaller core size, confirming what has also been observed in other past field studies (Laborde et al., 2012; Liu et al., 2014; Krasowsky et al., 2018).

The BC$_{aged,bg}$ cluster (Fig. 16a, 16d) exhibits two distinct modes within the same cluster. One mode is characterized by a peak $CT_{BC}$ [CMD] that is ~20 nm [~10 nm] larger than the other mode. Within the context of BC$_{aged,bg}$, this mode is referred to as the larger mode, while the other mode with smaller $CT_{BC}$ and CMD is referred to as the smaller mode. Based on past reported measurements of rBC core size and mixing state, we deduce that the larger and smaller modes are representative of biomass burning BC and fossil fuel BC, respectively (McMeeking et al., 2011a; Laborde et al., 2013; Liu et al., 2014; Sahu et al., 2012; Schwarz et al., 2008a; Krasowsky et al., 2018; Corbin et al., 2018). This suggests that BC$_{aged,bg}$ advecting over the Pacific Ocean during typical meteorological conditions contain rBC from both biomass burning and fossil fuel emissions sources. Aged background air masses are likely to contain aerosol from a mix of sources.

In addition to emissions source type, atmospheric aging also appears to have an observable effect on the mixing state. Table 2 lists the range of estimated "source-to-receptor" timescales for rBC-containing particles measured during LEO time periods L1 to L10. In short, the first campaign (September 2017) is broadly characterized by source-to-receptor timescales on the order of days to a week. The second campaign (December 2017) is characterized by timescales of less than one day. And the third campaign (November 2018) is characterized by timescales of less than one day for the first four days of the campaign, and timescales of approximately days to a week for the last two days of the campaign.

With regards to aging, we first observe that BC$_{ff}$ particles do not develop thick coatings within the timescales observed in this study. This suggests that a timescale of less than one day is not sufficient to thickly coat rBC-containing particles from fossil fuel combustion in the lower boundary layer, in the Southern California region. Although a modestly higher $CT_{BC}$ is observed during urban-dominated time periods, relative to $CT_{BC}$ ~ 0 nm observed by Krasowsky et al. (2018) inside the LA basin, this is likely due to the effects of local biomass burning emissions mixing into the broader urban plume in both December 2017 and November 2018, as discussed above (also see section S2). While we observed mostly thinly-coated BC from time periods dominated by fossil fuel emissions, we acknowledge that the timescale required to acquire coatings on BC will likely differ by location because of variations in meteorology, pollution concentrations, and emission source profiles.

On the other hand, BC$_{bb}$ were generally more thickly-coated, although the time evolution of the mixing state could not be quantified directly in this study. Fresh BC$_{bb}$ had slightly lower $CT_{BC}$ compared to that of aged BC$_{bb}$ (e.g., L3 vs. L9), but higher $CT_{BC}$ compared to that of fresh BC$_{ff}$ (e.g., L3 vs. L4). The overall higher $CT_{BC}$ for aged BC$_{bb}$ relative to fresh BC$_{bb}$

indicates that significant coating formation can occur within timescales of ~1 day to ~1 week for $BC_{bb}$, even after rapid coating formation that occurs soon after emission. An important caveat is that $CT_{BC}$ of $BC_{bb}$ may not be monotonically

increasing over time. Past studies have observed rapid coating of $BC_{bb}$ within one day to more than 100 nm (Perring et al., 2017; Morgan et al., 2020), but we observed a median $CT_{BC}$ of 47.7 nm for L3, which suggests that $CT_{BC}$ for $BC_{bb}$ might decrease during atmospheric transport under certain conditions and could again increase later at longer timescales (e.g., median $CT_{BC}$ of 54.0 nm for L9), although we would need simultaneous measurements near the point of biomass burning emissions in order to confirm this for a specific plume. Further research is necessary to confirm this process in more field

measurements, and to determine the various mechanisms that may be driving the potential loss of rBC coating in biomass burning plumes. We make no definitive claims about the rate of change of $CT_{BC}$ for $BC_{bb}$ throughout atmospheric transport since we measured $CT_{BC}$ at one location. Nonetheless, our measurements suggest that $CT_{BC}$ for fresh Southern California $BC_{bb}$ were generally lower than $CT_{BC}$ for aged Northern California $BC_{bb}$.

The contour plots for the first campaign (September 2017), shown in Fig. 16a and 16d, offer additional perspective on how aging can affect rBC mixing state within well-aged background air masses over longer aging timescales (~days to week). The first notable feature of the $BC_{aged,bg}$ cluster is that the smaller mode is significantly more coated than the $BC_{ff}$ clusters found in Fig. 16 for the second (December 2017) and third (November 2018) campaigns. The peak of the smaller mode of the $BC_{aged,bg}$ cluster is at least ~35 nm higher than the peak of the respective $BC_{ff}$ clusters in Fig. 16b, 16c, 16e, and 16f.

Assuming that this smaller mode represents fossil fuel influenced BC (e.g., urban, ship, aviation emissions), this confirms that while $BC_{ff}$ may not become thickly-coated within a day, they seem to acquire coatings over longer timescales.

The evolution of rBC mixing state and rBC size distribution has important implications on accurately assessing the regional climate benefits of black carbon reductions, particularly in California, and also reducing uncertainty in global radiative

forcing of BC. Understanding the impact of varying emissions source types and atmospheric aging in different regional contexts is crucial for accurately quantifying the enhancement of BC light absorption, and also for determining BC lifetime in the atmosphere since hygroscopic coating material can enhance the particle's susceptibility to wet deposition (Zhang et al., 2015). The rBC mixing state results from this study add to a growing body of evidence that suggests that biomass burning emissions and longer aging timescales generally lead to more thickly-coated rBC particles. These results also

emphasize the need for more field measurements of rBC mixing state in various regions around the world to further understand how different emissions source profiles and atmospheric aging ultimately effect rBC physical properties in various, real-world atmospheric contexts.

### 3.8 Comparison to past studies quantifying $CT_{BC}$ using the SP2

Overall, the range of $CT_{BC}$ calculated in this study is in agreement with reported values from past studies. Table 3 presents a comprehensive list of $CT_{BC}$ values from various studies, categorized by dominant emissions source type and sorted alphabetically by first author name.

For $BC_{bb}$, the mean $CT_{BC}$ ranged between ~40–70 nm in this study. This range overlaps with the range of values reported by Morgan et al. (2020), Pan et al. (2017), Sahu et al. (2012), Schwarz et al. (2008a), and Sedlacek et al. (2012). For $BC_{ff}$, the mean $CT_{BC}$ ranged between ~5–15 nm in this study. This range overlaps with the range of values reported by Krasowsky et al. (2018), Laborde et al. (2012), Liu et al. (2014), Sahu et al. (2012), McMeeking et al. (2011a), Corbin et al. (2018), and Schwarz et al. (2008a). For $BC_{aged,bg}$, the mean $CT_{BC}$ was ~60 nm in this study. This value falls within the range of values reported by Laborde et al. (2013), Schwarz et al. (2008a), and Shiraiwa et al. (2008).

An important caveat to note when making inter-study comparisons is that the studies that reported higher $CT_{BC}$ ranges (relative to this study) tended to have a lower value for the lower rBC core diameter threshold. For example, Gong et al. (2016) reports a $CT_{BC}$ range of 110–300 nm for biomass burning emissions using an rBC core diameter range of 80–180 nm. Since the scattering detection limit is accurate down to ~170 nm for the SP2, this implies that the inclusion of particles with rBC core sizes smaller than 170 nm will bias the average $CT_{BC}$ values higher because smaller rBC particles with optical diameters below the scattering detection will not be included in the LEO analysis. Dahlkötter et al. (2014), Gong et al. (2013), Perring et al. (2017), Taylor et al. (2014), Cheng et al. (2018), Metcalf et al. (2012), Raatikainen et al. (2015), and Sharma et al. (2017) all reported $CT_{BC}$ for rBC-containing particles in a size range that includes rBC cores smaller than 170 nm. There is value in reporting $CT_{BC}$ for rBC particles with core sizes smaller than 170 nm because it will show the relative abundance of coated rBC-containing particles exceeding the lower scattering detection limit, but care must be taken when comparing $CT_{BC}$ values calculated with varying rBC core size restrictions.

For future studies using the SP2, we suggest that at a minimum, the rBC core size range be explicitly stated if $CT_{BC}$ is being quantified and reported. Furthermore, it would be useful to establish some standardized guidelines for reporting $CT_{BC}$ so that future inter-study comparisons can serve as reliable benchmarks. As shown in Figure S22 and discussed earlier, the range of rBC core diameters used for the calculation of $CT_{BC}$ has a significant effect on the $CT_{BC}$ statistics. These ranges must be considered in order to accurately represent the physical parameterization of BC mixing state and size distributions in models.

**Table 3.** Summary table of rBC coating thickness values reported in previous studies using the SP2.

| Dominant source | Coating thickness (nm) | rBC core diameter (nm) | rBC age | Description | Time period | Reference |
|---|---|---|---|---|---|---|
| Biomass burning emissions | ~40–70[a] | 200–250 | ~days– wk | Ground-based measurements on Catalina Island (~70 km SW of Downtown LA) | 17–18 Nov 2018 | This study |
| | 105–136 | 140–220 | ~3–4 d | Airborne measurements of the Pagami Creek Fire plume (Minnesota, US) conducted over Germany | 16 Sep 2011 | Dahlkötter, 2014 |
| | 110–300 | 80–130 | – | Ground-based measurements in Shanghai, China | 5–10 Dec 2013 | Gong, 2016 |
| | 11–15 | 200–260 | – | Ground-based measurements in Paris, France | 15 Jan–15 Feb 2010 | Laborde, 2013 |
| | 100–300 | – | – | Ground-based measurements in London, during periods significantly influenced by solid fuel burning | 22–24 Jan 2012 | Liu, 2014 |
| | 40–120 | – | < 3 h | Airborne measurements across the Amazon and Cerrado | Sep and Oct 2012 | Morgan, 2020 |
| | 11–54 | 190–210 | < 10 s | Burning experiments in laboratory combustion chamber | – | Pan, 2017 |
| | 90–110 | 160–185 | < 2 d | Airborne measurements of the Yosemite Rim Fire, CA | Aug 2013 | Perring, 2017 |
| | 20–80 | 200 | – | Airborne measurements over California during ARCTAS-CARB campaign | 15–30 Jun 2008 | Sahu, 2012 |
| | 65±12 | 190–210 | 0.5–1.5 h | Airborne measurements over Houston and Dallas, TX | 20–26 Sep 2006 | Schwarz, 2008a |
| | 40–70 | – | ~days | Ground-based measurements of a wildfire plume from the Lake Winnipeg area in Canada, conducted in Long Island, NY | 2 Aug 2011 | Sedlacek, 2012 |
| | 79–110 | 130–230 | 1–2 d | Airborne measurements over wildfires in eastern Canada and North Atlantic | Jul–Aug 2011 | Taylor, 2014 |
| | ~5–15[a] | 200–250 | < 1 d | Ground-based measurements on Catalina Island (~70 km SW of Downtown LA) | 17–18 Nov 2018 | This study |
| Fossil fuel emissions | 22–40 | 130–160 | < 3 h | Airborne measurements over the Athabasca oil sands in Canada | 13 Aug–7 Sep 2013 | Cheng, 2018 |
| | 17–39 | 160–190 | < 3 h | Airborne measurements over the Athabasca oil sands in Canada | 13 Aug–7 Sep 2013 | Cheng, 2018 |
| | 50–130 | 60-80 | – | Ground-based measurements in Shanghai, China | 5–10 Dec 2013 | Gong, 2016 |
| | ~0-40 | 220-260 | – | Laboratory measurements of marine engines | Nov-Dec 2014 | Corbin, 2018 |
| | ~0–24 | 240–280 | < 7 h | Ground-based measurements in the Los Angeles basin | Aug–Oct 2016 | Krasowsky, 2018 |
| | < 30 | – | – | Ground-based measurements near central Manchester, UK | 3–16 Aug 2010 | McMeeking, 2011a |
| | 2±10 | 200–260 | – | Ground-based measurements in Paris, France | 15 Jan–15 Feb 2010 | Laborde, 2013 |
| | 0–50 | – | – | Ground-based measurements in London, during periods dominated by traffic sources | 31 Jan–1 Feb 2012 | Liu, 2014 |
| | 99±20 | 90-260 | – | Airborne measurements in the Los Angeles Basin and surrounding outflows | May 2010 | Metcalf, 2012 |
| | 88±4[b] | 180 | ~hours | Ground-based measurements in Gual Pahari, India | 3 Apr–14 May 2014 | Raatikainen, 2015 |
| | 0–40 | 200 | – | Airborne measurements over California during ARCTAS-CARB campaign | 15–30 Jun 2008 | Sahu, 2012 |
| | 20±10 | 190–210 | 2–3.5 d | Airborne measurements over Houston and Dallas, TX | 20–26 Sep 2006 | Schwarz, 2008a |
| | 30–40 | 200 | ~ 6 h | Ground-based measurements of fresh emissions from Japan, conducted on Fukue Island, Japan | Mar–Apr 207 | Shiraiwa, 2008 |
| | ~60[a] | 200–250 | ~days– wk | Ground-based measurements on Catalina Island (~70 km SW of Downtown LA) | 7–14 Sep 2017 | This study |
| Remote / background / continental / highly-aged | 130–300 | 60–80 | – | Ground-based measurements in Shanghai, China | 5–10 Dec 2013 | Gong, 2016 |
| | 37–93 | 200–260 | – | Ground-based measurements in Paris, France | 15 Jan–15 Feb 2010 | Laborde, 2013 |
| | 188±31 | 90-260 | – | Airborne measurements in the free troposphere | May 2010 | Metcalf, 2012 |
| | 75–100 | 150–200 | – | Ground-based measurements at the Pallas GAW (Finnish Arctic) | Dec 2011–Jan 2012 | Raatikainen, 2015 |
| | 90±5[b] | 180 | ~hours | Ground-based measurements in Mukteshwar, India | 9 Feb–31 Mar 2014 | Raatikainen, 2015 |

| | | | | | |
|---|---|---|---|---|---|
| 48±14 | 190–210 | – | Airborne measurements over Houston and Dallas, TX | 20–26 Sep 2006 | Schwarz, 2008a |
| < 30 nm | 190–210 | – | Airborne measurements over Costa Rica, 1-5 km | 6–9 Feb 2006 | Schwarz, 2008b |
| 20–36 | 160–180 | – | Ground-based measurements in Alert, Nunavut, Canada (within Arctic Circle) | Mar 2011–Dec 2013 | Sharma, 2017 |
| ~60 | 200 | ~days | Ground-based measurements of Asian continental air masses, conducted on Fukue Island, Japan | Mar–Apr 207 | Shiraiwa, 2008 |

[a] The range of values shown represent the approximate range of the mean $CT_{BC}$.

[b] The absolute coating thickness was calculated from the ratio of rBC core diameter to particle mobility diameter as presented in the study.

Note: A dash ("-") indicates that the value was not reported, or it could not be identified

**4 Conclusion**

This study investigates the concentration, size distribution, and mixing state of rBC on Catalina Island (~70 km southwest of Los Angeles) using a single-particle soot photometer (SP2). Measurements were taken during three separate campaigns with varying meteorological conditions and emission sources, in September 2017, December 2017, and November 2018. During the first campaign (7 to 14 September 2017), westerly winds dominated and thus the sampling location was upwind of the dominant regional sources of BC (i.e., urban emissions from the Los Angeles basin). The measurements from the first campaign were largely characteristic of well-aged background levels of rBC over the Pacific Ocean, away from the broader urban Los Angeles plume ($BC_{aged,bg}$). During the second and third campaigns (20 to 22 December 2017, 12 to 18 November 2018), due to atypical Santa Ana wind conditions, we measured biomass burning rBC ($BC_{bb}$) from large wildfires in California and fossil fuel rBC ($BC_{ff}$) from the Los Angeles basin. Furthermore, during the third campaign, rBC from the Camp Fire in Northern California was measured, allowing us to compare the mixing state of aged $BC_{bb}$ (from Camp Fire) to fresher rBC (from Southern California fires and urban Los Angeles emissions). The measurements from these three campaigns showed that rBC physical properties (rBC core size and mixing state) were influenced by (i) emissions source type, and (ii) atmospheric aging.

$BC_{bb}$ generally had larger core diameters than $BC_{ff}$. The MMD [CMD] of $BC_{bb}$ was observed to be ~180 nm [120 nm], while MMD [CMD] of $BC_{ff}$ was observed to be ~160 nm [100 nm]. $BC_{aged,bg}$ showed a bimodal rBC core size distribution, with MMD [CMD] peaks at ~170 nm [115 nm] for the larger mode, and ~153 nm [109 nm] for the smaller mode. The bimodal rBC core size distribution from the aged background during the first campaign (September 2017) showed that background rBC above the Pacific Ocean during typical meteorological conditions were likely from a mix of both fossil fuel and biomass burning emissions.

We found emissions source type also strongly affected rBC mixing state. On average, $BC_{ff}$ was either uncoated or very thinly-coated, with mean coating thickness ($CT_{BC}$) ranging from ~5 to 15 nm and mean fraction of thickly coated particles ($f_{BC}$) of less than 0.15. In contrast, $BC_{bb}$ was more thickly-coated, with mean $CT_{BC}$ ranging from ~40 to 70 nm and $f_{BC}$ ranging from ~0.23 to 0.47. $BC_{aged,bg}$ was characterized by a mean $CT_{BC}$ of ~60 nm and $f_{BC}$ of ~0.27, further confirming that a mix of biomass burning and urban emissions sources were likely mixed in these aged background air masses.

We also assessed the effect of aging on both $BC_{bb}$ and $BC_{ff}$. For $BC_{ff}$, we observed that timescales of less than one day were not sufficient for fossil fuel rBC particles to become thickly coated. This is in direct contrast to $BC_{bb}$, which has been shown in previous studies to acquire thick coatings within hours or even minutes, near the source of emissions. For $BC_{bb}$, we

observed higher values of $f_{BC}$ and $CT_{BC}$ during periods that included contributions from the Camp Fire in Northern

California, compared to periods of fresh biomass burning impacts from local Southern California fires (e.g., L3). The average $CT_{BC}$ during the Camp Fire impacted period was ~18 nm higher than the average $CT_{BC}$ during L3, when we identified Southern California fires as the main emission source. Likewise, we also observed an increased $CT_{BC}$ with aging for $BC_{ff}$, by comparing the aged $BC_{ff}$ mode of the $BC_{aged,bg}$ distribution to fresh $BC_{ff}$ during periods when emissions from the LA basin dominated. We found that coatings on $BC_{ff}$ within $BC_{aged,bg}$ were ~35 nm thicker than $BC_{ff}$ from fresh LA basin

emissions. Overall, our measurements suggest that aging increases the coating thickness on both $BC_{ff}$ and $BC_{bb}$, which is consistent with previous research. We did not quantify the rate of change of coating thickness since we were unable to track the evolution of the mixing state during source-to-receptor transport.

The measurements reported in this study agree with past research that investigated impacts of source type and aging on rBC

physical properties. This study further highlights the complexity of rBC mixing state and demonstrates how meteorology, emissions source type, and atmospheric aging can affect the size distribution and mixing state of BC, even within the same region. Further measurements of rBC physical properties, along with pollutant measurements that allow for robust source apportionment, would improve our understanding of BC mixing state in various regions with different atmospheric contexts. Given that we identified less than 20 studies that quantify $CT_{BC}$ using the LEO method, this study confirms that further

measurements are necessary to narrow the quantitative bounds of rBC mixing state in our climate system, which has important implications on BC absorption enhancement and atmospheric lifetime. We also suggest that future studies further examine the BC mixing state as a function of altitude, as well as the role of combustion conditions on mixing state (e.g., flaming versus smoldering), especially in real-world field measurements.

**Data availability**

Processed data is available at the following Harvard Dataverse repository:
https://dataverse.harvard.edu/dataverse/catalina_rbc_2017_2018.

DOI citations to individual datasets:
Ko, Joseph, 2019, "Time Series Data for Catalina Island rBC Measurements 2017-2018",
https://doi.org/10.7910/DVN/UJAGHY, Harvard Dataverse, V1

Ko, Joseph, 2019, "rBC Coating Thickness from Catalina Island rBC Measurements 2017-2018",
https://doi.org/10.7910/DVN/AAYMHH, Harvard Dataverse, V2

Ko, Joseph, 2019, "rBC Size Distribution from Catalina Island rBC Measurements 2017-2018",
https://doi.org/10.7910/DVN/CIMVS4, Harvard Dataverse, V1

Due to the extremely large file sizes for the raw SP2 data, they are not publicly available but may be available upon request to the corresponding author.

**Video supplement**

CAMS model output showing the Camp Fire and Southern California plumes during the November 2017 campaign: https://doi.org/10.5446/42893

NASA MODIS images showing the Camp Fire plume during the November 2017 campaign:
https://doi.org/10.5446/42892

CAMS model output showing the Camp Fire plume reaching Southern California during the December 2018 campaign: https://doi.org/10.5446/42943

Large-scale circulation of aerosols off the California coast during the December 2018 Campaign: https://doi.org/10.5446/42942

**Supplement**

[DOI link will be inserted once supplied by ACP]

**Competing interests**

The authors declare no competing interests.

**Acknowledgements**

This research was supported by the National Science Foundation under CAREER grant CBET-1752522. This research was also funded in part by the Indo-US Science and Technology Forum.

We acknowledge the use of data from the European Centre for Medium-Range Weather Forecasts (ECMWF). Neither the European Commission nor ECMWF is responsible for any use that may be made of the information it contains.

We acknowledge the NOAA Air Resources Laboratory (ARL) for the provision of the HYSPLIT transport and dispersion model and/or READY website (http://www.ready.noaa.gov) used in this publication.

We acknowledge the use of imagery from the NASA Worldview application (https://worldview.earthdata.nasa.gov/), part of the NASA Earth Observing System Data and Information System (EOSDIS).

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
