# Peer review of "Measurements to determine mixing state of black carbon emitted from the 2017/2018 California wildfires and urban Los Angeles"

_Atmospheric Chemistry and Physics, 2019_

## Referee Comment (RC1) · Anonymous Referee #1 · 24 Oct 2019

This paper presents SP2 measurements downwind of the USA and contrasts urban and biomass burning emissions after varying degrees of ageing. These observations are of importance to the regional and global climate modelling communities, as the evolution of black carbon in the atmosphere and formation of coatings affects both the optical properties and scavenging lifetime. Therefore, work of this nature is very relevant to ACP.

While the paper is well written and the graphs well presented, I find this paper somewhat lacking in terms of the interpretation side. The dataset is certainly interesting, however I find myself at odds with the conclusions. Given that addressing these reservations would likely change the take-home messages of the paper, I therefore recommend major revisions.

Major comments:

* Unless I have misunderstood something, the core conclusions of this paper regarding coatings with ageing timescales seem to be based on the assumption that both urban and biomass burning BC are emitted with thin coatings. However, there is much evidence to the contrary, as most SP2 measurements of biomass burning at or near source would indicate that they have thick coatings at the point of emission. Furthermore, the thickness of this coating can vary significantly fire to fire (see https://www.atmos-chem-phys.net/14/10061/2014/, https://www.nature.com/articles/ngeo2901, https://www.atmos-chem-phys-discuss.net/acp-2019-157/). It therefore doesn't seem correct to infer conclusions regarding the effect of ageing timescales on coating thicknesses when comparing aerosols from different sources. The authors should review their findings taking this into consideration.

* The conclusions section is long but mainly seems to recap the earlier observations rather than focus on the key scientific advancements being offered by this work. In order to properly judge this aspect of the paper and therefore its suitability for publication, this should be restructured.

Minor comments:

* Measurements of coating thickness can become biased if the particles are sufficiently small that the signal-to-noise ratios of the instrument's scattering channels aren't sufficient to successfully retrieve a coating thickness or a delay time. Was the rate of failed retrievals monitored? How was this reflected in the data?

* Setting 'calm' winds as zero on direction on figure 5 makes no sense as this also corresponds to north. The periods should probably be blanked out instead.

[Figure]

* The points plotted on figure S9 should be individually identified according to event.

---

## Referee Comment (RC2) · Anonymous Referee #2 · 13 Jan 2020

The manuscript presents measurements of refractory black carbon (rBC) mass and number loadings along with derived rBC mixing state properties (e.g., fraction of thickly coated rBC particles and coating thickness) for three events (air masses) that were encountered at a sampling station located Catalina Island (CA). The measurement campaign itself can best be described as a measurement opportunity. Emission sources were identified using HYSPLIT backtrajectory analysis coupled with inputs from the GDAS (Global Data Assimilation System) and HRRR (High-resolution Rapid Refresh) meteorological databases. Analysis was further augmented with data from local weather stations, local news reports, NASA MODIS (Aqua/Terra) imagery, and the California Department of Forestry and Fire Protection for spatial and temporal ex-

tents of sampled wildfire plumes. Armed with these datastreams the authors report on the correlation of rBC mass/number loadings and derived mixing state properties with emission source and estimated plume age. Examples of findings include the wildfire named "Camp Fire" that was ascribed as being responsible for a sample plume made of up thickly-coated rBC particles that were nominally a week old and slightly thinner coated particles (i.e., 36 nm and less) were ascribed to urban Los Angeles emissions. The authors also attribute biomass burning as a contributing source to rBC particles that were found to have a coating thickness ~10 nm and were about a day old. While this work is highly-focused, it contains useful observations on rBC mass loading and mixing state that are of value to the community and thus should be published. However, this reviewer has profound reservations regarding three areas in this manuscript: (i) source attribution and estimated plume age; (ii) the research team's use of the rBC number size distribution data; (iii) and the discussions about increasing rBC diameter with atmospheric aging. Therefore, it is recommended that this manuscript undergo a major revision and be resubmitted.

Source attribution and plume age. With respect to source attribution, absence of a measurement of a biomass burning tracer (e.g., levoglucosan) have forced the authors to rely on HYSPLIT for emission source composition. For those flow patterns that are direct from source to measurement site, this approach is considered robust. However, for those trajectories that are more convoluted the robustness of this link degrades. Further, absent some measure of plume age (e.g., f44 fragment from an aerosol mass spectrum), the author's, again, rely on backtrajectories for plume age estimates. To be clear, the SP2 mixing state analysis is considered robust and the issue is with plume composition (source) and age. For example, this reviewer is astounded that the author's ascribe a biomass burn (BB) source to rBC coating thicknesses that are on the order of 10 nm and that the estimated plume is about a day old. To the author's credit, they seem to be aware of this when they acknowledge that their derived rBC coating thicknesses for a plume aged about a day is in contrast with several other published values (14 cited works). Their sole rebuttal reference to these 14 references is from

a paper by this same group authors. As highlighted in the 14 references and through direct experience BB -generated rBC becomes thickly coated (often well above 50 nm) very quickly (within the first couple of hours). This is due to the fact that wildfire plumes contain large OA (organic aerosol) mass loadings and are very rich in organic materials that can lead to SOA (secondary organic aerosol) production and condensable material onto an rBC particle. To say that BB plumes are contributing to the plume containing coating thicknesses that are more characteristic of urban plumes needs to be reassessed. Especially given that the preponderance of published data and new measurements (e.g., recently conducted WE-CAN and FIREx field campaigns) all show that BB events create very thickly coated particles very quickly. To be clear, this reviewer is not questioning the derived coating thicknesses, but rather the source attribution and estimated plume age. Therefore, the authors need to make a much more convincing case that the rBC particles sampled during L4 actually contain any biomass burning particles. As outlined above, this becomes a much harder argument to make due to the lack of compositional information that could quantify whether the L4 plume actually contained any wildfire emissions.

Staying with plume age, at the other end of the age spectrum, the authors argue that biomass burn plumes sampled during L3, and L8, L9, L10, are all nominally aged about a week. While the rBC particles will become thickly coated in the near field, as outlined above, coating volatility and subsequently photochemistry will likely cause the initially thickly coated rBC particles to lose material with age - through plume dilution and molecular fragmentation. This, again, gets at the robustness of the estimated plume age. As discussed above, for more direct flow patterns, the use of HYSPLIT is likely robust, but if the flow patterns are circuitous, and cross-over other potential emission sources, the possibility, indeed likelihood, of entrainment of other emission sources must be considered. If the authors feel that their ages are robust, then they should, more fully, discuss how to interpret their findings with the larger community of SP2-based mixing state literature. To this reviewer, this is one of the core findings of this manuscript and as such, needs to be addressed.

Finally, was any analysis conducted on whether the plumes sampled underwent any precipitation events that could have altered the microphysical make up of plume (e.g., washed out larger size, more thickly-coated particles)? [The authors are encouraged to read Taylor et al., (2014)Size-dependent wet removal of black carbon in Canadian biomass burning plumes ACP. 14 13755] on the size-dependent wet removal of rBC particles. Such wet removal could explain the lack of thickly coated rBC particles that were estimated to be about a day old (assuming the age estimate is robust).

rBC number size distribution. This reviewer has a major concern with applying a log-normal curve fit to the rBC number size distributions for which there is no obvious peak in the number size distribution data. The troubling application can be readily seen in figure 7 for L5 number size distribution where there is no peak in the number size distribution data, yet one is derived via curve fitting. This reviewer does not consider that L1 and L10 contain enough data to derive a robust lognormal fit. While not shown, based on Figure 8, presumably simpler fits were found for L2, L4, L7. The authors are reminded that the detection efficiency for the SP2 drops below unity for particles diameters 75 nm and smaller (authors are encouraged to read the SP2 detection efficiency paper by Schwarz et al., The Detection Efficiency of the Single Particle Soot Photometer AST, 44, 2010). Given that, it is not clear how meaningful the reported number size median rBC core diameters really are (see Figure 8). This reviewer questions the value of reporting fit-derived number size median core diameters that are less than 80 nm (which constitutes 6 out of the 10 reported values). This issue could very well help explain the observation reported by the authors (page 21, line 467- 468) that they "observe that changes in the mass median diameter are not consistent with the changes in number median diameter". Perhaps this is due to the detection efficiency limitation cited above. This reviewer suggests that any discussions that use or make reference to those datasets for which there is no discernible peak in the number size data, be removed or, at a minimum, discussed with an open acknowledgement of the detection efficiency issue and how this will impact interpetation. Also, please reword the sentence (page 21, 469 - 471) "our results suggest that the number median diameter

could be a more useful metric when correlating core diameters to mixing state metrics since the SP2 measures characteristics of individual rBC particles on a number basis and the CT_BC is calculated for each measured particle." to reflect the impact(s) of detection efficiency issues with respect to smaller diameter rBC particles on the utility of the number median diameter metric.

Sticking with Figure 8 for an additional moment, the first reported blue bar for L1 has an amplitude consistent with $\sim$ 75 nm but reports a value above the bar of 53 nm. Please correct. Also, the author's reference a Table 3 (page 22, line 483 and again in the supplemental page 1), but this reviewer is unable to locate referenced table.

Increasing rBC diameter with atmospheric aging. On page 21 lines 460 - 471, the authors argue that the rBC core diameter increases with atmospheric aging through coagulation. Certainly the rBC core diameter will increase with age at the source (e.g., a wildfire) where particle concentration are sufficiently high ($10^4$ - $10^5$ $cc^{-1}$) such that coagulation will occur on a time scale that is competitive with condensation. But as the plume dilutes, the kinetics of coagulation will decrease (rate goes as $N^2$, where N is the number concentration). The highest rBC number concentrations cited (Figure 5, page 14) are $\sim$ 400 $cc^{-1}$. Homogeneous coagulation under these conditions would be over 600x slower than at $10^4$ cc-1. So, with the drop in particle concentration that occurs with age, it is not clear that coagulation will result in a measurable increase in rBC diameter. Given this, how can the authors explain the reported increase in rBC diameter with age? The studies that the author's cite that reference a coagulation mechanism, are analyzing data under high number concentration conditions than the authors encounter in the current study.

Other specific issues:

Page 1, line 18, 20. The passive voice exemplified by the use of the word "suspect". Are the author's hedging their bets? Suggest using a different - less passive - word.

Page 1, lines 23-25. The author's write "we conclude that an aging timescale on the

order of ~hours is not long enough for rBC to become thickly coated under the range of sources sampled and atmospheric conditions during this campaign." This is misleading as several papers that have studied biomass burning (and those currently under review and data currently being analyzed) have (and are) showing that rBC particle become thickly coated very quickly. While this might be true for urban plumes, it certainly is not for BB (biomass burning) plumes. Please clarify.

Page 2, lines 43-44. The author's write "BC is emitted mostly as an "external" mixture, physically separated from other aerosol species." This is a bit misleading. It is very dependent upon when the plume is sampled. With respect to biomass burning, research has shown that rBC becomes coated within the first few minutes following generation - due to the chemical richness of the smoke plumes. Please reword to reflect this.

Page 3, line 74 and 75. The authors need to be very disciplined in their use of "mixing state", as one can be describing the aerosol mixing state (e.g., external vs internal) or the particle mixing state (e.g., coated or uncoated rBC). Yes, the authors sort of point this out on page 2 (lines 48-50) but then start interchanging "internal mixing state" with mixing state. For example, on the opening sentence of the cited paragraph, are the authors referring to the internal mixing state or the aerosol mixing state? Later in this paragraph, the authors reference internal mixing state of rBC (line 80). Please ensure consistency.

Page 3, lines 74 - 75. Here are two additional references to the use of microscopy with quantifying rBC mixing state that the authors are encouraged to consider: Adachi, K., Chung, S. H., and Buseck, P. R.: (2010) Shapes of soot aerosol particles and implications for their ef- fects on climate, J. Geophys. Res. Atmos., 115. Adachi, K., Moteki, N., Kondo, Y., and Igarashi, Y.: (2016) Mixing states of light-absorbing particles measured using a transmission electron microscope and a single-particle soot photometer in Tokyo, Japan, JGR.,121, 9153–9164.

Page 3, lines 80 - 83. Authors are encouraged to review (include) the work by Sedlacek

et al., who investigated the utility of the SP2 lagtime methodology [Investigation of Refractory Black Carbon-Containing Particle Morphologies Using the Single-Particle Soot Photometer (SP2) (2015) Aero. Sci. Tech., 49:872]

Page 3, line 83. The authors are encouraged to review (include) the work by Moteki and Kondo who have also contributed significantly to improving the quantification of the rBC mixing state [Method to measure time-dependent scattering cross sections of particles evaporating in a laser beam (2008) J. Aer. Sci. 39:348].

Page 9, lines 226 - 228. The authors might consider reviewing (including) the work by Sedlacek et al., who looked at the SP2 lagtime for a biomass burn plume. [Determination of and Evidence for Non-core-shell structure of particles containing black carbon using the single particle soot photometer (SP2). (2012) GRL. 39]

Page 10, Line 266. As highlighted earlier, please refrain from relying on a passive voice (e.g, "suspect".)

Page 12: The authors show the back trajectories for each day of the campaign. Why not put this figure in the supplemental and, instead, show those trajectories for the specific periods under discussion. This would make it easier to evaluate the HYSPLIT datasets.

Page 14 line 307. The authors reference Figure S9, but I think they mean S8?

Page 15, lines 337 - 344. The authors are encouraged to review paper by Subramanian et al., [(2010) Black carbon over Mexico: the effect of atmospheric transport on mixing state, mass absorption cross-section, and BC/CO ratios ACP 10] where attention is drawn specifically to figures 3, 12 and 13.

Page 22, lines 507 - 508. As highlighted above, this reviewer has concerns regarding the estimated plume ages.

Supplemental: page 1. As noted earlier, there is no table 3 in the main manuscript.

[Figure]

Supplemental: page 1, line 7. Suggest that the authors review Lund et al., [(2018) Short Black Carbon lifetime inferred from a global set of aircraft observations, npj Climate and Atmospheric Science 1, 31 doi:10.1038/s41612-018-0040-x]

Supplemental: page 7. This is a stylist comment. Would suggest using a different color to denote the sample location on Catalina Island. The currently used green color is hard to discern with the yellow star.

---

## Author Comment (AC2) · 1 May 2020

Author Response to RC2

We appreciate the thoughtful and detailed review from Referee 2. We have taken the comments made by Referee 2 into careful consideration and they have helped improve our manuscript.

The general format of this response is as follows:

- Reviewer comments are in bold and labeled as (N.1), where N is the number of the comment block.

[Figure]

- Author response to comments are in regular, non-bolded text, and labeled as (N.2).

- Modifications in the manuscript are described in italics and labeled as (N.3).

=====================================

(1.1)

**Major comments regarding source attribution and estimated plume age**

(1.2)

We agree with the reviewer that the source attribution and plume age sections of the manuscript required some major revisions (e.g., section 3.7, formerly section 3.6). We now shift our focus towards comparing the mixing state during different known source impacts, rather than focusing on the plume age. As the reviewer notes, rBC from biomass burning (BB) is coated much more quickly than rBC from urban emissions, and BB rBC has also been observed to have thicker coating overall compared to its urban counterpart.

We would like to address the nuances associated with the specific concerns that the reviewer raises in the comment.

Regarding biomass burning source attribution:

First, we wanted to clarify that we are not definitively attributing ~10 nm coating thickness values to fresh BB rBC particles, and we changed the language in the revised text to make this clear. ~10 nm was the median coating thickness from a population of aerosols that had a larger spread of individual coating thickness values. The coating thickness values on the higher end of the distribution tail (and outliers) are likely attributable to the BB impacts. We clarified in the new text that the peaks in the 2$^{nd}$ campaign (e.g., L4) were likely dominated by urban emissions, but that we could not exclude the likely impact of BB emissions mixing into the broader urban plume. In
fact, we still believe that biomass burning did impact our measurements to some degree, even if it was a minor fraction of total sampled rBC. In particular, the Thomas Fire was one of the largest fires in California history, and it was still active during the 2nd campaign (20-22 December 2017). With the center of the Thomas Fire less than 150 km away, and strong atypical Santa Ana winds recorded before and during the time of measurements, it is hard to imagine BB having no impact on the regional rBC loading at the time. In addition to geographic proximity and meteorology, the air quality monitoring stations in Santa Barbara, Ventura, and Los Angeles all recorded elevated concentrations of PM2.5 right around this time period. Additionally, as part of the new supplemental analysis, the HYSPLIT dispersion model was run to simulate the plume dispersion of multiple active fires during the December 2017 campaign. The HYSPLIT dispersion model shows the plumes from the Thomas Fire and several other smaller Southern California fires directly impacting the point of measurement (Catalina Island). These results are included in the revised Supplement. We also added a new qualitative analysis in the Supplement using CALIPSO lidar transects in the Southern California region during the 20-22 December period. From the CALIPSO transects we observed aerosols that were attributed to BB sources present just off the coast of Southern California around this time. This data is also shown in a new section in the Supplement.

Second, since the paper was first submitted, we have obtained levoglucosan data from November 2018 (3rd campaign) that were collected by colleagues at USC who were conducting an independent air quality study in the LA Basin (Soleimanian et al. 2020). Although the reviewer's comment was particularly focused on the L4 period, we would like to point out that the conditions during the 2nd campaign (December 2017) and the first portion of the 3rd campaign (November 2018) were quite similar. Geographically, there were multiple fires throughout the Southern California region in both campaigns (see Figure 3). Both campaigns were also characterized by Santa Ana (i.e., northerly and easterly) winds. The weekly average concentration of levoglucosan between 7 to 14 November and 15 to 22 November was 187.5 ng m$^{-3}$ and 83.89 ng m$^{-3}$, respectively. Note that the 3rd campaign took place between 12 and 18 November 2018. For

reference, levoglucosan concentrations during July 2018 (non-wildfire season) ranged between ~4 and 17 ng m$^{-3}$. The elevated concentration of levoglucosan inside the LA Basin during November 2018 removes any lingering doubt that BB aerosols were mixed into the broader regional air mass that was measured on Catalina Island. Given that similar fire and meteorological conditions were present during the 2$^{nd}$ campaign (December 2017), we have high confidence that BB also played at least a minor role in this campaign as well.

Regarding plume age comments:

For the LEO periods mentioned (L3, L8, L9, and L10), the aging timescale range of ~days to a week was meant to serve as a range of possibility rather than an exact aging timescale. We fully acknowledge the limits of HYSPLIT, especially for complex trajectory patterns. That is exactly why we present a very general range of timescales that was based on physical distance from major sources rather than relying on the exact timing of crossovers from the back-trajectories. The reviewer also mentions the loss of rBC coating with aging. This is entirely consistent with the $CT_{BC}$ values measured during periods impacted by long range transport of biomass burning impacted air masses (e.g., L8 and L9). The median $CT_{BC}$ values were within the range of ~60-70 nm during this time period of impact from the Camp Fire. Previous airborne studies have measured average coating thickness values of ~100 nm within hours of emission within the plume. Given that our values are significantly lower, the rBC measured in our study likely did experience coating loss at some point during transport. We added a short discussion on this topic of coating loss in the coating thickness discussion section and below in the section (1.3). Furthermore, we have added a new section that comprehensively compares our campaign measurements with past mixing state studies conducted with an SP2.

Regarding precipitation comments:

Although the data were not reported, precipitation and cloud cover were monitored

throughout the campaigns. There were no precipitation events in the region during any of the measurement days, and most of the days were clear to partly cloudy.

(1.3)

*Major edits were made to section 3.5 (formerly section 3.4) and section 3.7 (formerly section 3.6). A new section 3.8 was added to comprehensively compare our results to past similar studies. Additional evidence (i.e. using CALIPSO lidar data, HYSPLIT dispersion model and levoglucosan measurements) and figures were also added to the Supplement to make our discussion on source attribution more robust. Specifically, please refer to Supplement section S2 and figures S11 through S20.*

*Revised main points regarding variability of coating thickness:*

- *Timescales of less than 24 hours were too short to significantly coat rBC from urban emissions. This is in direct contrast to biomass burning rBC, which has been shown in previous works to acquire thick coatings within hours or even minutes, near the source of emission.*

- *Aged rBC from biomass burning sources were generally more thickly coated, although the time evolution of the mixing state could not be quantified directly over the duration of transport. Periods of "fresh" biomass burning impacts were characterized by slightly thinner $CT_{BC}$ compared to aged biomass burning rBC particles (e.g., L3 vs. L8), but larger $CT_{BC}$ compared to fresh urban rBC particle (e.g., L3 vs. L4). This agrees with previous studies that have also observed thicker coatings in fresh biomass burning rBC relative to fresh urban rBC. The overall larger $CT_{BC}$ for aged biomass burning rBC relative to fresh biomass burning rBC indicates that there is significant coating formation that occurs between the timescale of ~1 day to ~1 week for biomass burning rBC, even after rapid coating formation that occurs soon after emission. An important caveat is that $CT_{BC}$ of biomass burning rBC may not be monotonically increasing over time.*

*Past studies have observed rapid coating of biomass burning rBC within the first few hours to more than 100 nm, but we observed a median $CT_{BC}$ of 42 nm for L3, which suggests that $CT_{BC}$ for biomass burning rBC might decrease at some point during atmospheric transport and then increase later at longer timescales (e.g., median CTBC of 68.6 nm for L9). We make no definitive claims about the rate of change of $CT_{BC}$ for biomass burning rBC throughout atmospheric transport since we only observed the $CT_{BC}$ from a single discrete point in space, but our measurements do suggest that $CT_{BC}$ for Southern California biomass burning rBC were generally lower than $CT_{BC}$ for Northern California biomass burning rBC.*

=====================================

(2.1)

**Major comments regarding number size distribution data**

(2.2)

Although we generally acknowledge the concerns about fitting a log-normal distribution to a set of observations without a discernable peak, we also believe that the log-normal fits have value and should be reported (with associated uncertainty clearly described). First, there have been a number of past studies that have also included log-normal fits for their number size distributions, even in cases where the peak in the measured data was ambiguous. At the end of (2.2) is a comprehensive, but not exhaustive, list of studies using the SP2 that have included log-normal fits to rBC number size distributions. Full references are provided at the end of the document.

Second, the physical lower bound on BC core size makes log-normal fitting reasonable in the Aitken range, even if it is below the SP2 detection limit. Single BC nanospheres (i.e., individual spherules) have been observed to be ~20-30 nm in diameter by using TEM imaging techniques (Ellis et al., 2016; Wentzel et al., 2003). Although the detection limit of the SP2 for rBC cores is ~70 nm, it seems reasonable to assume that the peak of the rBC number size distribution in this Aitken range would be between 50 and 80 nm (Kondo et al., 2011b), given that individual BC spherules are unlikely to be smaller than 20 nm. This would naturally imply that most (if not all) BC cores in the ambient air are larger than 20 nm, but smaller than the point at which we observe a sharp increase in the slope on the right-hand side of the number size distribution. This inflection point on the right-hand side is clearly observed from SP2 data, even when the peak is not completely discernable.

Third, even if there was an unmeasured bimodal peak beyond the detection limit of the SP2, the median of the extrapolated log-normal fit would not be a completely useless metric for comparison. As long as the log-normal fitting is consistent between all instances of distributions, it would serve to characterize the Aitken mode of the rBC core size distribution, even if there was another mode lurking in the ultrafine range. This would suggest that the existence of an unknown local maxima in the ultra-fine range is possible, but that it would not invalidate the inter-comparison of Aitken mode distributions for different time intervals.

Fourth and lastly, the appropriateness of the log-normal fit is not entirely contingent upon the explicit observation of a local maxima. It might be entirely inappropriate if we saw that all the observed data points deviated sporadically from the fit curve, but we observe the fit curve describing the observed number size distribution data points very well, with fairly small residuals. We see that the rate of change of the slope is well captured by the fit, which strongly suggests that a log-normal fit is likely representative of the actual distribution. Analogously, we find the LEO-fit for coating thickness quantification as a robust method for mixing state analysis, even though we only use the leading edge of what we expect to be a Gaussian signal. Indeed, the LEO-fit uses an even smaller fraction of the expected Gaussian scattering response compared to the log-normal fits for the number size distribution. Likewise, we are using the existing "edge" of size distribution to fit what we expect to be log-normal.

[Figure]

To address the reviewer's concern with this issue, we made a clear caveat in the text explaining the limitations of the extrapolation, in addition to the already existing disclaimer about the lower detection limit in the first paragraph of section 3.6 (formerly section 3.5). We made it clear and explicit that the peak based on log-normal fits are not definitive measured values, but rather modeled based on reasonable assumptions about the behavior of the distribution in the Aitken range.

The typo in Figure 8 regarding the wrong median value label has also been fixed.

List of publications that have used log-normal fits to the number size distribution data:

Cheng et al., 2018; Kondo et al., 2011a; Kondo, et al., 2011b; Krasowsky et al., 2018; Metcalf et al., 2012; Moteki et al., 2012; Raatikainen et al., 2017; Reddington et al., 2013; Sahu et al., 2012; Schwarz et al., 2008; Shiraiwa et al., 2008

(2.3)

*An additional caveat has been added to section 3.5 (formerly section 3.5) in the manuscript in tracked changes to address the comments and concerns made by RC2.*

*"Figure 10 shows that log-normal fits adequately capture the measured size distributions, though we cannot rule out the possibility of another rBC mode outside the detection limits of the SP2. Although the peak of the observed points is not always discernible (e.g., number size distribution for L5 in Figure 10), it is reasonable to fit these points assuming that a log-normal distribution is a realistic representation of ambient rBC number size distributions in the Aitken mode. The rate of change of the observed points is also captured very well qualitatively by the log-normal fits, further indicating its appropriateness."*

=====================================

(3.1)

**Major comments regarding increasing rBC diameter with atmospheric aging**

(3.2)

We agree with the reviewer that the effect of coagulation on the rBC core size is likely overplayed in the manuscript since the rBC number concentration is relatively low in the ambient air at the point of measurement, compared to the rBC number concentration very close to the source of combustion (e.g., in a tailpipe or in a BB flame). We would like to point out, however, that there is a noticeable shift in the rBC size distributions during time periods dominated by urban emissions (e.g., L4 and L5) relative to size distributions that were measured inside Los Angeles near a major highway by Krasowsky et al. (2018). This is a particularly useful comparison because the exact same SP2 was used with the same operating variables. Focusing on the number size distribution, we observe a larger count median diameter during the L4 and L5 periods compared to the count median diameters measured downwind of a highway in a polluted urban environment (Krasowsky et al., 2018). The size distribution of rBC can only be affected by, (i) the emission source type and/or (ii) coagulation of rBC-containing particles. Related to the reviewer's comment regarding source attribution, we believe that both of these factors likely played some role in the variability of rBC core sizes. We are quite confident that BB sources did contribute, at least in part, to time periods dominated by urban emissions. (see comment block 1 above for details). So, there is likely a source effect. It seems plausible that a mixture of BB impact and coagulation (at least near the source, within the polluted urban basin), contributed to this noticeable shift in the core size distribution.

The reviewer also notes that the cited studies were conducted under higher rBC concentrations than what we encountered in our study. However, while the studies mentioned did have higher campaign-averaged concentrations, the peak concentrations were within the same magnitude, especially for the Shiraiwa et al. study (2008), which took place in the East Asia outflow. The peak magnitudes reported in Shiraiwa et al. reached ~ 1 $\mu$g m$^{-3}$, which is within a factor of two relative to the larger peaks measured in our study (~0.6 $\mu$g m$^{-3}$). Shiraiwa et al. (2008) briefly mention that coagulation could

be a potential mechanism that explains why aged particles from China and Korea were larger than particles associated local urban emissions from Japan. While we agree that coagulation at measured concentrations would be slow and possibly negligible, we believe that coagulation could have played a minor role during atmospheric transport from the LA basin to Catalina Island. We make no attempt at quantifying the rate at which coagulation occurs for LA basin dominated air masses, but we qualitatively acknowledge that coagulation likely contributed to the growth of particles, as per the logic above, especially within the first few hours of aging.

(3.3)

*The focus of the paragraph mentioned by the reviewer has been shifted towards an emphasis on source-related impacts rather than impacts from atmospheric processing (i.e., coagulation). A short mention of coagulation still remains, but it serves as a qualitative acknowledgement of its likely minor effect on rBC size distributions. See section 3.6 (formerly section 3.5) for tracked changes.*

*Relevant excerpts from new text in section 3.6:*

*"A survey of past studies that have reported rBC mass median diameter (MMD) and count median diameter (CMD) shows that the source of emissions has a strong influence on rBC core diameter (Cheng et al., 2018). The MMD [CMD] for biomass burning influenced rBC, which has been reported to range from ~130 nm to 210 nm [100 to 140 nm], is generally much larger than the MMD for urban emissions influenced rBC, which has been reported to range from ~100 nm to 178 nm [38 to 80 nm] (Shiraiwa et al., 2007; Schwarz et al., 2008; McMeeking et al. 2010; Kondo et al., 2011a; Sahu et al. 2012; Metcalf et al., 2012; Cappa et al., 2012; Laborde et al., 2013; Liu et al., 2014; Taylor et al., 2014; Krasowsky et al., 2018). The MMD [CMD] for aged air masses in remote regions were reported to range from ~180 nm to 225 nm [90 nm to 120 nm] (Shiraiwa et al., 2008; Liu et al, 2010; McMeeking et al., 2010; Schwarz et al., 2010).*

*Figure 11 shows the rBC MMD and CMD based on the log-normal fits for each LEO*

*period in this study. Based on the source identification discussed in section 3.1 and section S2 in the Supplement, the MMD and CMD values in this study are generally consistent with the ranges reported in past studies. For LEO periods when measurements were strongly influenced by biomass burning emissions (L3, L8, L9, L10), MMD ranged from 149 nm to 171 nm, which is within the range of ~130 nm to 210 nm compiled from past studies. Similarly, when measurements were strongly influenced by urban emissions (L2, L4, L7), the MMD dropped, ranging from 112 nm to 129 nm. This falls within the range of ~100 nm to 178 nm previously reported for measurements of urban emissions from past studies."*

*"In addition to varying source type, coagulation is the only physical mechanism that increases rBC core size (Bond et al., 2013). Shiraiwa et al. (2008) observed an increase in rBC core diameters in aged plumes compared to more fresh urban plumes, suggesting that coagulation can alter the rBC size distribution during atmospheric transport (i.e., aging). Although the emissions source type appears to be the dominant influence on rBC core sizes in this study, there is evidence to suggest that coagulation did occur during transport from the Los Angeles basin to Catalina Island (~70 km away) in this study. For example, we observed an MMD [CMD] of 112 nm [53 nm] during L4, when we know that urban emissions were dominant, but this is noticeably larger than values of 93 nm [42 nm] reported in Krasowsky et al. (2018) for measurements conducted 114 meters downwind of a major highway. Furthermore, Laborde et al. (2013) observed an MMD of ~100 nm when impacted by fresh traffic emissions in Paris. Even though it was determined that L4 was predominantly urban emissions influenced, we cannot rule out the possibility of local wildfires influencing the size distribution as well. While the rBC size distribution from L4 suggests that coagulation plays at least a minor role, both factors (source type and coagulation) likely influence rBC size distributions to varying degrees in areas with heterogenous source profiles and relatively elevated rBC concentrations (e.g., polluted urban areas)."*

====================================

(4.1)

**Page 1, line 18, 20. The passive voice exemplified by the use of the word "suspect". Are the author's hedging their bets? Suggest using a different - less passive - word.**

(4.2)

The wording has been changed.

(4.3)

*New text:*

*"In contrast, during periods when measured rBC was dominated by emissions from the Southern California region, both fBCand $CT_{BC}$ were significantly lower, with a mean fBC of ~0.03 and median $CT_{BC}$ ranging from ~0 to 10 nm."*

=====================================

(5.1)

**Page 1, lines 23-25. The author's write "we conclude that an aging timescale on the order of ~hours is not long enough for rBC to become thickly coated under the range of sources sampled and atmospheric conditions during this campaign." This is misleading as several papers that have studied biomass burning (and those currently under review and data currently being analyzed) have (and are) showing that rBC particle become thickly coated very quickly. While this might be true for urban plumes, it certainly is not for BB (biomass burning) plumes. Please clarify.**

(5.2)

We agree with the reviewer and we have changed the main conclusions of our paper to reflect this. Further response to this specific issue has been discussed in more detail

above in comment block 1.

(5.3)

*Any text related to the generalization of thin coatings for particles aged less than 24 hours has either been removed or modified.*

*This was also discussed in greater detail in comment block 1 and applicable changes have been made in sections 3.5 and 3.7 (formerly sections 3.4 and 3.6).*

=====================================

(6.1)

**Page 2, lines 43-44. The author's write "BC is emitted mostly as an "external" mixture, physically separated from other aerosol species." This is a bit misleading. It is very dependent upon when the plume is sampled. With respect to biomass burning, research has shown that rBC becomes coated within the first few minutes following generation due to the chemical richness of the smoke plumes.**

**Please reword to reflect this.**

(6.2)

We acknowledge that BC can become coated very quickly and that this statement could potentially be misleading. The original intent was to give a conceptual overview of externally versus internally mixed BC. The description has been altered to remove any ambiguities regarding emission point and timescale since emission.

(6.3)

*The text in the introduction (section 1) has been altered to describe the two general types of mixing state without potentially misleading readers into believing that all BC is uncoated in the near-field plume.*

*"A hypothetical BC particle that is completely, physically separate from other non-BC aerosol species is considered externally mixed. On the other hand, BC is considered internally mixed if it is physically combined with another non-BC aerosol species (Bond et al., 2006; Schwarz et al., 2008). As freshly emitted BC particles are transported in the atmosphere, they can obtain inorganic and organic coatings from either gaseous pollutants that condense onto the BC, oxidation reactions on the BC surface, or the coalescence of other aerosol species onto the BC, making them more internally mixed (He et al., 2015). In short, externally mixed BC is referred to as "uncoated BC" and internally mixed BC is referred to as "coated BC." In general, the mixing state of BC describes the degree to which BC is internally mixed, with uncoated (i.e., externally mixed) BC particles on one end of the mixing state spectrum (Bond et al., 2013). The BC mixing state near the point of emission as well as the evolution of the mixing state can vary widely, depending on the source of emissions and atmospheric context."*

=====================================

(7.1)

**Page 3, line 74 and 75. The authors need to be very disciplined in their use of "mixing state", as one can be describing the aerosol mixing state (e.g., external vs internal) or the particle mixing state (e.g., coated or uncoated rBC). Yes, the authors sort of point this out on page 2 (lines 48-50) but then start interchanging "internal mixing state" with mixing state. For example, on the opening sentence of the cited paragraph, are the authors referring to the internal mixing state or the aerosol mixing state? Later in this paragraph, the authors reference internal mixing state of rBC (line 80). Please ensure consistency.**

(7.2) We acknowledge this potential for confusion and changed the language throughout the manuscript to ensure consistency. For the sake of simplicity and consistency, we initially define externally mixed BC as "uncoated BC" and internally mixed BC as "coated BC." Furthermore, we use the general term "mixing state," to refer to the extent

to which BC is coated, either at an individual particle level or aggregated (i.e., sample population-wide) level.

(7.3)

*We edited the text to ensure consistency between any language describing the mixing state. This topic was also discussed in comment block 6 above.*

=====================================

(8.1)

**Page 3, lines 74 - 75. Here are two additional references to the use of microscopy with quantifying rBC mixing state that the authors are encouraged to consider: Adachi, K., Chung, S. H., and Buseck, P. R.: (2010) Shapes of soot aerosol particles and= implications for their effects on climate, J. Geophys. Res. Atmos., 115. Adachi, K., Moteki, N., Kondo, Y., and Igarashi, Y.: (2016) Mixing states of light-absorbing particles measured using a transmission electron microscope and a single-particle soot photometer in Tokyo, Japan, JGR.,121, 9153–9164.**

(8.2)

Thank you for the references and suggestion. They have been added to the manuscript.

(8.3)

*These references have been added to the introduction of the manuscript where microscopy is briefly mentioned.*

=====================================

(9.1)

**Page 3, lines 80 - 83. Authors are encouraged to review (include) the work by Sedlacek et al., who investigated the utility of the SP2 lagtime methodology [In-**

**vestigation of Refractory Black Carbon-Containing Particle Morphologies Using the Single-Particle Soot Photometer (SP2) (2015) Aero. Sci. Tech., 49:872]**

(9.2)

Thank you for the suggestion. We have incorporated this reference into our study and expanded on our analysis by including discussion about negative la-times and rBC morphology in the discussion section. See also (11.1) below, which is related to this comment.

(9.3)

*See section 3.4 on negative lag-times and rBC morphology for newly inserted analysis and discussion.*

*Excerpt from new text:*

*"In this study, we observed negative lag-times, although at a relatively low rate, with flag,neg calculated to be much less than 0.1 throughout most of the measurement periods. We defined flag,neg to be identical to the "fraction of near surface rBC particles" metric used by Sedlacek et al. (2012), using a lag-time threshold of -1.25 $\mu s$ to account for uncertainties associated with the lag-time determination. The campaign-wide flag,neg was 0.017 for the first campaign (September 2017), 0.018 for the second campaign (December 2017), and 0.026 for the third campaign (November 2018). Comparatively, Dahlkötter et al. (2014) observed flag,neg of ~0.046 during an airborne field campaign measuring an aged biomass burning plume in Germany, and a much higher disintegration rate of ~0.4 to 0.5, based on a method that examines the tail end of the time-dependent scattering cross-section (Laborde et al., 2012). Sedlacek et al. (2012) reported flag,neg of more than 0.6 for ground-based measurements of a biomass burning plume in Long Island, New York, originating from Lake Winnipeg, Canada; and the scattering-cross section method was not used to calculate an additional disintegration rate."*

========================================

(10.1)

**Page 3, line 83. The authors are encouraged to review (include) the work by Moteki and Kondo who have also contributed significantly to improving the quantification of the rBC mixing state [Method to measure time-dependent scattering cross sections of particles evaporating in a laser beam (2008) J. Aer. Sci. 39:348].**

(10.2)

Thank you for the suggestion. This study was not initially included in the manuscript because the method described in Moteki and Kondo (2008) was not used for our mixing state analysis. Nevertheless, we have added the reference in the initial description of the LEO method because of its relevance to the Gao et al. (2007) method, which we used in our study.

(10.3)

*The reference has been added to section 2.7 in the manuscript.*

========================================

(11.1)

**Page 9, lines 226 - 228. The authors might consider reviewing (including) the work by Sedlacek et al., who looked at the SP2 lagtime for a biomass burn plume. [Determination of and Evidence for Non-core-shell structure of particles containing black carbon using the single particle soot photometer (SP2). (2012) GRL. 39]**

(11.2)

Thank you for the suggested work. We have added an additional short section about the morphology of rBC in the results and discussion section of the manuscript, and we use the same near-surface fraction analysis that Sedlacek et al. (2012) employed in

their study. The reference has been added as well.

(11.3)

*See section 3.4 on negative lag-times and rBC morphology for newly inserted analysis.
Also see comment block 9 above for related discussion.*

=====================================

(12.1)

**Page 10, Line 266. As highlighted earlier, please refrain from relying on a passive
voice (e.g, "suspect".)**

(12.2)

Passive voice removed.

(12.3)

*The word "suspect" has been removed from referenced text.*

=====================================

(13.1)

**Page 12: The authors show the back trajectories for each day of the campaign.
Why not put this figure in the supplemental and, instead, show those trajectories
for the specific periods under discussion. This would make it easier to evaluate
the HYSPLIT datasets.**

(13.2)

Thank you for the suggestion. Although we see the value in the suggestion, we prefer to
leave Figure 3 in its current state and add a*separate* HYSPLIT figure either in Section
3.7 or in the Supplement. Our reason for showing all the trajectories in Figure 3 is to
show the campaign-wide perspective on the source locations of the particles. We also

thought it would be useful for visually comparing between the different campaigns, and not just for 10 to 15-minute LEO time periods, which give limited snapshots instead of showing a broader campaign-wide "fingerprint" of trajectories.

(13.3)

*Additional figure with only LEO period back-trajectories has been added to the Supplement. This can also be added to Section 3.7 if it is determined to be more appropriate there.*

====================================

(14.1)

**Page 14 line 307. The authors reference Figure S9, but I think they mean S8?**

(14.2)

Thank you for catching this typo.

(14.3)

*Changed from Figure S9 to Figure S8.*

====================================

(15.1)

**Page 15, lines 337 - 344. The authors are encouraged to review paper by Subramanian et al., [(2010) Black carbon over Mexico: the effect of atmospheric transport on mixing state, mass absorption cross-section, and BC/CO ratios ACP 10] where attention is drawn specifically to figures 3, 12 and 13.**

(15.2)

Thank you for the paper suggestion. The figures you suggested were carefully reviewed and they were helpful in putting our results in context of past studies like Subramanian et al. (2010). Brief comparisons are made to the results presented in Sub-ramanian et al. (2010) to our results. Reference to the article has also been added to the manuscript.

(15.3)

*See minor additions in Section 3.3 and Section 3.7.*

======================================

(16.1)

**Page 22, lines 507 - 508. As highlighted above, this reviewer has concerns regarding the estimated plume ages.**

(16.2)

Appropriate changes have been made to the main conclusions from this paper, as described in more detail in Comment 1.1 above. Most importantly, all blanket statements regarding an aging timescale of more than one day required for thick coating have been altered or removed.

(16.3)

*See revised manuscript for tracked changes.*

======================================

(17.1)

**Supplemental: page 1. As noted earlier, there is no table 3 in the main manuscript.**

(17.2)

The table was accidently omitted. Apologies for any confusion.

(17.3)

*Table 3 has been merged with Table 2. The old Table 3 is now part of Table 2.*

====================================

(18.1)

**Supplemental: page 1, line 7. Suggest that the authors review Lund et al., [(2018) Short Black Carbon lifetime inferred from a global set of aircraft observations, npj Climate and Atmospheric Science 1, 31 doi:10.1038/s41612-018-0040-x]**

(18.2)

Thank you for the suggested article. The mean BC lifetime of ~4 days over the Pacific as suggested by Lund et al. (2018) further supports our estimated range of source-to-receptor timescales. We would like to clarify here that our loosely restrained timescales are only meant to give readers an idea of the range of possibilities regarding how long measured particles were transported in the atmosphere. Since the estimated value of ~4 days was meant to represent the mean, individual particles measured during our campaigns could certainly have been aged longer (i.e. ~week).

(18.3)

*Citation added to text.*

====================================

(19.1)

**Supplemental: page 7. This is a stylist comment. Would suggest using a different color to denote the sample location on Catalina Island. The currently used green color is hard to discern with the yellow star.**

(19.2)

Style change made as suggested.

(19.3)

*Green circle removed from the figure as it was unnecessary.*

=====================================

References

Cappa, C. D., Onasch, T. B., Massoli, P., Worsnop, D. R., Bates, T. S., Cross, E. S., Davidovits, P., Hakala, J., Hayden, K. L., Jobson, B. T., Kolesar, K. R., Lack, D. A., Lerner, B. M., Li, S. M., Mellon, D., Nuaaman, I., Olfert, J. S., Petäjä, T., Quinn, P. K., Song, C., Subramanian, R., Williams, E. J. and Zaveri, R. A.: Radiative absorption enhancements due to the mixing state of atmospheric black carbon, Science (80-. )., 337(6098), 1078–1081, doi:10.1126/science.1223447, 2012.

Cheng, Y., Li, S. M., Gordon, M. and Liu, P.: Size distribution and coating thickness of black carbon from the Canadian oil sands operations, Atmos. Chem. Phys., 18(4), 2653–2667, doi:10.5194/acp-18-2653-2018, 2018.

Ellis, A., Edwards, R., Saunders, M., Chakrabarty, R. K., Subramanian, R., Timms, N. E., van Riessen, A., Smith, A. M., Lambrinidis, D., Nunes, L. J., Vallelonga, P., Goodwin, I. D., Moy, A. D., Curran, M. A. J. and van Ommen, T. D.: Individual particle morphology, coatings, and impurities of black carbon aerosols in Antarctic ice and tropical rainfall, Geophys. Res. Lett., 43(22), 11,875-11,883, doi:10.1002/2016GL071042, 2016.

Kondo, Y., Matsui, H., Moteki, N., Sahu, L., Takegawa, N., Kajino, M., Zhao, Y., Cubison, M. J., Jimenez, J. L., Vay, S., Diskin, G. S., Anderson, B., Wisthaler, A., Mikoviny, T., Fuelberg, H. E., Blake, D. R., Huey, G., Weinheimer, A. J., Knapp, D. J. and Brune, W. H.: Emissions of black carbon, organic, and inorganic aerosols from biomass burning in North America and Asia in 2008, J. Geophys. Res. Atmos., 116(8), 1–25, doi:10.1029/2010JD015152, 2011a.

Kondo, Y., Sahu, L., Moteki, N., Khan, F., Takegawa, N., Liu, X., Koike, M. and Miyakawa, T.: Consistency and traceability of black carbon measurements made by laser-induced incandescence, thermal-optical transmittance, and filter-based photo-absorption techniques, Aerosol Sci. Technol., 45(2), 295–312, doi:10.1080/02786826.2010.533215, 2011b.

Krasowsky, T. S., Mcmeeking, G. R., Sioutas, C. and Ban-Weiss, G.: Characterizing the evolution of physical properties and mixing state of black carbon particles: From near a major highway to the broader urban plume in Los Angeles, Atmos. Chem. Phys., 18(16), 11991–12010, doi:10.5194/acp-18-11991-2018, 2018.

Liu, D., Flynn, M., Gysel, M., Targino, A., Crawford, I., Bower, K., Choularton, T., Jurányi, Z., Steinbacher, M., Huglin, C., Curtius, J., Kampus, M., Petzold, A.,Weingartner, E., Baltensperger,

U., and Coe, H.: Single particle characterization of black carbon aerosols at a tropospheric alpine site in Switzerland, Atmos. Chem. Phys., 10, 7389–7407, https://doi.org/10.5194/acp-10-7389-2010, 2010.

Liu, D., Allan, J. D., Young, D. E., Coe, H., Beddows, D., Fleming, Z. L., Flynn, M. J., Gallagher, M. W., Harrison, R. M., Lee, J., Prevot, A. S. H., Taylor, J. W., Yin, J., Williams, P. I. and Zotter, P.: Size distribution, mixing state and source apportionment of black carbon aerosol in London during winter time, Atmos. Chem. Phys., 14(18), 10061–10084, doi:10.5194/acp-14-10061-2014, 2014.

Metcalf, A. R., Craven, J. S., Ensberg, J. J., Brioude, J., Angevine, W., Sorooshian, A., Duong, H. T., Jonsson, H. H., Flagan, R. C. and Seinfeld, J. H.: Black carbon aerosol over the Los Angeles Basin during CalNex, J. Geophys. Res. Atmos., 117(8), 1–24, doi:10.1029/2011JD017255, 2012.

McMeeking, G. R., Hamburger, T., Liu, D., Flynn, M., Morgan, W. T., Northway, M., Highwood, E. J., Krejci, R., Allan, J. D., Minikin, A. and Coe, H.: Black carbon measurements in the boundary layer over western and northern Europe, Atmos. Chem. Phys., 10(19), 9393–9414, doi:10.5194/acp-10-9393-2010, 2010.

Moteki, N., Kondo, Y., Oshima, N., Takegawa, N., Koike, M., Kita, K., Matsui, H. and Kajino, M.: Size dependence of wet removal of black carbon aerosols during transport from the boundary layer to the free troposphere, Geophys. Res. Lett., 39(13), 2–5, doi:10.1029/2012GL052034, 2012.

Raatikainen, T., Brus, D., Hooda, R. K., Hyvarinen, A. P., Asmi, E., Sharma, V. P., Arola, A. and Lihavainen, H.: Size-selected black carbon mass distributions and mixing state in polluted and clean environments of northern India, Atmos. Chem. Phys., 17(1), 371–383, doi:10.5194/acp-17-371-2017, 2017.

Reddington, C. L., McMeeking, G., Mann, G. W., Coe, H., Frontoso, M. G., Liu, D., Flynn, M., Spracklen, D. V. and Carslaw, K. S.: The mass and number size distributions of black carbon aerosol over Europe, Atmos. Chem. Phys., 13(9), 4917–4939, doi:10.5194/acp-13-4917-2013, 2013.

Sahu, L. K., Kondo, Y., Moteki, N., Takegawa, N., Zhao, Y., Cubison, M. J., Jimenez, J. L., Vay, S., Diskin, G. S., Wisthaler, A., Mikoviny, T., Huey, L. G., Weinheimer, A. J. and Knapp, D. J.: Emission characteristics of black carbon in anthropogenic and biomass burning plumes over California during ARCTAS-CARB 2008, J. Geophys. Res. Atmos., 117(16), 1–20, doi:10.1029/2011JD017401, 2012.

Schwarz, J. P., Gao, R. S., Spackman, J. R., Watts, L. A., Thomson, D. S., Fahey, D. W., Ryerson, T. B., Peischl, J., Holloway, J. S., Trainer, M., Frost, G. J., Baynard, T., Lack, D. A., de Gouw, J. A., Warneke, C. and Del Negro, L. A.: Measurement of the mixing state, mass, and optical size of individual black carbon particles in urban and biomass burning emissions, Geophys. Res. Lett., 35(13), 1–5, doi:10.1029/2008GL033968, 2008.

Shiraiwa, M., Kondo, Y., Moteki, N., Takegawa, N., Sahu, L. K., Takami, A., Hatakeyama, S., Yonemura, S. and Blake, D. R.: Radiative impact of mixing state of black carbon aerosol in Asian outflow, J. Geophys. Res. Atmos., 113(24), 1–13, doi:10.1029/2008JD010546, 2008.

Soleimanian, E., Mousavi, A., Taghvaee, S., Shafer, M. M. and Sioutas, C.: Impact of secondary and primary particulate matter (PM) sources on the enhanced light absorption by brown carbon (BrC) particles in central Los Angeles, Sci. Total Environ., 705, 135902, doi:10.1016/j.scitotenv.2019.135902, 2020.

Subramanian, R., Kok, G. L., Baumgardner, D., Clarke, A., Shinozuka, Y., Campos, T. L., Heizer, C. G., Stephens, B. B., De Foy, B., Voss, P. B. and Zaveri, R. A.: Black carbon over Mexico: The effect of atmospheric transport on mixing state, mass absorption cross-section, and BC/CO ratios, Atmos. Chem. Phys., 10(1), 219–237, doi:10.5194/acp-10-219-2010, 2010.

Taylor, J. W., Allan, J. D., Allen, G., Coe, H., Williams, P. I., Flynn, M. J., Le Breton, M., Muller, J. B. A., Percival, C. J., Oram, D., Forster, G., Lee, J. D., Rickard, A. R., Parrington, M., and Palmer, P. I.: Size-dependent wet removal of black carbon in Canadian biomass burning plumes, Atmos. Chem. Phys., 14, 13755–13771, https://doi.org/10.5194/acp-14-13755-2014, 2014.

Wentzel, M., Gorzawski, H., Naumann, K. H., Saathoff, H. and Weinbruch, S.: Transmission electron microscopical and aerosol dynamical characterization of soot aerosols, J. Aerosol Sci., 34(10), 1347–1370, doi:10.1016/S0021-8502(03)00360-4, 2003.

---

## Author Response (AR1)

This document is organized as follows:

    (1) Author Response to RC1

    (2) Author Response to RC2

    (3) Marked-up version of manuscript

*Note: The author responses below are the updated versions of the responses posted in the ACP Interactive Discussion, under AC1 and AC2, on 1 May 2020.

**Author Response to RC1**

We appreciate the thoughtful and detailed review from Referee 1. We have taken the comments made by Referee 1 into
careful consideration and they have helped improve our manuscript.

The general format of this response is as follows:

- Reviewer comments are in bold and labeled as (N.1), where N is the number of the comment block.
- Author response to comments are in regular, non-bolded text, and labeled as (N.2).
- Modifications in the manuscript are described in italics and labeled as (N.3).

==================================

(1.1)

**Unless I have misunderstood something, the core conclusions of this paper regarding coatings with ageing timescales**
**seem to be based on the assumption that both urban and biomass burning BC are emitted with thin coatings.**
**However, there is much evidence to the contrary, as most SP2 measurements of biomass burning at or near source**
**would indicate that they have thick coatings at the point of emission. Furthermore, the thickness of this coating can**
**vary significantly fire to fire (see https://www.atmos-chem-phys.net/14/10061/2014/**
**https://www.nature.com/articles/ngeo2901, https://www.atmos-chem-physdiscuss. net/acp-2019-157/). It therefore**
**doesn't seem correct to infer conclusions regarding the effect of ageing timescales on coating thicknesses when**
**comparing aerosols from different sources. The authors should review their findings taking this into consideration.**

(1.2)

We agree that that it was presumptive and largely erroneous to make blanket statements about coating thickness without
properly taking the emission source(s) into account. As the reviewer noted, we agree that the existing literature shows
overwhelming evidence that biomass burning rBC particles are quickly coated after being emitted, and that they are, on average, significantly more coated than urban rBC particles of comparable atmospheric age. We made major revisions to section 3.7 (formerly section 3.6) and shifted the focus away from the aging timescale. The focus is now on the differences in mixing state during different identifiable source impacts (e.g., biomass burning versus urban). We left some discussion in the manuscript about the effects of aging on mixing state, but we made sure to keep comparisons consistent between the same source or mix of sources.

(1.3)

*See section 3.7 (formerly section 3.6) for major revisions shown in tracked changes. Please also see the Supplement for additional information on source attribution and also new additional analysis including measurements of levoglucosan (i.e., a good tracer for biomass burning) and lidar data from CALIPSO.*

==================================

(2.1)

**The conclusions section is long but mainly seems to recap the earlier observations rather than focus on the key scientific advancements being offered by this work. In order to properly judge this aspect of the paper and therefore its suitability for publication, this should be restructured.**

(2.2)

We agree that the conclusion should be restructured and focused on the most salient "key scientific advancements" rather than merely "recapping earlier observations." That being said, we also believe that recapping key observations and details in the conclusion section may be valuable to readers who might be quickly reading through the paper, trying to glean the most important take-away points from the abstract and/or conclusion.

Regarding significant changes to the conclusion, we have added clarification to the key scientific advancements. These key points have been slightly modified in light of comment (1.1) and additional comments from Referee 2. In addition, some extraneous details have been stripped from the conclusion as suggested. The main conclusion points of the manuscript are summarized below.

(2.3)

*Conclusion Point 1: The rBC size distribution was strongly affected by the emission source type. rBC particles measured during periods when biomass burning emissions were dominant ($BC_{bb}$) had larger rBC core diameters (CMD ~ 120 nm) relative to rBC particles measured during time periods dominated by urban emissions ($BC_{ff}$) (CMD ~ 100 nm). rBC particles from well-aged, background air masses ($BC_{aged,bg}$) were characterized by an MMD ~ 115 nm, which likely reflects a mix of large-scale transported BC from unidentified biomass burning and urban emissions.*

*Conclusion Point 2: $BC_{ff}$ were found to be either uncoated or very thinly coated, with mean $CT_{BC}$ less than ~15 nm and average $f_{BC}$ less than ~0.15.*

*Conclusion Point 3: $BC_{bb}$ had thicker coatings overall, with mean $CT_{BC}$ ranging from ~40 to 70 nm and $f_{BC}$ ranging from ~0.23 to 0.47.*

*Conclusion Point 4: $BC_{aged,bg}$ were found to have moderately thick coatings, with mean $CT_{BC}$ of ~60 nm and $f_{BC}$ of ~0.27.*

*Conclusion Point 5: Timescales of less than 24 hours were too short to significantly coat rBC from urban emissions. This is in direct contrast to biomass burning rBC, which has been shown in previous works to acquire thick coatings within hours or even minutes, near the source of emission.*

*Conclusion Point 6: Aged rBC from biomass burning sources were generally more thickly coated, although the time*
*evolution of the mixing state could not be quantified directly over the duration of transport. Periods of "fresh" biomass burning impacts were characterized by slightly thinner $CT_{BC}$ compared to aged biomass burning rBC particles (e.g., L3 vs. L8), but larger $CT_{BC}$ compared to fresh urban rBC particle (e.g., L3 vs. L4). This agrees with previous studies that have also observed thicker coatings in fresh biomass burning rBC relative to fresh urban rBC. The overall larger $CT_{BC}$ for aged biomass burning rBC relative to fresh biomass burning rBC indicates that there is significant coating formation that occurs*
*between the timescale of ~1 day to ~1 week for biomass burning rBC, even after rapid coating formation that occurs soon after emission. An important caveat is that $CT_{BC}$ of biomass burning rBC may not be monotonically increasing over time. Past studies have observed rapid coating of biomass burning rBC within less than one day to more than 100 nm, but we observed a median $CT_{BC}$ of ~48 nm for L3, which suggests that $CT_{BC}$ for biomass burning rBC might decrease at some point during atmospheric transport and then increase later at longer timescales (e.g., median $CT_{BC}$ ~54 nm for L9). We make no*
*definitive claims about the rate of change of $CT_{BC}$ for biomass burning rBC throughout atmospheric transport since we only observed the $CT_{BC}$ from a single discrete point in space, but our measurements do suggest that $CT_{BC}$ for Southern California biomass burning rBC were generally lower than $CT_{BC}$ for Northern California biomass burning rBC.*

*Conclusion Point 7: The high variability of the rBC measurements on Catalina Island during three different campaigns*
*demonstrates how meteorology, emissions source type, and atmospheric aging can drastically affect the physical properties and mixing state of the broader BC population within the same region.*

*See the updated manuscript (with track changes) for comprehensive view of changes made in the Conclusion section.*

========================================

(3.1)

**Measurements of coating thickness can become biased if the particles are sufficiently small that the signal-to-noise ratios of the instrument's scattering channels aren't sufficient to successfully retrieve a coating thickness or a delay time. Was the rate of failed retrievals monitored? How was this reflected in the data?**

(3.2)

To minimize the signal-to-noise ratio for the LEO analysis, we only mainly considered LEO-fits for particles with rBC core diameters between 200 and 250 nm (as mentioned in Section 2.7). For the particular SP2 unit used in our study, Krasowsky et al. found that a lower threshold of 200 nm was sufficient to reduce the scattering signal-to-noise ratio (Krasowsky et al.,

2018). Previous work by Taylor et al. (2014) defined a lower bound of 135 nm for LEO fitting, which corresponds to a 50% fraction of rBC with detectable split-detector notch position. We chose an even more conservative lower bound of 200 nm in this study to further minimize the scattering signal noise. The 250 nm upper bound is also conservative, considering that previous studies have reported LEO-fit coating thicknesses for particles with rBC core diameters up to 290 nm (Dahlkötter et al., 2014). To check for potential biases due to the saturation of the scattering signal at larger rBC diameters, a subset of the

SP2 data (from 7 September 2017) was assessed to see what proportion of the low-gain scattering channel data were saturated. None of the particles (in this subset of data) with rBC core diameters under 250 nm were found to have saturated scattering signals. The fraction of rBC-containing particles that were selected for LEO-fitting (with respect to all rBC particles measured) was not explicitly reported in the manuscript, but the total number of particles analyzed in each LEO period was listed in Table 2. A number of previous studies have also reported only the size range of LEO-fit rBC particles, without explicitly stating the fraction of particles that were LEO-fit versus not LEO-fit. We think that including the size range of the LEO-fit particles and the total number of particles analyzed in the manuscript is sufficient to show that we adequately constrained noise in the scattering signal and also analyzed enough particles to produce robust coating thickness statistics for the L1 to L10 analysis. Other rBC core size intervals were considered and further explanation is provided in the new text shown below (from section 2.7 in manuscript).

For the lag-time (delay time) analysis, a lower threshold of 170 nm was implemented for rBC core diameters. The reasoning for the lower limit is the same as explained above. The only difference is that we relaxed the lower threshold a bit (compared to 200 nm) because the accuracy of the scattering signal is not as crucial to the binary categorization of rBC-containing particles as "thickly-coated" versus "thinly-coated." Previous studies have conducted the delay time analysis with similar size ranges, or even lower thresholds (Krasowsky et al., 2018; Shiraiwa et al., 2007; McMeeking et al., 2011; Moteki and Kondo, 2007; and more).

(3.3)

*A sentence will be added to Section 2.6 to clarify that the lower threshold for rBC core diameter was set to 170 nm for the*

*lag-time (delay time) method.*

*"Only particles with an rBC core diameter greater than 170 nm were included in the calculation of $f_{BC}$ to account for the scattering detection limit of the instrument."*

*"In this study, the LEO "fast-fit" method was used with three points, and particles analyzed were restricted to those with rBC core diameters between 180 and 300 nm. Although the SP2 has been reported to accurately measure the volume equivalent diameter (VED) of scattering particles down to ~170 nm, a more conservative lower threshold of 180 nm was used for our study to reduce instrument noise at smaller VED values near the detection limit (Krasowsky et al., 2018). Specific rBC core diameter ranges were used for different analyses in this study and these ranges are explicitly defined*

*within each respective discussion. One exception was made to the 180–300 nm rBC core diameter restriction in section 3.7. For the analyses and discussion presented in section 3.7, the LEO coating thickness was calculated for all detectable rBC particles with non-saturated scattering signals. The rBC core size was not restricted in this section because the relative comparisons between characteristic coating thickness values were more important for the analysis, rather than the absolute value (which would likely be biased, as discussed further in section 3.8). In other words, the LEO-derived coating thickness*

*values in section 3.7 were not used to report representative averages for selective time periods, but rather were used for comparative and/or qualitative purposes."*

=====================================

(4.1)

**Setting 'calm' winds as zero on direction on figure 5 makes no sense as this also corresponds to north. The periods should probably be blanked out instead.**

(4.2)

Calm winds have been removed from the wind direction plot as suggested.

(4.3)

*See updated Figure 5.*

=====================================

(5.1)

**The points plotted on figure S9 should be individually identified according to event.**

(5.2)

Figure S9 (now Figure 12) has been modified to show the scatter between coating thickness and rBC core diameter for all measurements. 1-minute mean values for both coating thickness and rBC core diameter are used for the scatter plot. A

statistically significant positive correlation was found and is shown on the figure ($r = 0.5397$ with p-value $<0.001$).

(5.3)

*See updated Figure S9 (now Figure 12 in main manuscript).*

================================

References

Dahlkötter, F., Gysel, M., Sauer, D., Minikin, A., Baumann, R., Seifert, P., Ansmann, A., Fromm, M., Voigt, C. and

Weinzierl, B.: The Pagami Creek smoke plume after long-range transport to the upper troposphere over Europe - Aerosol properties and black carbon mixing state, Atmos. Chem. Phys., 14(12), 6111–6137, doi:10.5194/acp-14-6111-2014, 2014.

Krasowsky, T. S., Mcmeeking, G. R., Sioutas, C. and Ban-Weiss, G.: Characterizing the evolution of physical properties and mixing state of black carbon particles: From near a major highway to the broader urban plume in Los Angeles, Atmos.

Chem. Phys., 18(16), 11991–12010, doi:10.5194/acp-18-11991-2018, 2018.

McMeeking, G. R., Good, N., Petters, M. D., McFiggans, G. and Coe, H.: Influences on the fraction of hydrophobic and hydrophilic black carbon in the atmosphere, Atmos. Chem. Phys., 11(10), 5099–5112, doi:10.5194/acp-11-5099-2011, 2011.

Moteki, N., Kondo, Y., Miyazaki, Y., Takegawa, N., Komazaki, Y., Kurata, G., Shirai, T., Blake, D. R., Miyakawa, T. and

Koike, M.: Evolution of mixing state of black carbon particles: Aircraft measurements over the western Pacific in March

2004, Geophys. Res. Lett., 34(11), doi:10.1029/2006GL028943, 2007.

Shiraiwa, M., Kondo, Y., Moteki, N., Takegawa, N., Miyazaki, Y. and Blake, D. R.: Evolution of mixing state of black carbon in polluted air from Tokyo, Geophys. Res. Lett., 34(16), 2–6, doi:10.1029/2007GL029819, 2007.

**Author Response to RC2**

We appreciate the thoughtful and detailed review from Referee 2. We have taken the comments made by Referee 2 into careful consideration and they have helped improve our manuscript.

The general format of this response is as follows:

- Reviewer comments are in bold and labeled as (N.1), where N is the number of the comment block.
- Author response to comments are in regular, non-bolded text, and labeled as (N.2).
- Modifications in the manuscript are described in italics and labeled as (N.3).

==================================

(1.1)

**Major comments regarding source attribution and estimated plume age**

(1.2)

We agree with the reviewer that the source attribution and plume age sections of the manuscript required some major revisions (e.g., section 3.7, formerly section 3.6). We now shift our focus towards comparing the mixing state during different known source impacts, rather than focusing on the plume age. As the reviewer notes, rBC from biomass burning (BB) is coated much more quickly than rBC from urban emissions, and BB rBC has also been observed to have thicker coating overall compared to its urban counterpart.

We would like to address the nuances associated with the specific concerns that the reviewer raises in the comment.

Regarding biomass burning source attribution:

First, we wanted to clarify that we are not definitively attributing ~10 nm coating thickness values to fresh BB rBC particles, and we changed the language in the revised text to make this clear. ~10 nm was the median coating thickness from a population of aerosols that had a larger spread of individual coating thickness values. The coating thickness values on the higher end of the distribution tail (and outliers) are likely attributable to the BB impacts. We clarified in the new text that the peaks in the 2nd campaign (e.g., L4) were likely dominated by urban emissions, but that we could not exclude the likely impact of BB emissions mixing into the broader urban plume. In fact, we still believe that biomass burning did impact our measurements to some degree, even if it was a minor fraction of total sampled rBC. In particular, the Thomas Fire was one of the largest fires in California history, and it was still active during the 2nd campaign (20-22 December 2017). With the center of the Thomas Fire less than 150 km away, and strong atypical Santa Ana winds recorded before and during the time of measurements, it is hard to imagine BB having no impact on the regional rBC loading at the time. In addition to geographic proximity and meteorology, the air quality monitoring stations in Santa Barbara, Ventura, and Los Angeles all recorded elevated concentrations of $PM_{2.5}$ right around this time period. Additionally, as part of the new supplemental analysis, the HYSPLIT dispersion model was run to simulate the plume dispersion of multiple active fires during the December 2017 campaign. The HYSPLIT dispersion model shows the plumes from the Thomas Fire and several other smaller Southern California fires directly impacting the point of measurement (Catalina Island). These results are included in the revised Supplement. We also added a new qualitative analysis in the Supplement using CALIPSO lidar transects in the

Southern California region during the 20-22 December period. From the CALIPSO transects we observed aerosols that were attributed to BB sources present just off the coast of Southern California around this time. This data is also shown in a new section in the Supplement.

Second, since the paper was first submitted, we have obtained levoglucosan data from November 2018 (3[rd] campaign) that were collected by colleagues at USC who were conducting an independent air quality study in the LA Basin (Soleimanian et al. 2020). Although the reviewer's comment was particularly focused on the L4 period, we would like to point out that the conditions during the 2[nd] campaign (December 2017) and the first portion of the 3[rd] campaign (November 2018) were quite similar. Geographically, there were multiple fires throughout the Southern California region in both campaigns (see Figure 3). Both campaigns were also characterized by Santa Ana (i.e., northerly and easterly) winds. The weekly average concentration of levoglucosan between 7 to 14 November and 15 to 22 November was 187.5 ng m$^{-3}$ and 83.89 ng m$^{-3}$, respectively. Note that the 3[rd] campaign took place between 12 and 18 November 2018. For reference, levoglucosan concentrations during July 2018 (non-wildfire season) ranged between ~4 and 17 ng m$^{-3}$. The elevated concentration of levoglucosan inside the LA Basin during November 2018 removes any lingering doubt that BB aerosols were mixed into the broader regional air mass that was measured on Catalina Island. Given that similar fire and meteorological conditions were present during the 2[nd] campaign (December 2017), we have high confidence that BB also played at least a minor role in this campaign as well.

Regarding plume age comments:

For the LEO periods mentioned (L3, L8, L9, and L10), the aging timescale range of ~days to a week was meant to serve as a range of possibility rather than an exact aging timescale. We fully acknowledge the limits of HYSPLIT, especially for complex trajectory patterns. That is exactly why we present a very general range of timescales that was based on physical distance from major sources rather than relying on the exact timing of crossovers from the back-trajectories. The reviewer also mentions the loss of rBC coating with aging. This is entirely consistent with the $CT_{BC}$ values measured during periods impacted by long range transport of biomass burning impacted air masses (e.g., L8 and L9). The median $CT_{BC}$ values were within the range of ~60-70 nm during this time period of impact from the Camp Fire. Previous airborne studies have measured average coating thickness values of ~100 nm within hours of emission within the plume. Given that our values are significantly lower, the rBC measured in our study likely did experience coating loss at some point during transport. We added a short discussion on this topic of coating loss in the coating thickness discussion section and below in the section (1.3). Furthermore, we have added a new section that comprehensively compares our campaign measurements with past
mixing state studies conducted with an SP2.

Regarding precipitation comments:

Although the data were not reported, precipitation and cloud cover were monitored throughout the campaigns. There were no precipitation events in the region during any of the measurement days, and most of the days were clear to partly cloudy.

(1.3)

*Major edits were made to section 3.5 (formerly section 3.4) and section 3.7 (formerly section 3.6). A new section 3.8 was added to comprehensively compare our results to past similar studies. Additional evidence (i.e. using CALIPSO lidar data, HYSPLIT dispersion model and levoglucosan measurements) and figures were also added to the Supplement to make our*
*discussion on source attribution more robust. Specifically, please refer to Supplement section S2 and figures S11 through S20.*

*Revised main points regarding variability of coating thickness:*
- *Timescales of less than 24 hours were too short to significantly coat rBC from urban emissions. This is in direct*
*contrast to biomass burning rBC, which has been shown in previous works to acquire thick coatings within hours or even minutes, near the source of emission.*
- *Aged rBC from biomass burning sources were generally more thickly coated, although the time evolution of the mixing state could not be quantified directly over the duration of transport. Periods of "fresh" biomass burning impacts were characterized by slightly thinner $CT_{BC}$ compared to aged biomass burning rBC particles (e.g., L3 vs.*
*L8), but larger $CT_{BC}$ compared to fresh urban rBC particle (e.g., L3 vs. L4). This agrees with previous studies that have also observed thicker coatings in fresh biomass burning rBC relative to fresh urban rBC. The overall larger $CT_{BC}$ for aged biomass burning rBC relative to fresh biomass burning rBC indicates that there is significant coating formation that occurs between the timescale of ~1 day to ~1 week for biomass burning rBC, even after rapid coating formation that occurs soon after emission. An important caveat is that $CT_{BC}$ of biomass burning rBC*
*may not be monotonically increasing over time. Past studies have observed rapid coating of biomass burning rBC within less than one day to more than 100 nm, but we observed a median $CT_{BC}$ of ~48 nm for L3, which suggests that $CT_{BC}$ for biomass burning rBC might decrease at some point during atmospheric transport and then increase later at longer timescales (e.g., median $CT_{BC}$ ~54 nm for L9). We make no definitive claims about the rate of change of $CT_{BC}$ for biomass burning rBC throughout atmospheric transport since we only observed the $CT_{BC}$ from a single*
*discrete point in space, but our measurements do suggest that $CT_{BC}$ for Southern California biomass burning rBC were generally lower than $CT_{BC}$ for Northern California biomass burning rBC.*
=====================================

(2.1)

**Major comments regarding number size distribution data**

(2.2)

Although we generally acknowledge the concerns about fitting a log-normal distribution to a set of observations without a discernable peak, we also believe that the log-normal fits have value and should be reported (with associated uncertainty clearly described). First, there have been a number of past studies that have also included log-normal fits for their number size distributions, even in cases where the peak in the measured data was ambiguous. At the end of (2.2) is a comprehensive, but not exhaustive, list of studies using the SP2 that have included log-normal fits to rBC number size distributions. Full references are provided at the end of the document.

Second, the physical lower bound on BC core size makes log-normal fitting reasonable in the Aitken range, even if it is below the SP2 detection limit. Single BC nanospheres (i.e., individual spherules) have been observed to be ~20-30 nm in diameter by using TEM imaging techniques (Ellis et al., 2016; Wentzel et al., 2003). Although the detection limit of the SP2 for rBC cores is ~70 nm, it seems reasonable to assume that the peak of the rBC number size distribution in this Aitken range would be between 50 and 80 nm (Kondo et al., 2011b), given that individual BC spherules are unlikely to be smaller than 20 nm. This would naturally imply that most (if not all) BC cores in the ambient air are larger than 20 nm, but smaller than the point at which we observe a sharp increase in the slope on the right-hand side of the number size distribution. This inflection point on the right-hand side is clearly observed from SP2 data, even when the peak is not completely discernable.

Third, even if there was an unmeasured bimodal peak beyond the detection limit of the SP2, the median of the extrapolated log-normal fit would not be a completely useless metric for comparison. As long as the log-normal fitting is consistent between all instances of distributions, it would serve to characterize the Aitken mode of the rBC core size distribution, even if there was another mode lurking in the ultrafine range. This would suggest that the existence of an unknown local maxima in the ultra-fine range is possible, but that it would not invalidate the inter-comparison of Aitken mode distributions for different time intervals.

Fourth and lastly, the appropriateness of the log-normal fit is not entirely contingent upon the explicit observation of a local maxima. It might be entirely inappropriate if we saw that all the observed data points deviated sporadically from the fit curve, but we observe the fit curve describing the observed number size distribution data points very well, with fairly small residuals. We see that the rate of change of the slope is well captured by the fit, which strongly suggests that a log-normal fit is likely representative of the actual distribution. Analogously, we find the LEO-fit for coating thickness quantification as a robust method for mixing state analysis, even though we only use the leading edge of what we expect to be a Gaussian signal. Indeed, the LEO-fit uses an even smaller fraction of the expected Gaussian scattering response compared to the log-normal fits for the number size distribution. Likewise, we are using the existing "edge" of size distribution to fit what we expect to be log-normal.

To address the reviewer's concern with this issue, we made a clear caveat in the text explaining the limitations of the extrapolation, in addition to the already existing disclaimer about the lower detection limit in the first paragraph of section 3.6 (formerly section 3.5). We made it clear and explicit that the peak based on log-normal fits are not definitive measured values, but rather modeled based on reasonable assumptions about the behavior of the distribution in the Aitken range.

The typo in Figure 8 regarding the wrong median value label has also been fixed.

List of publications that have used log-normal fits to the number size distribution data:

Cheng et al., 2018; Kondo et al., 2011a; Kondo, et al., 2011b; Krasowsky et al., 2018; Metcalf et al., 2012; Moteki et al.,
2012; Raatikainen et al., 2017; Reddington et al., 2013; Sahu et al., 2012; Schwarz et al., 2008; Shiraiwa et al., 2008

(2.3)
*An additional caveat has been added to section 3.5 (formerly section 3.5) in the manuscript in tracked changes to address the comments and concerns made by RC2.*

*"Figure 10 shows that log-normal fits adequately capture the measured size distributions, though we cannot rule out the possibility of another rBC mode outside the detection limits of the SP2. Although the peak of the observed points is not always discernible (e.g., number size distribution for L5 in Figure 10), it is reasonable to fit these points assuming that a log-normal distribution is a realistic representation of ambient rBC number size distributions in the Aitken mode."*

================================

(3.1)
**Major comments regarding increasing rBC diameter with atmospheric aging**

(3.2)
We agree with the reviewer that the effect of coagulation on the rBC core size is likely overplayed in the manuscript since the rBC number concentration is relatively low in the ambient air at the point of measurement, compared to the rBC number concentration very close to the source of combustion (e.g., in a tailpipe or in a BB flame). We would like to point out, however, that there is a noticeable shift in the rBC size distributions during time periods dominated by urban emissions (e.g.,
L4 and L5) relative to size distributions that were measured inside Los Angeles near a major highway by Krasowsky et al. (2018). This is a particularly useful comparison because the exact same SP2 was used with the same operating variables. Focusing on the number size distribution, we observe a larger count median diameter during the L4 and L5 periods compared to the count median diameters measured downwind of a highway in a polluted urban environment (Krasowsky et al., 2018). The size distribution of rBC can only be affected by, (i) the emission source type and/or (ii) coagulation of rBC-containing particles. Related to the reviewer's comment regarding source attribution, we believe that both of these factors likely played some role in the variability of rBC core sizes. We are quite confident that BB sources did contribute, at least in part, to time periods dominated by urban emissions. (see comment block 1 above for details). So, there is likely a source effect. It seems plausible that a mixture of BB impact and coagulation (at least near the source, within the polluted urban basin), contributed to this noticeable shift in the core size distribution.

The reviewer also notes that the cited studies were conducted under higher rBC concentrations than what we encountered in our study. However, while the studies mentioned did have higher campaign-averaged concentrations, the peak concentrations were within the same magnitude, especially for the Shiraiwa et al. study (2008), which took place in the East Asia outflow. The peak magnitudes reported in Shiraiwa et al. reached ~ 1 $\mu g\ m^{-3}$, which is within a factor of two relative to the larger peaks measured in our study (~0.6 $\mu g\ m^{-3}$). Shiraiwa et al. (2008) briefly mention that coagulation could be a potential mechanism that explains why aged particles from China and Korea were larger than particles associated local urban emissions from Japan. While we agree that coagulation at measured concentrations would be slow and possibly negligible, we believe that coagulation could have played a minor role during atmospheric transport from the LA basin to Catalina Island. We make no attempt at quantifying the rate at which coagulation occurs for LA basin dominated air masses, but we qualitatively acknowledge that coagulation likely contributed to the growth of particles, as per the logic above, especially within the first few hours of aging.

(3.3)
*The focus of the paragraph mentioned by the reviewer has been shifted towards an emphasis on source-related impacts rather than impacts from atmospheric processing (i.e., coagulation). A short mention of coagulation still remains, but it serves as a qualitative acknowledgement of its likely minor effect on rBC size distributions. See section 3.6 (formerly section 3.5) for tracked changes.*

*Relevant excerpts from new text in section 3.6:*
*"A survey of past studies that have reported rBC mass median diameter (MMD) and count median diameter (CMD) shows that the source of emissions has a strong influence on rBC core diameter (Cheng et al., 2018). The MMD [CMD] for biomass burning influenced rBC, which has been reported to range from ~130 nm to 210 nm [100 to 140 nm], is generally much larger than the MMD for urban emissions influenced rBC, which has been reported to range from ~100 nm to 178 nm [38 to 80 nm] (Shiraiwa et al., 2007; Schwarz et al., 2008; McMeeking et al. 2010; Kondo et al., 2011a; Sahu et al. 2012; Metcalf et al., 2012; Cappa et al., 2012; Laborde et al., 2013; Liu et al., 2014; Taylor et al., 2014; Krasowsky et al., 2018).*

*The MMD [CMD] for aged air masses in remote regions were reported to range from ~180 nm to 225 nm [90 nm to 120 nm] (Shiraiwa et al., 2008; Liu et al, 2010; McMeeking et al., 2010; Schwarz et al., 2010).*

*Figure 11 shows the rBC MMD and CMD based on the log-normal fits for each LEO period in this study. Based on the source identification discussed in section 3.1 and section S2 in the Supplement, the MMD and CMD values in this study are generally consistent with the ranges reported in past studies. For LEO periods when measurements were strongly influenced by biomass burning emissions (L3, L8, L9, L10), MMD ranged from 149 nm to 171 nm, which is within the range of ~130 nm to 210 nm compiled from past studies. Similarly, when measurements were strongly influenced by urban emissions (L2, L4, L7), the MMD dropped, ranging from 112 nm to 129 nm. This falls within the range of ~100 nm to 178 nm previously reported for measurements of urban emissions from past studies."*

*"Another explanation for varying rBC core size is coagulation (Bond et al., 2013). Shiraiwa et al. (2008) observed an increase in rBC core diameters in aged plumes compared to fresher urban plumes, suggesting that coagulation can alter the rBC size distribution during atmospheric transport (i.e., aging). Although the emissions source type appears to be the dominant influence on rBC core sizes in our study, there is evidence to suggest that coagulation also played a role during transport from the Los Angeles basin to Catalina Island (~70 km away). For example, we observed a $MMD_{fit}$ [$CMD_{fit}$] of 112 nm [53 nm] during L4, when $BC_{ff}$ was measured. This is noticeably larger than values of 93 nm [42 nm] reported in Krasowsky et al. (2018) for measurements conducted 114 meters downwind of a major highway in Los Angeles. Furthermore, Laborde et al. (2013) observed an $MMD_{fit}$ of ~100 nm for $BC_{ff}$ in Paris, which is again lower than the value of 112 nm calculated for L4. Even though it was determined that L4 was characterized by $BC_{ff}$, we cannot rule out the effects of local wildfires influencing the size distribution as well (as explained in the Supplement section S2). While the rBC size distribution from L4 suggests that coagulation plays at least a minor role, both factors (source type and coagulation) likely influence rBC size distributions to varying degrees in areas with varying emissions source types and relatively elevated rBC concentrations (e.g., polluted urban areas).*

===================================

(4.1)

**Page 1, line 18, 20. The passive voice exemplified by the use of the word "suspect". Are the author's hedging their bets? Suggest using a different - less passive - word.**

(4.2)

The wording has been changed.

(4.3)

*New text:*

*"In contrast, during periods when measured rBC was dominated by emissions from the Southern California region, both $f_{BC}$ and $CT_{BC}$ were significantly lower, with a mean $f_{BC}$ of ~0.03 and median $CT_{BC}$ ranging from ~0 to 10 nm."*

================================

(5.1)

**Page 1, lines 23-25. The author's write "we conclude that an aging timescale on the order of ~hours is not long enough for rBC to become thickly coated under the range of sources sampled and atmospheric conditions during this campaign." This is misleading as several papers that have studied biomass burning (and those currently under review and data currently being analyzed) have (and are) showing that rBC particle become thickly coated very quickly.**

**While this might be true for urban plumes, it certainly is not for BB (biomass burning) plumes. Please clarify.**

(5.2)

We agree with the reviewer and we have changed the main conclusions of our paper to reflect this. Further response to this specific issue has been discussed in more detail above in comment block 1.

(5.3)

*Any text related to the generalization of thin coatings for particles aged less than 24 hours has either been removed or modified.*

*This was also discussed in greater detail in comment block 1 and applicable changes have been made in sections 3.5 and 3.7 (formerly sections 3.4 and 3.6).*

================================

(6.1)

**Page 2, lines 43-44. The author's write "BC is emitted mostly as an "external" mixture, physically separated from other aerosol species." This is a bit misleading. It is very dependent upon when the plume is sampled. With respect to biomass burning, research has shown that rBC becomes coated within the first few minutes following generation due to the chemical richness of the smoke plumes.**

**Please reword to reflect this.**

(6.2)

We acknowledge that BC can become coated very quickly and that this statement could potentially be misleading. The original intent was to give a conceptual overview of externally versus internally mixed BC. The description has been altered to remove any ambiguities regarding emission point and timescale since emission.

(6.3)

*The text in the introduction (section 1) has been altered to describe the two general types of mixing state without potentially misleading readers into believing that all BC is uncoated in the near-field plume.*

*"A BC particle that is physically separate from other non-BC aerosol species is considered externally mixed. On the other hand, BC is considered internally mixed if it is physically combined with another non-BC aerosol species (Bond et al., 2006; Schwarz et al., 2008a). As freshly emitted BC particles are transported in the atmosphere, they can obtain inorganic and organic coatings from either gaseous pollutants that condense onto the BC, oxidation reactions on the BC surface, or the coalescence of other aerosol species onto the BC, making them more internally mixed (He et al., 2015). In general, the*
*mixing state of BC describes the degree to which BC is internally mixed (Bond et al., 2013). The BC mixing state near the point of emission as well as the evolution during aging in the atmosphere of the mixing state can vary widely, depending on the source of emissions and atmospheric context.*

======================================

(7.1)

**Page 3, line 74 and 75. The authors need to be very disciplined in their use of "mixing state", as one can be describing the aerosol mixing state (e.g., external vs internal) or the particle mixing state (e.g., coated or uncoated rBC). Yes, the authors sort of point this out on page 2 (lines 48-50) but then start interchanging "internal mixing state" with mixing state. For example, on the opening sentence of the cited paragraph, are the authors referring to the internal mixing**
**state or the aerosol mixing state? Later in this paragraph, the authors reference internal mixing state of rBC (line 80). Please ensure consistency.**

(7.2) We acknowledge this potential for confusion and changed the language throughout the manuscript to ensure consistency. For the sake of simplicity and consistency, we initially define externally mixed BC as "uncoated BC" and
internally mixed BC as "coated BC." Furthermore, we use the general term "mixing state," to refer to the extent to which BC is coated, either at an individual particle level or aggregated (i.e., sample population-wide) level.

(7.3)

*We edited the text to ensure consistency between any language describing the mixing state. This topic was also discussed in*
*comment block 6 above.*

================================

(8.1)

**Page 3, lines 74 - 75. Here are two additional references to the use of microscopy with quantifying rBC mixing state that the authors are encouraged to consider: Adachi, K., Chung, S. H., and Buseck, P. R.: (2010) Shapes of soot aerosol particles and= implications for their effects on climate, J. Geophys. Res. Atmos., 115. Adachi, K., Moteki, N., Kondo, Y., and Igarashi, Y.: (2016) Mixing states of light-absorbing particles measured using a transmission electron microscope and a single-particle soot photometer in Tokyo, Japan, JGR.,121, 9153–9164.**

(8.2)

Thank you for the references and suggestion. They have been added to the manuscript.

(8.3)

*These references have been added to the introduction of the manuscript where microscopy is briefly mentioned.*

================================

(9.1)

**Page 3, lines 80 - 83. Authors are encouraged to review (include) the work by Sedlacek et al., who investigated the utility of the SP2 lagtime methodology [Investigation of Refractory Black Carbon-Containing Particle Morphologies Using the Single-Particle Soot Photometer (SP2) (2015) Aero. Sci. Tech., 49:872]**

(9.2)

Thank you for the suggestion. We have incorporated this reference into our study and expanded on our analysis by including discussion about negative la-times and rBC morphology in the discussion section. See also (11.1) below, which is related to this comment.

(9.3)

*See section 3.4 on negative lag-times and rBC morphology for newly inserted analysis and discussion.*

*Excerpt from new text:*

*"In this study, we observed negative lag-times, although at a relatively low rate, with $f_{lag,neg}$ calculated to be much less than 0.1 throughout most of the measurement periods. We defined $f_{lag,neg}$ to be identical to the "fraction of near surface rBC particles" metric used by Sedlacek et al. (2012), using a lag-time threshold of -1.25 μs to account for uncertainties*

*associated with the lag-time determination. The campaign-wide $f_{lag,neg}$ was 0.017 for the first campaign (September 2017),*

*0.018 for the second campaign (December 2017), and 0.026 for the third campaign (November 2018). Comparatively,*

*Dahlkötter et al. (2014) observed $f_{lag,neg}$ of ~0.046 during an airborne field campaign measuring an aged biomass burning*

*plume, and additionally calculated a higher fragmentation rate of ~0.4 to 0.5, based on their aforementioned alternative*

*method (Laborde et al., 2012). Sedlacek et al. (2012) reported $f_{lag,neg} > 0.6$ for ground-based measurements of a biomass*

*burning plume in Long Island, New York, originating from Lake Winnipeg, Canada."*

*See new section 3.4 in manuscript for full details regarding negative lag-times and rBC morphology.*

===================================

(10.1)

**Page 3, line 83. The authors are encouraged to review (include) the work by Moteki and Kondo who have also contributed significantly to improving the quantification of the rBC mixing state [Method to measure time-dependent scattering cross sections of particles evaporating in a laser beam (2008) J. Aer. Sci. 39:348].**

(10.2)

Thank you for the suggestion. This study was not initially included in the manuscript because the method described in Moteki and Kondo (2008) was not used for our mixing state analysis. Nevertheless, we have added the reference in the initial description of the LEO method because of its relevance to the Gao et al. (2007) method, which we used in our study.

(10.3)

*The reference has been added to section 2.7 in the manuscript.*

===================================

(11.1)

**Page 9, lines 226 - 228. The authors might consider reviewing (including) the work by Sedlacek et al., who looked at**
**the SP2 lagtime for a biomass burn plume. [Determination of and Evidence for Non-core-shell structure of particles containing black carbon using the single particle soot photometer (SP2). (2012) GRL. 39]**

(11.2)

Thank you for the suggested work. We have added an additional short section about the morphology of rBC in the results
and discussion section of the manuscript, and we use the same near-surface fraction analysis that Sedlacek et al. (2012) employed in their study. The reference has been added as well.

(11.3)

*See section 3.4 on negative lag-times and rBC morphology for newly inserted analysis. Also see comment block 9 above for*
*related discussion.*

========================================

(12.1)

**Page 10, Line 266. As highlighted earlier, please refrain from relying on a passive voice (e.g, "suspect".)**

(12.2)

Passive voice removed.

(12.3)
*The word "suspect" has been removed from referenced text.*

========================================

(13.1)

**Page 12: The authors show the back trajectories for each day of the campaign. Why not put this figure in the**
**supplemental and, instead, show those trajectories for the specific periods under discussion. This would make it**
**easier to evaluate the HYSPLIT datasets.**

(13.2)

Thank you for the suggestion. Although we see the value in the suggestion, we prefer to leave Figure 3 in its current state
and add a *separate* HYSPLIT figure either in Section 3.7 or in the Supplement. Our reason for showing all the trajectories in
Figure 3 is to show the campaign-wide perspective on the source locations of the particles. We also thought it would be
useful for visually comparing between the different campaigns, and not just for 10 to 15-minute LEO time periods, which
give limited snapshots instead of showing a broader campaign-wide "fingerprint" of trajectories.

(13.3)
*Additional figure with only LEO period back-trajectories has been added to the Supplement. This can also be added to*
*Section 3.7 if it is determined to be more appropriate there.*

========================================

(14.1)

**Page 14 line 307. The authors reference Figure S9, but I think they mean S8?**

(14.2)

Thank you for catching this typo.

(14.3)

*Changed from Figure S9 to Figure S8.*

==================================

(15.1)

**Page 15, lines 337 - 344. The authors are encouraged to review paper by Subramanian et al., [(2010) Black carbon over Mexico: the effect of atmospheric transport on mixing state, mass absorption cross-section, and BC/CO ratios ACP 10] where attention is drawn specifically to figures 3, 12 and 13.**

(15.2)

Thank you for the paper suggestion. The figures you suggested were carefully reviewed and they were helpful in putting our results in context of past studies like Subramanian et al. (2010). Brief comparisons are made to the results presented in Subramanian et al. (2010) to our results. Reference to the article has also been added to the manuscript.

(15.3)

*See minor additions in Section 3.3 and Section 3.7.*

==================================

(16.1)

**Page 22, lines 507 - 508. As highlighted above, this reviewer has concerns regarding the estimated plume ages.**

(16.2)

Appropriate changes have been made to the main conclusions from this paper, as described in more detail in Comment 1.1 above.  Most importantly, all blanket statements regarding an aging timescale of more than one day required for thick coating have been altered or removed.

(16.3)

*See revised manuscript for tracked changes. Specifically section 3.7 and section S2 in the Supplement.*

==================================

(17.1)

**Supplemental: page 1. As noted earlier, there is no table 3 in the main manuscript.**

(17.2)

The table was accidently omitted. Apologies for any confusion.

(17.3)

*Table 3 has been merged with Table 2. The old Table 3 is now part of Table 2.*

==================================
(18.1)

**Supplemental: page 1, line 7. Suggest that the authors review Lund et al., [(2018) Short Black Carbon lifetime inferred from a global set of aircraft observations, npj Climate and Atmospheric Science 1, 31 doi:10.1038/s41612-018-0040-x]**

(18.2)

Thank you for the suggested article. The mean BC lifetime of ~4 days over the Pacific as suggested by Lund et al. (2018) further supports our estimated range of source-to-receptor timescales. We would like to clarify here that our loosely restrained timescales are only meant to give readers an idea of the range of possibilities regarding how long measured particles were transported in the atmosphere. Since the estimated value of ~4 days was meant to represent the mean, individual particles measured during our campaigns could certainly have been aged longer (i.e. ~week).

(18.3)

*Citation added to text.*

==================================
(19.1)

**Supplemental: page 7. This is a stylist comment. Would suggest using a different color to denote the sample location on Catalina Island. The currently used green color is hard to discern with the yellow star.**

(19.2)

Style change made as suggested.

(19.3)

*Green circle removed from the figure as it was unnecessary.*

======================================

[revised manuscript text omitted]

There were some periods in which we observed a relative increase in the MMD$_{fit}$, but concurrent decrease in the CMD$_{fit}$. For example, L5 exhibits a relatively high MMD$_{fit}$ (~171 nm), which  suggests that this was a biomass burning dominated  time period, but the CMD$_{fit}$ is the second lowest of all the LEO periods (~53 nm). L10  exhibits a similar pattern. In  such  seemingly contradictory situations, it is likely that biomass burning aerosols are entrained into a broader urban plume (e.g., from Los Angeles basin). An urban plume with no biomass burning influence is expected to exhibit a very low CMD$_{fit}$. As  biomass burning aerosol gets entrained,  MMD$_{fit}$ is expected to change more than  CMD$_{fit}$ due to its larger rBC core size relative to urban rBC cores. (For a unit increase in diameter, the mass weighting will increase proportionally to the third power, while the size weighting will increase proportionally on a first order basis.) This may explain why i some cases we observe  relatively high MMD$_{fit}$ values along with relatively low CMD$_{fit}$ values,  highlight the need to examine both the number and mass size distributions for rBC core size analysis from the different sources in future studies.

 Another explanation for varying rBC core size is coagulation  (Bond et al., 2013). Shiraiwa et al. (2008) observed an increase in rBC core diameters in aged plumes compared to  fresher urban plumes, suggesting that coagulation can alter the rBC size distribution during atmospheric transport (i.e., aging). Although the emissions source type appears to be the dominant influence on rBC core sizes in our study, there is evidence to suggest that coagulation also played a role during transport from the Los Angeles basin to Catalina Island (~70 km away). For example, we observed  MMD$_{fit}$ [CMD$_{fit}$] of 112 nm [53 nm] during L4, when BC$_{ff}$ was measured.  and This is noticeably larger than values of 93 nm [42 nm] reported in Krasowsky et al. (2018) for measurements conducted 114 meters downwind of a major highway in Los Angeles. Furthermore, Laborde et al. (2013) observed an MMD$_{fit}$ of ~100 nm for BC$_{ff}$ in Paris, which is again lower than the value of 112 nm calculated for L4. Even though it was determined that L4 was characterized by BC$_{ff}$, we cannot rule out the effects of local wildfires influencing the size distribution as well (as explained in the Supplement section S2). While the rBC size distribution from L4 suggests that coagulation plays at least a minor role, both factors (source type and coagulation) likely influence rBC size distributions to varying degrees in areas with varying emissions source types and relatively elevated rBC concentrations (e.g., polluted urban areas).

~~There is variability in the median core diameters of both the mass-based and number-based size distributions. Looking specifically at the mass size distribution, the median diameter ranges between 112 nm (L4) and 171 nm (L5, L10). A study by Laborde et al. (2013) discusses the relationship between rBC core diameter and air mass type. According to Laborde et al., an average rBC core diameter of ~100 nm was observed for fresh urban emissions, while diameters of ~200 nm were observed for "continental air masses," which would be expected to be include a larger contribution of aged rBC. The mass size distributions from our LEO periods do not strictly adhere to a positive correlation between aging time and average rBC core diameter as reported by Laborde et al. (2013). In fact, sometimes we observe the opposite relationship. For example, as discussed in the previous section, L5 includes important contributions from freshly emitted rBC, but the mass median diameter is the largest out of all LEO periods (171 nm). Moteki et al. (2012) found negative correlation between aging timescale and rBC core size due to the fractal morphology of the rBC collapsing into a spherical morphology. This mechanism is in direct contrast to the coagulation mechanism described by Laborde et al. (2013), which would serve to increase the mass median rBC core diameter.~~

In our study, we observed a mix of rBC core diameters for different periods. For example, L8 and L9 exhibit higher mass median diameters than L4, L6, and L7. We assert that rBC measured during L8 and L9 are more aged than for L4, L6, and L7. As mentioned previously, L5 is inconsistent with the pattern of higher mass median diameter MMD with greater contributions of aged particles.

**3.7 Impact of emissions source and aging on rBC mixing state**

The dominant drivers for increased rBC core size (i.e., emission source type and aging) are also the driving factors influencing main drivers for increased CTBC. Figure 12 shows a scatter plot as a function of CTBC and rBC core diameter. The statistically significant correlation (r = 0.55, τ = 0.43) confirms that there is an in direct relationship between these two physical characteristics of BC. In other words, biomass burning (as opposed to fossil fuel) and longer aging generally seem to increase both the rBC core size and as well as the BC coating, and vice versa. Figure 13 shows the CTBC distributions for different rBC core size ranges, and a similar relationship between the two variables can be observed. As the core size increases (lighter to darker curves), a broader right-hand side tail is observed in the biomass burrBC, and it confirms that similar factors influence these attributes in As seen in Figure 12S10, we see a statistically significant, moderate positive correlation (r = 0.55, τ = 0.43R2 = 0.36) between median one-minute mean CTBC and one-minute mean rBC core diameter the median number-based diameter for rBC. This is consistent with the coagulation mechanism increasing core size with increasing age, as suggested by previous studies (Krasowsky et al., 2018; Laborde et al., 2013; Shiraiwa et al., 2008). In other words, since CTBC generally increases with atmospheric aging, the positive correlation between CTBC and number-based diameter found in this study supports previous suggestions that rBC core sizes increase with atmospheric aging. Additionally, we observe that changes in the mass median diameter are not consistent with the changes in number median diameter, although the figure is not presented here. Although previous studies, like Laborde et al. (2013), have focused on the mass median diameter, our results suggest that the number median diameter could be a more useful metric when correlating core diameters to mixing state metrics since the SP2 measures characteristics of individual rBC particles on a number basis and the CTBC is calculated for each measured particle.

CTBC distributions for each campaign, implying larger average CTBC for particles with larger rBC cores.

Source type (e.g., urban versus biomass emissions, and different types of fuels burned) can also play a significant role in determining rBC core size (Sahu et al., 2012; Pan et al., 2017; Laborde et al., 2013; Moteki et al., 2012; Metcalf et al. 2012; Wang et al. 2018). Past studies suggest that rBC cores from biomass burning emissions are larger. For

~~example, Metcalf et al. (2012) reported a mass median diameter of ~122 nm for rBC from urban emissions in Los~~
~~Angeles, while Sahu et al. (2012) reported a mass median diameter of ~190 nm for rBC from biomass burning~~

6 Evolution and variety of state

The dominant  factors that influence rBC core size (i.e., emission source type and aging) also
influence rBC mixing state. Figure 12 shows a scatter plot of
one-minute mean $CT_{BC}$ versus one-minute mean rBC core diameter. A statistically significant, positive
correlation (p < 0.001) was found, with $r = 0.55$. ==PLEASE ADD P VALUE … THAT'S WHAT DETERMINENS==
==SIGNIFICANCE.==. Further details regarding the statistical tests used to calculate the correlation coefficients and to conduct
the hypothesis testing can be found in section 3.5. a the potential correlation

~~is true. Both tests returned a p-value of ~0, which means that we could reject the null hypothesis with near 100% confidence,~~

implies a

The significant correlation  confirms
these two physicals characteristics characteristicss of BC.  that larger
contributions from biomass burning (as opposed to fossil fuel) and longer aging timescales are associated with
increases in both the rBC core size and the BC coating thickness. Figure 13 shows the $CT_{BC}$
distributions for different rBC core size ranges, and a similar relationship between the two variables can be observed. As the
core size increases (lighter to darker curves), a broader right-hand side tail is observed in the $CT_{BC}$ normalized distributions
for each campaign, implying higher mean average $CT_{BC}$ for particles with larger rBC cores.

[Figure]

**Figure 12.** rBC coating thickness versus rBC core diameter. Each point on the plot represents a 1-minute mean. Data from all three campaigns are shown. $CT_{BC}$ values are calculated for particles with rBC core diameters between 200–250 nm. The line represents the least-squares linear regression to the one-minute mean data points. There is a statistically significant positive correlation shown between $CT_{BC}$ and rBC core diameter, as shown in the summary box in the top left corner.

[Figure]

**Figure 13.** Distributions of BC coating thickness ($CT_{BC}$) aggregated by campaign and varying rBC core diameter ranges used in the LEO analysis. Panels (a) through (d) in the left column show the normalized frequency distributions, while panels (e) through (h) in the right column show the absolute frequency distributions. Within each panel, each line represents a distribution for a particular rBC core diameter range, with darker lines representing larger diameter ranges and vice versa.

[revised manuscript text omitted]

 Within the context of source identification discussed in previous sectionsidentifiable sources discussed in previous sections (section 3.1 and S2), it is clear that these distinct clusters in Fig. 16 are strongly influenced by emissions source type. The BC$_{bb}$ large cluster is  present in the third campaign (November 2018) when impacts from long-range transported biomass burning emissions were identified, but not in the second campaign (December 2018). Furthermore, a BC$_{ff}$ cluster is present in both the second (December 2017) and third campaign, but not in the first campaign (September 2017). This simpliesconfirms that fresh (age < 1 d), urban emissions from the LA basin and the surrounding southern California region lead to a distinct cluster characterized by~~are characterized by thin coatings and smaller core size, confirming what has also been observed in other past field studies (Laborde et al., 2012; Liu et al., 2014; Krasowsky et al., 2018).

The BC$_{aged,bg}$aged cluster (Fig. 16a, 16d) exhibits two distinct modes within the same cluster. One mode is characterized by a peak $CT_{BC}$ [CMD] that is ~20 nm [~10 nm] higher than the other mode~~, and a peak count mean diameter that is ~10 nm larger than the other modeBC$_{cont}$showsaged, continental , ambient airBC$_{cont}$blowingtowards CaliforniaContinental-scaleboth, as various plumes are entrained into one another throughout long-range transport.~~.

The time evolution of both $CT_{BC}$ and rBC core size is represented in a matrix series of scatter plots in Fig. 6 and 7. In each of the figures, the scatter between $CT_{BC}$ and rBC count mean diameter are grouped into six-hour time intervals for both the second (December 2017) and third (November 2018) periods, respectively.  In these figures, the time evolution of the BC physical properties can be examined in detail and compared to periods of known emissions source impacts. A similar figure for the first campaign (September 2017) is included in the Supplement as Fig. S23. .

In addition to emissions source type, atmospheric aging also appears to have an observableed effect on the mixing state. discernible effect on the physical attributes of BC.

Table 3 lists the range of estimated "source-to-receptor" timescales for rBC-containing particles measured during LEO time periods L1 to L10. In short, the first campaign (September 2017) is broadly characterized by source-to-receptor timescales on the order of days to a week. The second campaign (December 2017) is characterized by timescales of less than one day. FinallyAnd, the third campaign (November 2018) is characterized by timescales of less than one day for the first four days of the campaign, and timescales of approximately days to a week for the last two days of the campaign.

With regards to aging, we can first observeconclude that rBC fromwithin fresh, urban emissions dominated air masses$BC_{ff}$ particles do not developexhibit thick coatings within the timescales observed in this studyon average, in this study. This suggests that a timescale of less than one day is not sufficient to thickly coat urban rBC-containing particles in the lower boundary layer, in the Los AngelesA region. Although a modestly higher $CT_{BC}$ is observed during urban-dominated time periods, relative to $CT_{BC} \sim 0$ nm observed by Krasowsky et al. (2018) inside the LA basin, this is likely due to the effects of local biomass burning emissions mixing into the broader urban plume in both December 2017 and November 2018, as discussed above (also see section S2). While we observed mostly thinly-coated rBC from these urban-dominated time periods in this study, it must bewe acknowledged that the timescale required to acquire coatings on BC will likely this timescale ( < 1 d) cannot be applied as a blanket conditionstatement for all urban BC. The rate of coating is largely a function ofdiffer by location because of variations in local meteorology, pollution concentrations, and emission source profiles, which can widely vary from region to region.

Regarding the aging of forOn the other hand, biommass burning BCBC$_{bb}$biomass burning sources rBC, aAged rBC from biomass burning sources were generally more thickly coatedthickly-coated, although the time evolution of the mixing state could not be quantified directly in this study over the duration of transport. Periods with of "fresh" biomass burningFresh BC$_{bb}$ impacts were characterized byhad slightly lowerthinner $CT_{BC}$ CTBC compared to that of aged biomass burning rBC particlesBC$_{bb}$ (e.g., L3 vs. L98), but higherhigherlarger $CT_{BC}$CTBC compared to fresh urban rBC particlethat of fresh BC$_{ff}$ (e.g., L3 vs. L4). This agrees with previous studies that have also observed thicker coatings in fresh biomass burning rBC relative to fresh urban rBC. The overall higherlarger $CT_{BC}$ CTBC for aged biomass burning rBCBC$_{bb}$ relative to fresh biomass burningBC$_{bb}$ rBC indicates that there is some ignificant significant coatingcoating formation thatcan occur s between thewithin timescales of $\sim$1 day to $\sim$1 week for biomass burning rBCBC$_{bb}$, even after rapid coating formation that occurs soon after emission. An important caveat is that $CT_{BC}$ CTBC of BC$_{bb}$ biomass burning rBC may not be simply monotonically increasing over time.

Past studies have observed rapid coating of biomass burning rBCBC$_{bb}$ within the first few hoursone day to more than 100 nm (Perring et al., 2017; Morgan et al., 2020), but we observed a median $CT_{BC}$CTBC of 47.72 nm for L3, which suggests that

$CT_{BC}$ for $BC_{bb}$  might decrease  during atmospheric transport  and could again increase later at longer timescales (e.g., median $CT_{BC}$ of 54.0 nm for L9), although we would need simultaneous measurements near the point of biomass burning emissions in order to confirm this theory for a specific plume. Previous studies have noted that the competing processes of dilution-driven evaporation and oxidation-driven condensation determine the abundance of organic aerosol relative to carbon monoxide ($\Delta OA/\Delta CO$) in biomass burning plumes (Garofalo et al., 2019). Although the time evolution of $\Delta OA/\Delta CO$ is not necessarily indicative of rBC mixing state evolution, the same physical processes (i.e., evaporation and condensation) must apply to  rBC coating formation and loss potential evaporation. The conflicting observations from various studies showing $\Delta OA/\Delta CO$ either increasing, decreasing, or staying relatively stable in the near-field (timescale of ~hours) suggests that rBC coating in dense fresh biomass burning plumes may undergo similar competing physical mechanisms. Preliminary results from dDeveloping research showes that well-aged $BC_{bb}$ (>7 days) haves than fresh $BC_{bb}$ (< 5 h), providing emerging evidence that rBC coating may not always monotonically increase (Sedlacek et al., 2019). Further research is necessary to confirm this process in more field measurements, and to determine the various mechanisms that may be driving the loss of rBC coating in biomass burning plumes. We make no definitive claims about the rate of change of $CT_{BC}$  for $BC_{bb}$ throughout atmospheric transport since we measured $CT_{BC}$ at one location. Nonetheless, our measurements suggest that $CT_{BC}$ for fresh Southern California $BC_{bb}$ were generally lower than $CT_{BC}$ for aged Northern California $BC_{bb}$.

The contour plots for the first campaign (September 2017), shown in Fig. 16a and 16d, offer additional perspective on how aging can affect  BC mixing state within continental well-aged background air masses over longer aging time-scales (~days to week). The first notable feature of the $BC_{aged,bg}$cluster is that the smaller mode is significantly more coated than the thinly-coated$BC_{ff}$ clusters found in  Fig. 16 for the second (December 2017) and third (November 2018) campaigns. The peak of the smaller mode of the $BC_{aged,bg}$cluster is at least 35 nm higher than the peak of the thinly-coated $BC_{ff}$ clusters in Fig. 16b, 16c, 16e, and 16f. Assuming that this smaller mode represents fossil fuel influenced BC (i.e., urban BC), this confirms that while urban BC may not become thickly-coated within a day, they seem to acquire coatings over longer timescales.

Another prominent feature of the $BC_{aged,bg}$ cluster is the shift in mass mean diameter (MMD) (Fig. 16a) and count mean diameter (CMD) (Fig. 16d) peaks, relative to the $BC_{bb}$ and $BC_{ff}$ clusters in the December 2017 (Fig. 16b, 16e) November 2018 (Fig. 16c, 16f) plots.

representative of fossil fuel influenced (urban) BC,Specifically examining theComparing peaks of the smaller mode of the $BC_{aged,bg}$ cluster in both Fig. 16a and 16d to the respective peaks of the $BC_{ff}$ clusters in Fig. 16b, 16c, 16e, and 16f; we observe that the MMD is generally lower  for the smaller mode of $BC_{aged,bg}$ while the CMD is generally higher , relative to the respective peaks of the thinly-coated clusters. The apparent lower MMD in $BC_{aged,bg}$ compared to $BC_{ff}$ can be explained by the impact of local biomass burning sources  in both the second (September 2017) and third (November 2018) campaigns,  as previously mentioned in section 3.6. On the other hand, the overall higher peak CMD for the lower mode of the $BC_{aged,bg}$ cluster implies  that either (i) urban BC is coagulating over long aging timescales (days to week), (ii) source-specific variables like fuel type and combustion conditions are influencing the initial core size distribution, or (iii) the urban area in which the rBC is emitted contains a much higher concentration of rBC, leading to more coagulation in the near-field before continental-scale transport. Any combination of these three explanations could contribute to the overall increase in the peak rBC diameter of the urban mode. We do not attempt to quantify the extent to which each factor  contributes to core size increases in this study, though (i) is unlikely to be important given the dependence of coagulation rate on number concentrations.  Further research needs to be conducted to accurately characterize the relative importance of each factor.

Shifting focus to the larger mode in the $BC_{aged,bg}$ cluster in Fig. 16a and 16d, which we attribute to biomass burning BC, we notice lower  CMD and MMD, relative to the $BC_{bb}$ clusters  in Fig. 16c and 16f. Based on the assumption that the initial rBC size distribution of the biomass burning rBC from the first campaign and third campaign are similar, selective wet deposition and/or increased hygroscopicity of thickly-coated rBC mayalsosprobabilitycanfor a wellbiomass burning mode$BC_{cont}$s.. could~~

~~For the first campaign (September 2017), source identification analysis suggests that the measured rBC-containing particles were likely from aged biomass burning emissions and other unidentified sources of aged rBC-containing particles. The fact that all regional emissions were downwind of the sampling site during the first campaign suggests that measured rBC included negligible contributions from fresh emissions from the Los Angeles basin. There were no active wildfires in the~~

Southern California region at the time of the first campaign, but there were significant wildfires in the Pacific Northwest and the northern tip of California (near the California-Oregon border) around the time of measurements, as discussed in section 3.1. We suggest that measured rBC included contributions from these wildfires (see section 3.1), though we make no attempt
to quantitatively determining the relative contribution from these wildfires to our measurements.

**Table 3.** Estimated source-to-receptor timescales for rBC containing particles measured during the different LEO periods. Further detail on the methodology to determine these estimates can be found in the supplemental section S1.

[revised manuscript text omitted]

Camp Fire) to fresher particles (from Southern California fires and urban Los Angeles emissions).

The measurements from these three campaigns showed that rBC physical properties (rBC core size and coating thickness mixing state) were strongly influenced by (i1) emissions source type, and (ii2) atmospheric aging.

During periods when biomass burning emissions dominated measurements, we it was observed found that the average rBC core size, $f_{BC}$, and BC coating thickness ($CT_{BC}$) increased compared to when fossil fuel (urban) emissions were dominant. rBC from air masses dominated by biomass burning emissions ($BC_{bb}$) were generally had found to have e the larger average core diameters than relative to rBC from air masses dominated by urban emissions ($BC_{ff}$). The MMD mass mean diameter [CMD count mean diameter] of $BC_{bb}$ was observed to be ~180 nm [120 nm], while MMD [CMD] of $BC_{ff}$ was observed to be

~160 nm [100 nm]. rBC from aged, continental air masses ($BC_{aged}$) $BC_{aged,bg}$ $BC_{cont}$ showed a bimodal rBC core size distribution, with MMD [CMD] peaks at ~170 nm [115 nm] for the upper larger mode, and ~153 nm [109 nm] for the lower smaller mode. The bimodal rBC core size distribution in the well-aged aged background continental air mass suggests that continental-scale air masses background rBC aged rBC above the Pacific Ocean during typical meteorological conditions are likely a mix of both urban (i.e., smaller rBC cores) and biomass burning (i.e., larger rBC cores) emissions. The larger

CMD of the lower smaller mode for $BC_{aged,bg}$ $BC_{contaged}$ compared to the CMD of $BC_{ff}$ suggests that either (i) coagulation is increased sing the size of $BC_{aged,bg}$ $BC_{contaged}$ somewhere between the source and receptor, and/or (ii) the initial source and combustion conditions for $BC_{aged,bg}$ $BC_{contaged}$ were different than for $BC_{ff}$ specifically in this study. More accurate methods of source apportionment would be needed to definitively quantify the relative contribution of each factor, but both factors likely effect  aged urban rBC particles . The  CMD [MMD] of the larger mode for BC$_{aged,bg}$cont was smaller than that of BC$_{bb}$, which suggests that (i) selective wet deposition of larger particles , and/or (ii) increased wet scavenging of thickly-coated particles due to increased hygroscopicity, contributes to the shift in the rBC size distribution for biomass burning rBC-containing particles over long-range atmospheric transport.

Similar trends are  observed for the impact of emissions source type on rBC mixing state. On average, BC$_{ff}$ were either uncoated or very thinly-coated, with mean coating thickness ($CT_{BC}$) ranging from ~5 to  15 nm and mean fraction of thickly coated particles ( $f_{BC}$) of less than 0.15. In contrast, BC$_{bb}$ was more thickly-coated, with mean $CT_{BC}$ ranging from ~40 to 70 nm and $f_{BC}$ ranging from ~0.23 to 0.47. BC$_{aged,bg}$ was characterized by a mean $CT_{BC}$ of ~60 nm and $f_{BC}$ of ~0.27, confirming that a mix of biomass burning and urban emissions sources are likely entrained into these aged background air masses.

By estimating approximate source-to-receptor timescales (i.e., age) and also comparing the physical properties of fresh rBC to that of BC$_{aged,bg}$, we assessed the  effect of aging on both BC$_{bb}$ and BC$_{ff}$. For BC$_{ff}$, we observed that timescales of less than one day were not sufficient for urban rBC particles to become thickly coated. This is in direct contrast to biomass burning rBC, which has been shown in previous studies to acquire thick coatings within hours or even minutes, near the source of emissions. For BC$_{bb}$, we observed higher values of $f_{BC}$ and $CT_{BC}$ during periods that included contributions from the Camp Fire in Northern California, compared to periods of fresh biomass burning impacts from local Southern California fires (e.g., L3). The average $CT_{BC}$ during the Camp Fire impacted period was ~18 nm higher than the average $CT_{BC}$  during L3, when we identified Southern California fires as the main emission source. Likewise, we also observed an increase in the urban rBC-containing particles by comparing the aged urban mode of the BC$_{aged,bg}$ distributionof fresh BC$_{ff}$ during periods when emissions from the LA basin dominated. We found that coatings on  aged urban particles within  BC$_{aged,bg}$ were ~35 nm thicker than BC$_{ff}$ from fresh LA basin emissions. Overall, our measurements suggest  that aging increases the coating thickness on both BC$_{ff}$ and BC$_{bb}$, which is consistent with previous research. We did not quantify the rate of change of coating thickness since we were unable track the evolution of the mixing state during source-to-receptor transport.

The measurements reported in this study agree  with  past research that investigates  thereported in that a number of previous studies that have been~~

[revised manuscript text omitted]

---

## Author Response (AR2)

This document is organized as follows:

(1) Author Response to RC1

(2) Author Response to RC2

(3) Marked-up version of manuscript

**Author Response to RC1**

We appreciate the thoughtful and detailed review of the revised manuscript from Referee 1.

The general format of this response is as follows:

- Reviewer comments are in bold and labeled as (N.1), where N is the number of the comment block.
- Author response to comments are in regular, non-bolded text, and labeled as (N.2).
- Modifications in the manuscript are described in italics and labeled as (N.3).

=====================================

(1.1)

**Include information on how the SP2 was calibrated for both BC mass and scattering cross section.**

(1.2)

The calibration material used for both the rBC mass and scattering cross section is described now in section 2.2.

(1.3)

*The incandescence signal was calibrated using Aquadag and the scattering signal was calibrated using polystyrene latex*
*spheres.*

=====================================

(2.1)

**Should specify what scattering model was used as part of the LEO retrievals, specifically whether the effect of the BC specifically on the scattering cross section was included through the use of a core-shell model.**

(2.2)

Further details have been added to section 2.7. The effect of rBC on the scattering cross section is inherently built into the core-shell model assumption of the LEO method.

(2.3)

*Refractive indices of (2.26+1.26i) and (1.5+0i) were selected for rBC cores and rBC coating material, respectively. These*
*parameters were selected based on recommendations and results from previous studies (Moteki et al., 2010; Dahlkötter et al, 2014; Taylor et al., 2014; Taylor et al., 2015).*

==========================================

(3.1)

**Performing a p-test on the correlations in figures 12 and S9 isn't appropriate. This is effectively testing the probability that these might be two variables varying independently of one another that had coincidentally correlated (the null hypothesis), which is normally of use when there are few data points and statistical power is limited. But given there are a large number of data points here, the p value becomes vanishingly small to the point of being meaningless. I think it is sufficient to simply report the R value along with regression results.**

(3.2)

P-value removed as suggested.

(3.3)

*Changes made in figures 12 and S9 (now figures 13 and 9), and the accompanying text.*

==========================================

(4.1)

**Table 3 appears to be missing.**

(4.2)

Table added.

(4.3)

*See section 3.8.*

**Author Response to RC2**

We appreciate the thoughtful and detailed review of the revised manuscript from Referee 2.

The general format of this response is as follows:
- Reviewer comments are in bold and labeled as (N.1), where N is the number of the comment block.
- Author response to comments are in regular, non-bolded text, and labeled as (N.2).
- Modifications in the manuscript are described in italics and labeled as (N.3).

======================================

(1.1)

**Source attribution and increasing rBC diameter with atmospheric aging. As with the first review, this Reviewer still has issues with the authors interpretation of the derived rBC mixing state with respect to contributions from emission source vs plume age. The heart of the issue is that a fixed ground site cannot conduct a Lagrangian (or even a pseudo Lagrangian) sampling strategy. This inescapable limitation for ground sites is why aircraft operations are so important - aircraft put the ground site measurements in context. Unfortunately, no such aircraft measurements were available during this Catalina Island measurement campaign. This absence greatly complicates the authors ability to interpret the collected data in a region that is characterized by a dynamic mixing of fresh urban emissions, very aged aerosols, and biomass burning. For example, the authors argue (page 27, line 630-633) that significant correlation between thicker coatings and larger rBC modes is due to contributions from both differing sources (urban vs. wildfire) and longer time scales. This reviewer is not convinced plume aging plays any role in what the authors observe. Instead, the far simpler explanation is that the changes observed with the rBC particles are driven solely by the mixing of emissions sources (urban vs wildfire) - wildfires generate larger diameter rBC particles that are more thickly coated that become mixed with smaller diameter, more thinly coated urban rBC particles. Period. Such a simplified explanation also reconciles the unpersuasive argument by the authors (Page 27 lines 613-625) that coagulation is partly responsible for observed increase in rBC CMD (count median diameter). As this Reviewer pointed out in the original review, homogeneous coagulation under the conditions characteristic of this measurement period would be over 600x slower than at $10^4$ cc-1, readily relegating coagulation as being negligible. Further, to go from 93 nm to 112 nm (lines 615 - 619) would require a HUGH amount of coagulation. Bottom line: source emissions are driving observed changes in rBC microphysical properties. The authors need to reassess the data as being primarily driven by source and less about aging - especially given the absence of concomitant aircraft measurements coupled with the complexities of the measurement location (dynamic mix of urban, BB, and aged aerosols) that may not be captured by HYSPLIT.**

(1.2)

We agree that emission source type is the primary driver of rBC microphysical properties, but we also believe aging plays a non-negligible role, especially for coating enhancement. We acknowledge that coagulation will likely play a tiny, if not negligible, role in affecting the rBC core size in this study. Section 3.6 has been modified to reflect this emphasis on source emission type. We have left in place a very brief discussion on the possibility of coagulation in highly concentrated plumes, especially considering the precedence of Shiraiwa et al. (2008). We make it explicit that this could only be significant in large plumes with number concentrations that are orders of magnitude higher than the concentrations we measured (e.g., large wildfire plumes, or heavily polluted urban plumes on meteorologically stagnant days in Chinese megacities). The reviewer notes multiple times that the concentrations measured on-site are too low for any significant coagulation to occur. We agree with this statement. However, we would like to clarify that we never claimed that coagulation was happening at any significant rate near the point of measurement. Rather, we were referring to coagulation that could potentially occur in plumes with high number concentrations (near sources), well before the plume arrives to the measurement site. Nonetheless, we have minimized the discussion of coagulation in the rBC size discussion and made sure to emphasize that emission source type is the dominant influence of rBC core size differences in our measurements.

       While we agree that aircraft measurements are highly valuable and offer many advantages in tracking discrete plumes, we do not agree that the lack of such measurements leaves us in a position where we are unable to analyze emissions sources and source-to-receptor timescales. It is our view that we have gathered significant evidence based on a host of auxiliary data to support our comparisons (see Supplement and section 3.1). In particular, we have evidence that aging does play a role in our observed mixing state. For example, when we compared fresh vs. aged $BC_{bb}$, we were careful to isolate this comparison to periods when biomass burning was the dominant emissions source. Specifically, we were comparing emissions from two distinct, large wildfires (Camp Fire vs. Thomas Fire) that were more than 500 km apart, and evidence suggests that regional urban emissions were not strongly influencing these periods of comparison. Back-trajectories, raw wind data, and satellite imagery show that these measured air masses were coming from off the coast, away from the influence of the LA basin. In other words, these were very carefully chosen time periods in which we did not observe "dynamic mixing" of both fossil fuel and biomass burning rBC. We can even see clearly from the rBC size and coating thickness that these periods (L3 vs. L9)

were dominated by biomass burning.

       We would also like to address the reviewer's concern that other (non-LA) fossil-fuel emission sources, like ship emissions and emissions from other nearby cities, can confound the comparison between aged $BC_{ff}$ and fresh $BC_{ff}$. First, there is a general agreement in existing literature that fossil-fuel emissions produce more thinly-coated, smaller rBC particles (McMeeking et al., 2011; Laborde et al., 2013; Liu et al., 2014; Sahu et al., 2012; Schwarz et al., 2008a; Krasowsky et al., 2018). Second, measurements of ship emission conducted by Corbin et al. (2018) show that fresh rBC from ships, regardless of fuel type, were uncoated to thinly-coated. Corbin et al. (2018) also report a very broad rBC size distribution for ship emissions, which is uncharacteristic of any distribution seen in our data. Third, the winds for the first campaign were almost entirely westerly, making it improbable that emissions from Baja California or San Diego were influencing our measurements in any significant way. For ship emissions or other miscellaneous urban emissions to confound the comparison between fresh and aged $BC_{ff}$ presented in our analysis, the average coating thickness for such rBC particles must be markedly different near the emissions source. Past studies suggest that fossil fuel emissions in general are thinly-coated, even ship emissions. In our data, we observe a remarkable consistency in the $BC_{ff}$ clusters centered ~160 nm MMD in Figures 17a – 17c, with the only major difference being a significantly higher $CT_{BC}$ for the well-aged background period (i.e., first campaign). The only logical explanation is that aging has played a role in increasing the coating thickness for this mode of rBC particles.

(1.3)

*Changes throughout the text. Most significant changes made in sections 3.6 and 3.7.*

===================================

(2.1)

**Black carbon number size distributions. As highlighted in my original review, this reviewer has a major concern with using a CMD from a lognormal curve fit to rBC number size distributions for which there is no obvious peak in the actual number size distribution data. Small changes in distribution width could easily shift the CMD and, in turn, impact how the authors interpret the findings (e.g., lines 602 - 604; "..some periods in which we observed a relative**

**increase in the MMDfit, but concurrent decrease in CMDfit." As stated before, any discussion using derived CMDs for which there is not discernible peak in the actual data must be deemphasized. Also, on line 604, the authors cite 53 nm as the CMD, but according to their text they are referencing "L5" in figure 11, which shows the CMD = 59 nm.**

(2.2)

Although we understand the concern that the reviewer raises, we would like to mention again (similar to our previous response) that any potential shift in the $CMD_{fit}$ from the "hidden," real distribution in the lower range below the detection limit would not change any of our conclusions. For example, additional peaks below 70 nm, or non-lognormal distribution shapes skewed towards smaller particles sizes, would make the actual CMD even smaller, which would only strengthen our conclusions. The reviewer specifically mentions the text describing the concurrent increase in $MMD_{fit}$ and decrease in $CMD_{fit}$. It is very unlikely that any variation of the actual size distribution below the 70 nm detection limit would change our interpretation. In fact, if anything, the actual CMD is more likely to be lower than $CMD_{fit}$, further emphasizing the divergence between CMD and MMD in these LEO periods (L5, L10). More explicit caveat is added to the first paragraph of section 3.6 to describe why $CMD_{fit}$ is appropriate to use in the context of our findings and data. If the reviewer still finds this reasoning unacceptable after reading this, we will just remove all mentions of $CMD_{fit}$ in which the peak is not distinguishable. These points are secondary to the main conclusions of the study. Please refer to comments 21-23 for related discussion. Please also refer to previous discussion of this issue in the first response to comments document on the ACP public interactive discussion.

Link to previous response to comments:

(2.3)

*See tracked changes in section 3.6.*

=================================

(3.1)

**Line 9: presumably the authors mean "composition" when referring to "…coating properties". Please clarify.**

(3.2)

Wording changed.

(3.3)

*These uncertainties are largely due to BC's heterogeneous nature in terms of its spatiotemporal distribution, mixing state, and coating composition*

=================================

(4.1)

**Line 235: Can the authors give a citation for the assertion that the probability density function of lag-time values is**

**often bimodal? As the partitioning of thickly and thinly coated rBC particles is very**

**sensitive to the cut off value (in the present manuscript the authors use 1.8 usec).**

(4.2)

Citation added.

(4.3)

*A probability density function of the lag-time values often results in a bimodal distribution (Moteki and Kondo, 2007; McMeeking et al., 2011b).*

=================================

(5.1)

**Line 239: only DBC > 170 nm; this and discussion later of this (Line 258) should go earlier in Section 2.2, which**

**discusses instrumentation**

(5.2)

Although we appreciate the structural suggestion, we believe that this sentence is appropriate in this section because it is explicitly talking about the lower-bound filter applied during the lag-time analysis.

(5.3)

*No changes.*

=================================

(6.1)

**Line 257: on line 239 the authors use 170 nm; here they use 180 nm? Please try to remain consistent with ranges and sizes if at all possible.**

(6.2)

It could be helpful to have these ranges consistent. But it would take considerable effort to redo the analysis and for little benefit and negligible impact on results. We made sure to explicitly mention the size ranges clearly for each analysis.

(6.3)

*No changes.*

================================

(7.1)

**Figure 4: the figure legend refers to rBC count median diameter (CMD) but the Y-axis on trace "C" is labeled rBC Count Mean Diameter. Please correct the graph axis label.**

(7.2)

Legend label has been changed to be consistent. This is actually the count mean diameter (the mean diameter of all calculated rBC cores above detection limit).

(7.3)

*Figure 4, section 3.2*

================================

(8.1)

**Line 346: They state Section 3.4 in the figure caption but it is really Section 3.5.**

(8.2)

Typo corrected.

(8.3)

*Figure 4 caption changed.*

================================

(9.1)

**Fig. 5c: A pedantic comment: mass and number concentrations on opposite axes from figure 4.**

(9.2)

Labels switched for consistency.

(9.3)

*Figure 5C changed to reflect comment.*

================================

(10.1)

**Lines 359-389: a lot of time discussing previous fBC ratios after stating (line 246) that they couldn't be compared across different studies.**

(10.2)

The sentence at the end of section 2.6 was added as a caveat and we used the phrase, "not necessarily comparable from one study to the next," to suggest that care must be taken to ensure comparisons are logical. In our paper, we believe that comparisons are valid because of the similar sampling conditions in referenced studies, and also because we discuss the relative difference in $f_{BC}$ during periods of varying source influences within our own data.

(10.3)

*No changes.*

==================================

(11.1)

**Figure 6: It is difficult to discern, but this reviewer is concerned that in trace 6B (? as there is not A, B, C label on figure) but it seems as if the lag-time data maybe exhibiting behavior characteristic of particle coincidence as at the positive and negative lag-times seem evenly distributed about zero lag-time. What does the actual full lag-time distribution look like? That is, please redo figure 7 to contain both positive and negative lag-times.**

(11.2)

We believe that coincident particles are unlikely given the concentrations of the measurements. Also, the distribution of dots in panel (b) is not symmetric about zero. The vast majority of particles are actually concentrated in the dark band between 0 and ~8µs. We do not feel it is necessary to redo figure 7 because the main point of the plot is to examine the behavior of particles with negative lag-times, not to examine the distribution of all lag-times.

(11.3)

*No changes.*

==================================

(12.1)

**Fig. 7: This reviewer's concern is that the authors are confounding things here – the different rBC core values come from different sources (each with a wide spread); thus when you plot them you get a difference, but they're arguing it's a causal relation (lines 437-439).**

(12.2)

The fact that particles come from various sources doesn't change the main point of this figure. This is a purely physical comparison between two variables for rBC particles with negative lag-times. The purpose of this figure is to validate a similar observation found in Sedlacek et al. (2015), which hypothesized that smaller particles have decreased fragmentation rates.

(12.3)

*No changes.*

==================================

(13.1)

**Line 457: Not necessarily true: thickly-coated particles could fragment.**

(13.2)

We never deny that this is a possibility, but the comparison of $f_{BC}$ and $f_{lag,neg}$ (as described in second to last paragraph of section 3.4) suggests that thickly-coated particles with a core-shell structure would not fully explain all observations of negative lag-times. Therefore, we find that morphological differences must be playing a role in producing these negative lag-times, as proposed by Dahlkötter et al. (2014) and others.

(13.3)

*No changes.*

==================================

(14.1)

**Line 475: This reviewer would like to see Figure S9 (CT_BC vs f_BC) in the main manuscript rather than in supplemental. Also, how does this slope compare to figure 3 in Subramanian et al., (ACP 10, 2010)?**

(14.2)

The figure has been moved to section 3.5. The figure is not completely comparable because the LEO method was not used for coating thickness quantification in Subramanian et al. (2010), and the data averaging was different. We used 10-minute averaged data points while the figure in Subramanian et al. shows a contour plot from the particle-by-particle data. Therefore, the comparison of slopes in the text seems unnecessary and beside the main point of the plot. The main point is to show that the LEO-derived coating thickness and lag-time derived $f_{BC}$ are significantly correlated, and appropriate to use in conjunction to describe the rBC mixing state.

(14.3)

*Figure S9 is now Figure 9 in main text.*

==================================

(15.1)

**Line 500: Since only rBC particle diameters between 200 and 250 nm are included in the table should be stated in the text as well.**

(15.2)

Added to text.

(15.3)

*Particles with rBC diameters between 200 and 250 nm were used in the comparison of these ten time periods.*

==================================

(16.1)

**Fig. 9: Are these only for 200-250 nm particles, as for the previous figure? If so, state in text and caption.**

(16.2)

Added to caption of figure (now Fig. 10)

(16.3)

*Violin plots that show the distribution of rBC coating thickness values for particles with rBC diameters between 200 and 250 nm, calculated for each LEO time period, L1 through L10.*

==================================

(17.1)

**Fig. 9: The fact that there are an appreciable number of negative coating thicknesses (e.g., L2, L4, L5, L6, L7) should be discussed somewhere. Why/how/what confidence does this leave for the others?**

**What does a negative coating thickness mean? How do you get a negative coating?**

(17.2)

A brief discussion on negative coating thicknesses and associated uncertainties has been added to section 2.7.

(17.3)

*Negative LEO-derived coating thickness values are reported throughout the results and discussion section. These non-physical results are caused by instrument noise from both the incandescence and scattering detectors. The per-particle*

*uncertainty associated with both of these signals result in a spread of coating thickness values that may at times go below zero. As a hypothetical example, a thinly-coated particle that has its scattering cross section underestimated and its rBC mass equivalent diameter overestimated may result in a negative coating thickness value. According to Metcalf et al. (2012), per-particle coating thickness uncertainty is ~40%, with the uncertainty reduced for larger rBC particles. In contrast to per-particle uncertainty, systemic uncertainty, which is the uncertainty associated with the average of a population, is largely*

*caused by the choice of parameters for the rBC core, namely its refractive index and density (Taylor et al., 2015). These systemic errors complicate direct comparison between measurements conducted with different set of parameters assumed for the rBC core, but they do not affect comparisons within the same set of measurements. Negative LEO-derived coating thickness values have been reported in a host of past studies, and further details regarding both per-particle and systemic uncertainties can be found in these studies (Metcalf et al., 2012; Laborde et al., 2013; Krasowsky et al., 2018; Taylor et al.,*

*2015).*

==================================

(18.1)

**Line 535: This reviewer would not expect much diurnal variability in the MBL height over the ocean at Catalina, and even if so, it would be AT MOST probably a factor of 2 – not nearly enough to explain**

**the order of magnitude difference (line 533). The authors should be able to get measurements of this for Catalina or nearby locations from radiosondes, lidar, etc. to show what typical changes are near**

**the coast.**

(18.2)

We do not claim that this is the sole reason for the difference in concentration compared to other periods. We only note that this may be a contributing factor based on the time of measurement. The concentration is also secondarily important to the discussion.

(18.3)

*No changes.*

==================================

(19.1)

**Line 539: They stated already on lines 365, 385, 510 that BB has thicker coating that FF.**

(19.2)

Extraneous sentence deleted.

(19.3)

*Deleted: A number of previous studies have suggested that rBC from biomass burning emissions are generally more thickly-*

*coated (Sahu et al., 2012; Schwarz et al. 2008a; Dahlkötter et al., 2014).*

==================================

(20.1)

**Line 570: This goes in the section that introduced the SP2 way above this.**

(20.2)

We believe that this information is appropriate in this section as quick reference, so that the reader does not have to recall information from a previous section. This puts it directly in context of the discussion at hand.

(20.3)

*No changes.*

==================================

(21.1)

**Line 578: "it is reasonable" … "assuming" – lots of guesses. Not always reasonable, especially if you have a mixture of
BB and FF, you wouldn't expect a single lognormal. Also, at some point you run into individual spherules that cuts
off your lower end. The issue is how robust any such fit is the how usable the CMD is as discussed above. The authors**

**are encouraged to restrict their discussions to MMD, where the experimental data exhibit a peak, and discuss these
values - which the authors do: 594-598 - and remove discussions about CMD (especially for those conditions when the
fit CMD is not support by actual data at the derived peak).**

(21.2)

We did not see bimodal distributions in these ten LEO periods. The bimodal distributions seen in the contour plots (Figure

17) are a result of including all data from the entire campaign period. The LEO periods were specifically chosen to represent periods of interest in which one source (and therefore one mode) was dominating. As previously mentioned in comment 2, the uncertainty regarding CMD for some distributions would ultimately not change the final conclusions made in this section. The bottom-line distinction is that we observed larger rBC cores during biomass burning periods as opposed to fossil fuel dominated periods.

(21.3)

*See tracked changes in section 3.6.*

================================

(22.1)

**Line 607: This reviewer is not convinced of the author's logic: that MMD is expected to change more than CMD.**

(22.2)

The following paragraph that the reviewer is referring to has been deleted from text.

(22.3)

*Deleted:* There were some periods in which we observed a relative increase in the $MMD_{fit}$, but concurrent decrease in the $CMD_{fit}$. For example, L5 exhibits a relatively high $MMD_{fit}$ (~171 nm), which suggests that this was a biomass burning dominated time period, but the $CMD_{fit}$ is the second lowest of all the LEO periods (~59 nm). L10 exhibits a similar pattern. In such seemingly contradictory situations, it is likely that biomass burning aerosols are entrained into a broader urban plume (e.g., from Los Angeles basin). An urban plume with no biomass burning influence is expected to exhibit a very low $CMD_{fit}$. As biomass burning aerosol gets entrained, $MMD_{fit}$ is expected to change more than $CMD_{fit}$ due to its larger rBC core size relative to urban rBC cores. (For a unit increase in diameter, the mass weighting will increase proportionally to the third power, while the size weighting will increase proportionally on a first order basis.) This may potentially explain why in some cases we observe relatively high $MMD_{fit}$ along with relatively low $CMD_{fit}$, though chemical source apportionment would be necessary to definitively support this claim. We make no final verdict on the quantitative probability of this happening; rather we are only acknowledging the possibility of such a scenario through a viable physical phenomenon.

================================

(23.1)

**Line 611: Is there the possibility of smaller, undetected modes? Also, why assume only a single lognormal? As the authors show in their manuscript (figures 15 and 16), evidence is present for two**
**populations in some of the LEO periods.**

(23.2)

A direct comparison cannot be made between the ten discrete LEO periods and the scatter plots shown in figures 15 and 16. The LEO periods vary in time range from ~15 minutes to an hour. The time interval for each subplot in figures 15 and 16 is six hours. Bimodal distributions are apparent in figures 15 and 16 because of the longer time period represented. The shorter LEO periods were specifically representative of one mode.

(23.3)

*See tracked changes in section 3.6*

========================================

(24.1)

**Line 627: The opening sentence "The dominant factors that influence rBC core size (i.e., emission source type and aging) also influence rBC mixing state." in this paragraph is very misleading. It is VERY unlikely that atmospheric aging explains the increase in rBC mode; rather it is the increasing contribution from BB emissions which contain both larger diameter rBC particles that are also more thickly coated. Please reword this sentence.**

(24.2)

Removed language regarding aging in this sentence.

(24.3)

*The dominant factor that influences rBC core size (i.e., emission source type) also strongly influences rBC mixing state. Figure 13 shows a scatter plot of one-minute mean $CT_{BC}$ versus one-minute mean rBC core diameter, using data from all three campaigns.*

========================================

(25.1)

**Line 628: Please state that the data shown in Figure 12 is for all campaigns (it is currently stated in the caption only).**

(25.2)

Edits made to end of sentence.

(25.3)

*The dominant factor that influences rBC core size (i.e., emission source type) also strongly influences rBC mixing state. Figure 13 shows a scatter plot of one-minute mean $CT_{BC}$ versus one-minute mean rBC core diameter, using data from all three campaigns.*

========================================

(26.1)

**Line 629: give r^2 rather than r.**

(26.2)

Changed to $r^2$ in text and figure.

(26.3)

*A positive correlation was found, with $r^2 = 0.30$.*

========================================

(27.1)

**Line 633: As discussed above and what this reviewer considers a major concern of this manuscript, how can aging affect rBC core size? The only mechanism that aging can contribute to the rBC core size change is coagulation. But the concentrations during the sampling periods are much too low to make coagulation important or even a contributing factor. Furthermore, the authors are confounding things. Longer aging timescales "are associated with**

**increases …" Incorrect. Long aging timescales correspond to LARGER (NOT increases in) rBC diameter – one is a difference (larger than) and one has a temporal aspect to it, which cannot be inferred from a measurement from a single location. This Reviewer maintains that larger BC diameters come from fires farther away – with potentially different fuel sources. Thus, the larger diameters could result from different fires, with age having nothing to do with**

**it.).**

(27.2)

Sentence edited. Please see comment 1 for further details.

(27.3)

*The significant correlation suggests that larger contributions from biomass burning (as opposed to fossil fuel) are associated*

*with increases in both the rBC core size and the BC coating thickness.*

=====================================

(28.1)

**Fig. 12: This is not so surprising: they have a number of different sources, each with a spread of values, which, in the limit of pure FF and BB plumes will give FF having lower rBC core diameter and lower CT_BC, and BB having the**

**larger. Then there are essentially two points, with a positive slope. The authors are encouraged to colormap the points into BB and FF data which will likely show this.**

(28.2)

We appreciate the suggestion to colormap the figure, but with figures 15-17 described later on in the text, we believe it is unnecessary.

(28.3)

*No changes.*

=====================================

(29.1)

**Fig. 13a: Somewhat of an increase in coating with core diameter, which makes sense (for the same reason as in the**

**previous comment) – not the same history; Fig. 13b and 13c have a decrease in coating thickness with increasing DBC. It should also be noted that the conclusion of increasing coating thickness with rBC diameter is derived from very weak signals (E, F, G, and H). What are the error bars associated with the derived coating thicknesses?**

(29.2)

The figure has been moved to the supplement to deemphasize it because it is not essential to the section being discussed. It is more of a qualitative supplement to Figure 12, showing that there is some relationship between the rBC core size and $CT_{BC}$ distribution. The main point is to show the broadening of the tail of the $CT_{BC}$ distribution as rBC core size increases.

(29.3)

*Figure is now figure S22 in supplement.*

=====================================

(30.1)

**Fig. 16: To this Reviewer, this figure makes a robust argument that source is what is dominating the rBC mixing state properties and NOT atmospheric processing (i.e., aging).**

(30.2)

We agree that source type is the dominant factor affecting mixing state, and we have changed language throughout text to reflect that. We do however leave some discussion on aging in the text because we do have evidence to suggest that aging played a smaller role. This also agrees with numerous past studies that showed that aged particles had thicker coatings. Citations are not added here because there are numerous citations throughout the manuscript that mentions this.

(30.3)

*No direct changes.*

==================================

(31.1)

**Line 708: The authors assume that all their rBC is either urban or BB, and implicitly seem to assume that the urban is LA basin. However, for their first period the winds were from the west and concentrations were very low. How about ship emissions? There's a rather large shipping route near there, and the rBC from that could look very**

**different from LA basin. Also, if it's quite aged, it could be from San Diego, or Tijuana, which might have a very different mix of diesel/auto/industry than LA, therefore restricting comparisons between their first period and when they know it's from LA basin might not be justifiable.**

(31.2)

In section 3.1, we explicitly mention ships/aviation as potential sources contributing to background measurements during the first campaign (September 2017). We have removed urban in particular spots to avoid confusion that fossil fuel rBC is purely limited to emission from the LA urban basin. For further clarification, we are not assuming "urban" emissions are solely from the LA basin. We are purely making the best assumptions about the source based on physical characteristics that largely match those from fossil fuel emissions (based on previous studies and our source apportionment). For example, we also mention the possibility of long-range transport from East Asia as potential sources of "fossil-fuel" rBC. Other urban areas in Southern California, like Santa Barbara, Ventura, and San Diego are also mentioned in section 3.5. Also, please refer to comment 1 for related discussion.

(31.3)

*Assuming that this smaller mode represents fossil fuel influenced BC (e.g. urban, ship, aviation emissions), this confirms that while $BC_{ff}$ may not become thickly-coated within a day, they seem to acquire coatings over longer timescales.*

==================================

(32.1)

**Line 710: The authors write "In addition to emissions source type, atmospheric aging also appears to have an observable effect on the mixing state." As discussed above, absence a Lagrangian measurement, such a conclusion is**

**not justifiable. Instead, and as highlighted in this review, the likely cause for observed differences is simply differing**
**sources getting mixed together.**

(32.2)

We partially disagree with the reviewer's statement that "Lagrangian" aircraft measurements are the only means to infer the effect of aging on rBC. We acknowledge fully the limitations of a singular, ground-based measurement site, but we believe that meteorological data, satellite data, the SP2 data itself, and auxiliary data sources (e.g. qualitative information regarding
regional sources) can give us good approximations of where particles are coming from at different time periods. Plus, aircraft measurements are not purely Lagrangian and have limitations in the timescales of aging that can be measured. As the plume dilutes, the likelihood of entraining aerosol from multiple sources increases, and the plume itself cannot be followed for days. Thus, aircraft measurements have their advantages, but also associated disadvantages relative to fixed ground measurements.

(32.3)

*See related comments 1, 31, and 34.*

===================================

(33.1)

**Line 710-711: There is not Table 3.**

(33.2)

This is supposed to be Table 2.

(33.3)

*See section 3.5.*

===================================

(34.1)

**Lines 728-730: In the absence of Lagrangian measurements, the authors cannot conclude this. Could be (likely is) different sources.**

(34.2)

We disagree with the statement that this cannot be concluded based on our data and information we gathered. For example,
our comparison of fresh $BC_{bb}$ versus aged $BC_{bb}$ in this section is made between two well-defined periods where biomass burning was identified to be the most dominant source. Referring to Figure S21, it is shown clearly that L3 is likely not affected by the major urban emission source, the LA basin. In fact, the winds are northwesterly, coming off the coast near Santa Barbara, where the Thomas Fire was active during this the entirety of the 2nd campaign (December 2017). As a reminder, this active fire was over 1,000 km², located in both Santa Barbara and Ventura counties, northwest of LA.
Likewise, L9 is also clearly not affected by the regional urban emissions, as the back-trajectory comes in from the west. The giant plume from the Northern California Camp Fire can even be seen visibly from satellite imagery, advecting down off the coast of California, towards the island (see Fig. S7). We acknowledge the uncertainty associated with reliance on meteorological data, back-trajectory modeling and other auxiliary sources of information; but we still think that we can conclude this based on the evidence provided. We have updated the text to explicitly acknowledge the limitations of our methodology and the inherent uncertainties associated with our sources of auxiliary data.

(34.3)

*No direct changes.*

===================================

(35.1)

**Line 743: they're citing an AGU presentation**

(35.2)

This sentence including this reference has been deleted.

(35.3)

*fresh $BC_{bb}$ (< 5 h), providing emerging evidence that rBC coating may not always monotonically increase (Sedlacek et al., 2019).*

===================================

(36.1)

**Line 747 (at end of paragraph): I wrote "much verbage" – lots of words but they didn't say much in this paragraph**

(36.2)

Middle chunk of paragraph deleted. See below.

(36.3)

*Deleted: Previous studies have noted that the competing processes of dilution-driven evaporation and oxidation-driven condensation determine the abundance of organic aerosol relative to carbon monoxide ($\Delta OA/\Delta CO$) in biomass burning*

*plumes (Garofalo et al., 2019). Although the time evolution of $\Delta OA/\Delta CO$ is not necessarily indicative of rBC mixing state evolution, the same physical processes (i.e., evaporation and condensation) must apply to OA rBC coating formation and loss. The conflicting observations from various studies showing $\Delta OA/\Delta CO$ either increasing, decreasing, or staying relatively stable in the near-field (timescale of ~hours) suggest that rBC coating in dense fresh biomass burning plumes may undergo similar competing physical mechanisms.*

===================================

(37.1)

**Line 754: "Assuming …" is FF influence BC necessarily urban? Baja California/ship emissions/etc. are possibilities that aren't urban but that are BB.**

(37.2)

Same comment as above. See comment 31.

(37.3)

*Assuming that this smaller mode represents fossil fuel influenced BC (e.g. urban, ship, aviation emissions), this confirms that while BC_ff may not become thickly-coated within a day, they seem to acquire coatings over longer timescales.*

=====================================

(38.1)

**Line 764: urban BC coagulation – as highlighted several times here and in my original review, the authors can estimate this since they have concentration data. But I think concentrations are too low for this to be a major contributor. Also, the authors assume that period 1 is either urban or BB, and there are other options.**

(38.2)

Whole chunk of text, including the referenced line, has been deleted.

(38.3)

*See tracked changes in section 3.7 to see deleted chunk.*

=====================================

(39.1)

**Line 800: Are there results from HIPPO (or whatever the latest one was) on aged BC? I think there is something out from the NOAA (Murphy's) group, but they didn't cite it.**

(39.2)

We are not sure which study this is in reference to. All the studies that we found that quantified rBC coating thickness have been summarized in Table 3 (which we made sure to include this time).

(39.3)

*See table 3 in section 3.8.*

=====================================

(40.1)

**Page 37: Table 3 missing.**

(40.2)

Included in new revision.

(40.3)

*See section 3.8.*

=====================================

(41.1)

**Line 836: any CO measurements?**

(41.2)

No CO measurements were taken.

(41.3)

*No changes.*

======================================

(42.1)

**Line 842: Again, the authors assume only urban (and specifically LA basin urban) or BB as the only choices.**

(42.2)

Changed language to avoid confusion. See comment 31.

(42.3)

*The bimodal rBC core size distribution from the aged background during the first campaign (September 2017) suggests that background rBC above the Pacific Ocean during typical meteorological conditions are likely a mix of both fossil fuel (i.e., smaller rBC cores) and biomass burning (i.e., larger rBC cores) emissions.*

======================================

(43.1)

**Line 867: The authors assume period 1 is aged urban particles – what about ship exhaust?**

(43.2)

Changed language to avoid confusion. See comment 31.

(43.3)

*By estimating approximate source-to-receptor timescales (i.e., age) and also comparing the physical properties of fresh rBC to that of $BC_{aged,bg}$, we 
[revised manuscript text omitted]

~~necessarily indicative of rBC mixing state evolution, the same physical processes (i.e., evaporation and condensation) must apply to OA rBC coating formation and loss. The conflicting observations from various studies showing ΔOA/ΔCO either increasing, decreasing, or staying relatively stable in the near-field (timescale of ~ hours) suggest that rBC coating in dense fresh biomass burning plumes may undergo similar competing physical mechanisms. Preliminary results from developing research showed that well-aged $BC_{bb}$ (>7 days) had thinner coatings than fresh $BC_{bb}$ (< 5 h), providing emerging evidence~~

Further research is necessary to confirm this process in more field measurements, and to determine the various mechanisms that may be driving the potential loss of rBC coating in biomass burning plumes. We make no definitive claims about the rate of change of $CT_{BC}$ for $BC_{bb}$ throughout atmospheric transport since we measured $CT_{BC}$ at one location. Nonetheless, our measurements suggest that $CT_{BC}$ for fresh Southern California $BC_{bb}$ were generally lower than $CT_{BC}$ for aged Northern California $BC_{bb}$.

The contour plots for the first campaign (September 2017), shown in Fig. 17a and 17d, offer additional perspective on how aging can affect rBC mixing state within well-aged background air masses over longer aging timescales (~days to week). The first notable feature of the $BC_{aged,bg}$ cluster is that the smaller mode is significantly more coated than the $BC_{ff}$ clusters found in Fig. 17 for the second (December 2017) and third (November 2018) campaigns. The peak of the smaller mode of the $BC_{aged,bg}$ cluster is at least ~35 nm higher than the peak of the respective $BC_{ff}$ clusters in Fig. 17b, 17c, 17e, and 17f. Assuming that this smaller mode represents fossil fuel influenced BC (e.g., urban, ship, aviation emissions), this confirms that while  $BC_{ff}$ may not become thickly-coated within a day, they seem to acquire coatings over longer timescales.

166a) 166d) 166b, 17166e) 166c, 17166f) 166a and 17166d 166b, 17166c, 17166e, and 17166f;

 166a and 17166d,  166c and 17166f.

~~wet deposition and/or increased hygroscopicity of thickly-coated rBC could explain this apparent decrease in the overall rBC core size. Moteki et al. (2012) found that larger rBC particles were more effectively removed through wet deposition. McMeeking et al. (2011a) reported that more thickly-coated particles were more hygroscopic, which would in turn lead to a higher likelihood of wet scavenging. Either, or both, of these mechanisms could explain why the peak rBC diameter of the larger mode in the $BC_{aged,bg}$cluster is lower than the peak rBC diameter of the $BC_{bb}$cluster, though we cannot eliminate the~~

[revised manuscript text omitted]

| | | | | measurements in Paris, France | 2010 | |
|---|---|---|---|---|---|---|
| | 99±200–50 | 90-260– | –– | Airborne measurements in the Los Angeles Basin and surrounding outflowsGround-based measurements in London, during periods dominated by traffic sources | May 201031 Jan–1 Feb 2012 | Metcalf, 2012Liu, 2014 |
| | 88±4[b]99±20 | 18090–260 | ~hours– | Ground-based measurements in Gual Pahari, IndiaAirborne measurements in the Los Angeles Basin and surrounding outflows | 3 Apr–14 May 2014May 2010 | Raatikainen, 2015Metcalf, 2012 |
| | 0–4088±4[b] | 200180 | –~hours | Airborne measurements over California during ARCTAS-CARB campaignGround-based measurements in Gual Pahari, India | 15–30 Jun 20083 Apr–14 May 2014 | Sahu, 2012Raatikainen, 2015 |
| | 20±100–40 | 190–210200 | 2–3.5 d– | Airborne measurements over Houston and Dallas, TXAirborne measurements over California during ARCTAS-CARB campaign | 20–26 Sep 200615–30 Jun 2008 | Schwarz, 2008aSahu, 2012 |
| | 30–4020±10 | 200190–210 | ~ 6 h2–3.5 d | Ground-based measurements of fresh emissions from Japan, conducted on Fukue Island, JapanAirborne measurements over Houston and Dallas, TX | Mar–Apr 20720–26 Sep 2006 | Shiraiwa, 2008Schwarz, 2008a |
| | ~60[a]30–40 | 200–250200 | ~days–wk–6 h | Ground-based measurements on Catalina Island (~70 km SW of Downtown LA)Ground-based measurements of fresh emissions from Japan, conducted on Fukue Island, Japan | 7–14 Sep 2017Mar–Apr 207 | This studyShiraiwa, 2008 |
| Remote / background / continental / highly-aged | 130–300–60[a] | 60–80200–250 | –~days–wk | Ground-based measurements in Shanghai, ChinaGround-based measurements on Catalina Island (~70 km SW of Downtown LA) | 5–10 Dec 20137–14 Sep 2017 | Gong, 2016This study |
| | 37–93130–300 | 200–26060–80 | –– | Ground-based measurements in Paris, FranceGround-based measurements in Shanghai, China | 15 Jan–15 Feb 20105–10 Dec 2013 | Laborde, 2013Gong, 2016 |
| | 188±3137–93 | 90-260200–260 | –– | Airborne measurements in the free troposphereGround-based measurements in Paris, France | May 201015 Jan–15 Feb 2010 | Metcalf, 2012Laborde, 2013 |
| | 75–100188±31 | 150–20090–260 | –– | Ground-based measurements at the Pallas GAW (Finnish Arctic)Airborne measurements in the free troposphere | Dec 2011–Jan 2012May 2010 | Raatikainen, 2015Metcalf, 2012 |
| | 90±5[b]75–100 | 180150–200 | ~hours | Ground-based measurements in Mukteshwar, IndiaGround-based measurements at the Pallas GAW (Finnish Arctic) | 9 Feb–31 Mar 2014Dec 2011–Jan 2012 | Raatikainen, 2015Raatikainen, 2015 |
| | 48±1490±5[b] | 190–210180 | –hours | Airborne measurements over Houston and Dallas, TXGround-based measurements in Mukteshwar, India | 20–26 Sep 20069 Feb–31 Mar 2014 | Schwarz, 2008aRaatikainen, 2015 |
| | < 30 nm48±14 | 190–210190–210 | –– | Airborne measurements over Costa Rica, 1-5 kmAirborne measurements over Houston and Dallas, TX | 6–9 Feb 200620–26 Sep 2006 | Schwarz, 2008bSchwarz, 2008a |
| | 20–36< 30 nm | 160–180190–210 | –– | Ground-based measurements in Alert, Nunavut, Canada (within Arctic Circle)Airborne measurements over Costa Rica, 1-5 km | Mar 2011–Dec 20136–9 Feb 2006 | Sharma, 2017Schwarz, 2008b |
| | ~6020–36 | 200160–180 | ~days– | Ground-based measurements of Asian continental air masses, conducted on Fukue Island, JapanGround-based measurements in Alert, Nunavut, Canada (within Arctic Circle) | Mar–Apr 207Mar 2011–Dec 2013 | Shiraiwa, 2008Sharma, 2017 |
| | ~60 | 200 | ~days | Ground-based measurements of Asian continental air masses, conducted on Fukue Island, Japan | Mar–Apr 207 | Shiraiwa, 2008 |

[a] The range of values shown represent the approximate range of the mean $CT_{BC}$.

[b] The absolute coating thickness was calculated from the ratio of rBC core diameter to particle mobility diameter as presented in the study.

Note: A dash ("-") indicates that the value was not reported, or it could not be identified

**4 Conclusion**

This study investigates the concentration, size distribution, and mixing state of rBC on Catalina Island (~70 km southwest of Los Angeles) using a single-particle soot photometer (SP2). Measurements were taken during three separate campaigns with varying meteorological conditions and emission sources, in September 2017, December 2017, and November 2018. During the first campaign (7 to 14 September 2017), westerly winds dominated and thus the sampling location was upwind of the dominant regional sources of BC (i.e., urban emissions from the Los Angeles basin). The measurements from the first campaign were largely characteristic of well-aged background levels of rBC over the Pacific Ocean, away from the broader urban Los Angeles plume (BC$_{aged,bg}$). During the second and third campaigns (20 to 22 December 2017, 12 to 18 November 2018), due to atypical Santa Ana wind conditions, we  measured biomass burning rBC (BC$_{bb}$)  from large wildfires in California and fossil fuel rBC (BC$_{ff}$) from the Los Angeles basin. Furthermore, during the third campaign, rBC from the Camp Fire in Northern California was measured, allowing us to compare the mixing state of aged  BC$_{bb}$ (from Camp Fire) to fresher rBC (from Southern California fires and urban Los Angeles emissions). The measurements from these three campaigns showed that rBC physical properties (rBC core size and mixing state) were influenced by (i) emissions source type, and (ii) atmospheric aging.

BC$_{bb}$ generally had larger core diameters than BC$_{ff}$. The MMD [CMD] of BC$_{bb}$ was observed to be ~180 nm [120 nm], while MMD [CMD] of BC$_{ff}$ was observed to be ~160 nm [100 nm]. BC$_{aged,bg}$ showed a bimodal rBC core size distribution, with MMD [CMD] peaks at ~170 nm [115 nm] for the larger mode, and ~153 nm [109 nm] for the smaller mode. The bimodal rBC core size distribution from the aged background during the first campaign (September 2017) showed that background rBC above the Pacific Ocean during typical meteorological conditions were likely from a mix of both fossil fuel  and biomass burning  emissions. somewhere between the source and receptor, and/or (ii) the initial source and combustion conditions for BC$_{aged,bg}$ were different than for BC$_{ff}$ in this study. More accurate methods of source apportionment would be needed to quantify the relative contribution of each factor, but both factors are viable mechanisms that can affect fossil fuel we attribute the emissions source to be dominant influencing factor with respect to rBC core size, in this study ~~particles. The CMD [MMD] of the larger mode for BC$_{aged,bg}$ was smaller than that of BC$_{bb}$, which suggests that (i) selective wet deposition of larger particles, and/or (ii) increased wet scavenging of thickly-coated particles due to increased hygroscopicity, contributes to the shift in the rBC size distribution for biomass burning rBC-containing particles over long-range atmospheric transport.~~

[revised manuscript text omitted]